# MODEL EQUALITY TESTING: WHICH MODEL IS THIS API SERVING?

**Irena Gao, Percy Liang, Carlos Guestrin**
irena@cs.stanford.edu, pliang@cs.stanford.edu, guestrin@stanford.edu
Stanford University

## ABSTRACT

Users often interact with large language models through black-box inference APIs, both for closed- and open-weight models (e.g., Llama models are popularly accessed via Amazon Bedrock and Azure AI Studios). In order to cut costs or add functionality, API providers may quantize, watermark, or finetune the underlying model, changing the output distribution — possibly without notifying users. We formalize detecting such distortions as Model Equality Testing, a two-sample testing problem, where the user collects samples from the API and a reference distribution, and conducts a statistical test to see if the two distributions are the same. We find that tests based on the Maximum Mean Discrepancy between distributions are powerful for this task: a test built on a simple string kernel achieves a median of 77.4% power against a range of distortions, using an average of just 10 samples per prompt. We then apply this test to commercial inference APIs from Summer 2024 for four Llama models, finding that 11 out of 31 endpoints serve different distributions than reference weights released by Meta.

## 1 INTRODUCTION

Since running a large language model requires compute and technical expertise, many users rely on black-box APIs to handle inference. This applies to both closed-weight models, like GPT and Claude, and open-weight ones: e.g., Amazon Bedrock, Microsoft Azure, and the seven other companies in Figure 1 all compete to offer Llama models as a service. While users can sample from black-box APIs, they have little to no insight into the underlying implementation of the model, including into questions like:

1. **How has the API modified the language model's distribution?** To drive down costs, API providers may quantize or prune large model weights; they may also watermark outputs or incorrectly implement some decoding parameters. These changes distort the resulting distribution of completions. The problem is when such distortions are undisclosed: users assume that calling a third-party API is exactly equivalent to working with the original model. For example, benchmarks like HELM (Liang et al., 2022) evaluate models through third-party APIs, but quantized or watermarked models may be less capable than the intended model.

2. **Is the API changing over time?** Language model inference endpoints may also drift over time without notifying users (Chen et al., 2023; Eyuboglu et al., 2024), e.g., due to finetuning or updates to the inference stack. Unstable APIs affect research reproducibility (Pozzobon et al., 2023) and can disrupt user productivity in human-AI teams (Bansal et al., 2019).

Under the status quo, neither users nor regulators have a way to rigorously answer these questions for themselves.[1] These concerns are important to address: tens of thousands of developers already rely on black-box inference APIs for applications (Amazon, 2024), and this dependence will increase as LLMs —

---

[1]These problems are already experienced by users: e.g., see threads 1, 2 and 3.

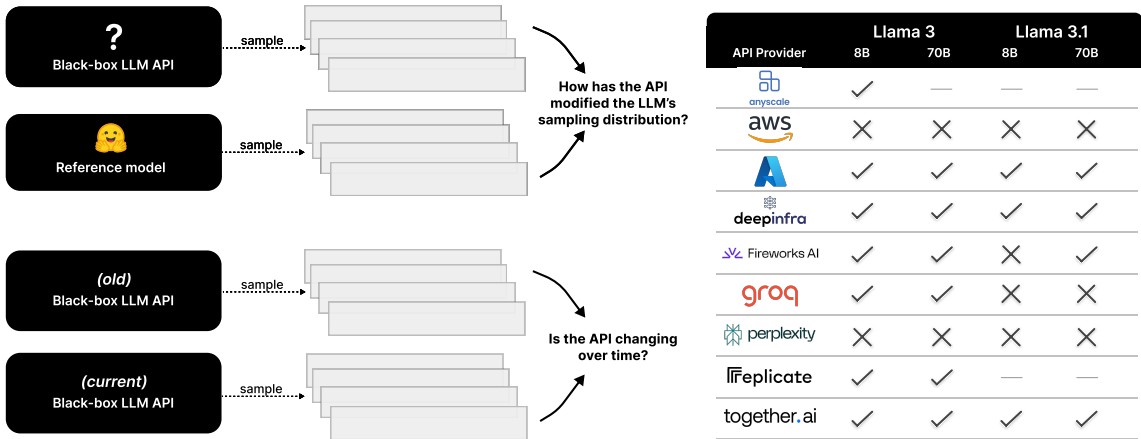

Figure 1: *(Left)* We formalize auditing black-box language model inference APIs as Model Equality Testing. This enables us to assess an API's faithfulness to a reference distribution and its stability over time. *(Right)* We evaluate candidate tests and apply the most powerful one to Llama model APIs from Summer 2024, finding that 11 of 31 endpoints deviate from reference weights released by Meta.

and the corresponding infrastructure costs for hosting — grow larger. For example, most users *must* rely on third-party APIs to use Llama 3.1 405B because of its size.

The current approach to this problem is for an outside auditor to monitor APIs' accuracies on multiple-choice or short-answer benchmarks (ArtificialAnalysis, 2024); these studies typically decode from the language model greedily. Such audits can be a poor match for user needs. Greedy decoding only checks that the modes of the next-token distributions match, rather than the overall distribution over completions, which is problematic because users often *sample* from models. Short-answer benchmarks cover only a small slice of possible prefixes, which may significantly differ from a particular user's task: e.g., common applications like code generation, dialogue, and summarization are longform tasks. Ideally, users could personally audit APIs on their custom tasks. Such a method should be sample-efficient, apply to tasks without automated verifiers, and assess with confidence whether the *overall* distribution of completions has shifted in a statistically significant way.

We provide such a method. Suppose a user wishes to audit a an API on their task of interest. The user collects two samples: one from a reference distribution $P$ and one from the test API's distribution $Q$. For example, to answer if an API has modified the distribution of an open-weight model, $P$ might be from reference model weights released on Hugging Face. To answer if the API is changing over time, $P$ might be from the API at an earlier point in time (Figure 1 left). The user then conducts a two-sample test for whether $P = Q$ to examine if the API's distribution is statistically indistinguishable from the reference.

Our setting is challenging because the distributions being compared are high-dimensional: they are defined over multi-token completions from large vocabularies. Two-sample kernel tests based on estimating the Maximum Mean Discrepancy (MMD) between $P$ and $Q$ (Gretton et al., 2012) are flexible tools for this setting, as they allow us to specify a featurization to reduce dimensionality. We find that a simple string kernel based on the Hamming distance between completions is particularly sample-efficient. In simulations (§4), this test achieves a median of 77.4% power against a wide range of distortions — e.g., quantization, watermarking, and finetuning — using an average of just 10 samples per prompt for distributions over 20–25 prompts. We then apply this test to nine commercial inference API providers across four Llama models (§5, Figure 1 right). Our test flags 11 out of these 31 endpoints, with each audit costing less than $1.

Because our test statistic is an estimate of a distance, we also explore how this machinery can quantify statistical distances between black-box endpoints. In §4.3, we estimate pairwise distances between the output distributions of 13 language models — without requiring log probability access — and find that models within the same family (e.g., the Llama family or GPT-3.5 family) output more similar distributions than models within the same size range (e.g., 7B or 70B models). In §5, we estimate the effect size of deviations between API endpoints and reference weights, finding that some implementations are further from the reference weights than if the provider had substituted in an entirely different language model.

**Summary of contributions.**   We unify several API auditing tasks under the formalization of Model Equality Testing, a two-sample distribution testing problem, and empirically validate kernel-based tests for this problem. We then apply this test to audit popular commercial inference APIs. To enable users to audit APIs for custom applications, we open-source a Python package. We also encourage future research in Model Equality Testing by releasing a dataset of 1 million LLM completions from five models.[2]

## 2   THE MODEL EQUALITY TESTING PROBLEM

Suppose an auditor is interested in a task parameterized by a distribution $\pi$ over $m$ prompts and a maximum completion length of $L$ tokens. The auditor has sample access to a reference distribution $P$ and API distribution $Q$, both operating on the same vocabulary $\mathcal{V}$ with the same decoding parameters. The auditor samples $N$ prompt-completion pairs[3] $z := (x, y)$ from each distribution:

$$\begin{aligned}
\mathcal{D}_P &:= \{z^{(i)} : x^{(i)} \sim \pi, y^{(i)} \sim P(\cdot \mid x^{(i)})\}_{i=1}^{N}, \\
\mathcal{D}_Q &:= \{z^{(i)} : x^{(i)} \sim \pi, y^{(i)} \sim Q(\cdot \mid x^{(i)})\}_{i=1}^{N}.
\end{aligned} \tag{1}$$

We wish to use these samples to test if $P = Q$, i.e., distinguish between the hypotheses

$$\begin{aligned}
H_0 : &\quad \pi(x)P(y \mid x) = \pi(x)Q(y \mid x), \\
H_1 : &\quad \pi(x)P(y \mid x) \neq \pi(x)Q(y \mid x).
\end{aligned} \tag{2}$$

We require that the Type-1 error rate is controlled at $\alpha$. A good test will maximize power against unknown $Q$ and generalize across several language models and prompt distributions $\pi$. We are particularly interested in *sample-efficient* tests that are cheap to run: such tests are powerful when $N$ is small compared to the size of the vocabulary $|\mathcal{V}|$ and the maximum completion length $L$.[4] The latter parameters modulate the size of the space that the joint distributions are defined over: the set of all prompt-completion pairs has size $m|\mathcal{V}|^L$, where $m$ is the number of prompts captured in $\pi$. Effective tests must navigate this high-dimensional space well. Fortunately, we expect the distributions in practice to be lower-dimensional, as language typically only places significant mass on a small number of tokens at each position.

Throughout the paper, we will use $\mathbf{n}_s\{S\}$ to denote the count of object $s$ in string or sample $S$, $\pi P$ to denote the joint distribution of prompts and completions under $P$, and $\pi Q$ to denote the joint distribution under $Q$.

## 3   METHOD

To tackle the problem, we employ a two-sample kernel test from Gretton et al. (2012). This test uses samples $\mathcal{D}_P$ and $\mathcal{D}_Q$ to estimate the Maximum Mean Discrepancy (MMD) between $P$ and $Q$, which is a measure of the distance between the two distributions. Intuitively, if the estimated MMD is large, we reject the null hypothesis that $P = Q$.

---

[2]Package, experiment code, and dataset: `https://github.com/i-gao/model-equality-testing`.

[3]In Appendix C.6, we discuss how to extend this setup to unequal sample sizes.

[4]As a concrete example, Llama-3 uses a vocabulary size of $|\mathcal{V}| = 128\,256$, and users often sample $L = 250$ tokens for longform generation tasks.

The MMD is defined with respect to a unit-norm kernel function $k$ and its associated feature map $\phi$. For our two joint distributions $\pi P$ and $\pi Q$, the MMD is defined as the squared distance between the expected features from each distribution:

$$
\begin{aligned}
\text{MMD}_k\left(\pi P, \pi Q\right) &= \left\|\mathbb{E}_{z \sim \pi P}\left[\phi(z)\right] - \mathbb{E}_{z \sim \pi Q}\left[\phi(z)\right]\right\|^2, \\
&= \mathbb{E}_{z,z' \sim \pi P}\left[k(z, z')\right] + \mathbb{E}_{z,z' \sim \pi Q}\left[k(z, z')\right] - 2\mathbb{E}_{z \sim \pi P, z' \sim \pi Q}\left[k(z, z')\right].
\end{aligned}
\tag{3}
$$

For simplicity, we select kernels of the form $k(z, z') = \mathbf{1}\{x = x'\}\tilde{k}(y, y')$, where $\tilde{k}$ is a prompt-agnostic kernel over completions. Then $\text{MMD}_k(\pi P, \pi Q) = \mathbb{E}_{x \sim \pi}\left[\text{MMD}_{\tilde{k}}\left(P(y \mid x), Q(y \mid x)\right)\right]$.

To conduct a two-sample test with samples $\mathcal{D}_Q$ and $\mathcal{D}_P$, the test statistic is the empirical estimator of (3):

$$
\widehat{\text{MMD}}(\mathcal{D}_Q, \mathcal{D}_P) = \frac{1}{N(N-1)}\left[\sum_{z,z' \in \mathcal{D}_P} k(z, z') + \sum_{z,z' \in \mathcal{D}_Q} k(z, z')\right] - \frac{2}{N^2}\sum_{z \in \mathcal{D}_P}\sum_{z' \in \mathcal{D}_Q} k(z, z').
\tag{4}
$$

We can compute p-values by simulating the test statistic's distribution under the null, i.e., by repeatedly sampling both $\mathcal{D}_Q$ and $\mathcal{D}_P$ from $P$ and computing (4). Alternatively, to avoid drawing extra samples from $P$, we can use the permutation procedure (Lehmann et al., 1986), at a potential cost to power. This procedure repeatedly shuffles samples between $\mathcal{D}_Q$ and $\mathcal{D}_P$ to recompute the test statistic (Appendix A.2).

**Kernel choice.** The choice of kernel $(k, \phi)$ determines the test's semantics and power. For example, setting $\phi(z)$ to be an indicator of whether $y$ passes an automated verifier for $x$, when one is available, leads to rejecting the null when $P, Q$ result in different task accuracies. However, because $\phi$ is relatively coarse, the MMD may be zero even when $P \neq Q$, limiting the test's power. At the other extreme are *universal* kernels, which guarantee that the MMD is zero if and only if $P = Q$ (Gretton et al., 2012). One such universal kernel for strings is the computationally expensive all-substrings kernel (Borgwardt et al., 2006):

$$
\tilde{k}_{\text{all}}(y, y') = \sum_{s \in \mathcal{V}^{\leq L}} \mathbf{n}_s\{y\} \cdot \mathbf{n}_s\{y'\},
\tag{5}
$$

where $\mathbf{n}_s\{y\}$ is the number of times $s$ appears in $y$. Another universal kernel is the one-hot kernel:

$$
\tilde{k}_{\text{one-hot}}(y, y') = \mathbf{1}\{y = y'\},
\tag{6}
$$

which results in a classical two-sample multinomial test between the joint distributions. While universal kernels can eventually detect differences between any $P, Q$ with enough samples, they may have low power in the small-sample regime. For example, the one-hot MMD measures if there are more exact match collisions within $\mathcal{D}_P$ or $\mathcal{D}_Q$ than between them, but in small samples, we may see no duplicate completions at all.

We posit that other string kernels, though not universal, provide more powerful features for testing with small samples. Specifically, we investigate a fast kernel related to the Hamming distance between completions:

$$
\tilde{k}_{\text{hamming}}(y, y') = \sum_{i=1}^{L} \mathbf{1}\{y_i = y_i'\},
\tag{7}
$$

where $y$ shorter than $L$ is right-padded with a special token. Intuitively, a test based on this kernel rejects if a significantly larger number of substitutions are needed to align completions between $\mathcal{D}_P$ and $\mathcal{D}_Q$ than within each sample. The associated Hamming MMD is a pseudo-metric, as it is zero when $P = Q$ and obeys the triangle inequality, but may not separate all distributions (Appendix A.1). Despite this limitation, we find in the following sections that this kernel is empirically effective and well-suited to common distortions we encounter with language models: quantization, watermarking, finetuning, and related distortions tend to result in detectable inter-sample Hamming distances.

## 4 EVALUATING TESTS IN SIMULATIONS

In this section, we evaluate our test's power using different kernels at checking equivalence between known pairs of distributions. Specifically, we evaluate if tests can detect when a language model has been quantized or watermarked (§4.1), finetuned (§4.2), or swapped out for a different model altogether (§4.3).

All experiments in this section are run on a longform language modeling task. The prompt distribution $\pi$ is a uniform distribution over $m = 25$ random 100-character strings sampled from English, German, Spanish, French, and Russian Wikipedia (Box 1). The maximum completion length is $L = 50$, and we sample using temperature 1. Power is computed from 100 Monte Carlo simulations. We estimate p-values by simulating the empirical distribution of the test statistic under the null 1000 times; in Appendix C, we validate that the permutation procedure results in the same trends.

---

**Box 1: Sample prompt for the Wikipedia language modeling task**

Continue the paragraph. Do not output anything except the continuation to the paragraph. Start the continuation immediately.
"The British Arab Commercial Bank PLC (BACB) is an international wholesale bank incorporated in the U..."

---

To evaluate tests' generalization across prompts, we repeat power experiments over ten different prompt distributions, where we resample 25 Wikipedia strings for each $\pi$. All tests are conducted at a significance level of $\alpha = 0.05$. Additional details can be found in Appendix B.

### 4.1 DETECTING QUANTIZATION AND WATERMARKING

In our first experiments, the reference distribution $P$ represents full-precision weights published on Hugging Face. We evaluate if tests can distinguish $P$ from alternative distributions $Q$:

- **Quantized models.** These alternatives represent the model inferenced at lower precisions: nf4 (Dettmers et al., 2024), int8 (Dettmers et al., 2022), and fp16. Some accounts suggest that quantization particularly degrades Llama-3 models on longform tasks (Reddit, 2024; Panda, 2024).

- **Watermarked models.** Some providers may watermark outputs so that they are later detectable as having been generated by the platform. We apply the watermarking algorithm from Kirchenbauer et al. (2023) with default bias of 2.5.

We repeat evaluations for 5 instruction-tuned models: Mistral 7B Instruct (Jiang et al., 2023), Llama-3 8B and 70B Instruct, and Llama-3.1 8B and 70B Instruct (Meta, 2024). Additional models are in Appendix C.1.

**Tests.** We compare three choices of kernels: the Hamming kernel (7), the all-substrings kernel (5), and the one-hot kernel (6). We also evaluate two tests from the multinomial testing literature (Balakrishnan & Wasserman, 2018; Bhattacharya & Valiant, 2015):

$$\mathrm{L1}(\mathcal{D}_P, \mathcal{D}_Q) = \sum_{z \in \mathcal{D}_P \cup \mathcal{D}_Q} \left| \frac{\mathbf{n}_z\{\mathcal{D}_P\} - \mathbf{n}_z\{\mathcal{D}_Q\}}{N} \right| \tag{8}$$

$$\chi^2(\mathcal{D}_P, \mathcal{D}_Q) = N^2 \sum_{z \in \mathcal{D}_P \cup \mathcal{D}_Q} \frac{(\mathbf{n}_z\{\mathcal{D}_P\} - \mathbf{n}_z\{\mathcal{D}_Q\})^2 - \mathbf{n}_z\{\mathcal{D}_P\} - \mathbf{n}_z\{\mathcal{D}_Q\}}{\mathbf{n}_z\{\mathcal{D}_P\} + \mathbf{n}_z\{\mathcal{D}_Q\}}. \tag{9}$$

**Results.** Figure 2 (left) compares the empirical sample complexities of each test. To draw out a sample complexity curve, we vary the number of samples from $N = 10m$ to $N = 100m$, where $m = 25$ is the number of prompts in the prompt distribution. We observe that the Hamming MMD test attains the highest power with the fewest samples: at an average of 10 samples per prompt, this test has a median power

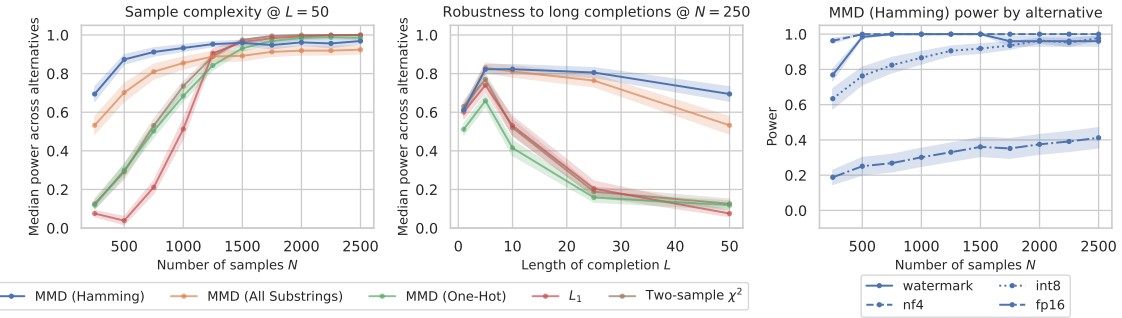

Figure 2: *(Left)* Sample complexity of tests. At an average of just 10 samples per prompt, the Hamming MMD test is able to detect quantization and watermarking with nontrivial power. Curves first median power across alternative distributions $Q$, averaged over language models and prompt distributions, with shaded standard errors. Results stratified by language model and alternative are in Appendix C.1. *(Middle)* While other tests rapidly degrade in power when the user is interested in longer completions, the Hamming MMD test maintains power best across completion lengths. *(Right)* Power of the Hamming MMD test, stratified by alternative distribution. The test is significantly less powerful against the fp16 alternative.

of 77.4% across alternatives. In Figure 2 (right), we break down power by alternative distribution. The Hamming test is strong on all alternatives except fp16, where the initial power at $N = 10m$ is much smaller. This suggests that fp16 and fp32 differ in ways that are not captured by the Hamming kernel.

To accommodate user tasks which require very long completions, it is important that tests retain power as the completion length $L$ increases, even though the size of the sample space grows exponentially with $L$. In Figure 2 (middle), we fix $N = 250$ (i.e., $N = 10m$) and vary the completion length $L$ from 1 to 50 tokens. We observe that the Hamming MMD and all substring tests are more robust to increasing completion length than the other tests. This result is consistent with the intuition that a clever string kernel — as opposed to a one-hot kernel — can help MMD tests generalize to high-dimensional spaces.

## 4.2 DETECTING FINETUNING

Given their effectiveness in detecting quantized and watermarked models, we next ask if MMD-based tests can detect when a model has been finetuned. We finetune Llama-3 8B Instruct on two datasets: a disjoint, *i.i.d.* split of the testing Wikipedia task, and an out-of-distribution code dataset (Chaudhary, 2023). We use a small learning rate of $1 \times 10^{-6}$ with AdamW (Loshchilov, 2017). We then use the Hamming MMD test to compare finetuned checkpoints $Q$ to the original model as the reference distribution $P$.

Figure 3 (upper left) plots power against the checkpoint number of the finetuned model. The Hamming MMD test is always able to detect finetuning with nontrivial (greater than 50%) power, even after a single epoch (42 optimization steps). One might expect that finetuning on the out-of-distribution code dataset would not affect the model's distribution on the Wikipedia testing task, but we find this is not the case. Finetuning affects the model on other distributions enough to be detectable by statistical tests. These results suggest that it is challenging to isolate the effects of full finetuning to any single distribution, which may have implications for tasks such as unlearning or model editing (Hase et al., 2024).

## 4.3 DISTINGUISHING IF SAMPLES COME FROM DIFFERENT LANGUAGE MODELS

We next explore if the Hamming MMD test can distinguish whether two bodies of text are generated from the same language model. In this setting, $P$ and $Q$ are two different language models, drawn from a pool of 13 instruction-tuned models (Figure 3 right), including eight open-weight models (Abdin et al., 2024; Groeneveld et al., 2024; Team et al., 2024) and and five OpenAI closed-weight models. In order to compare

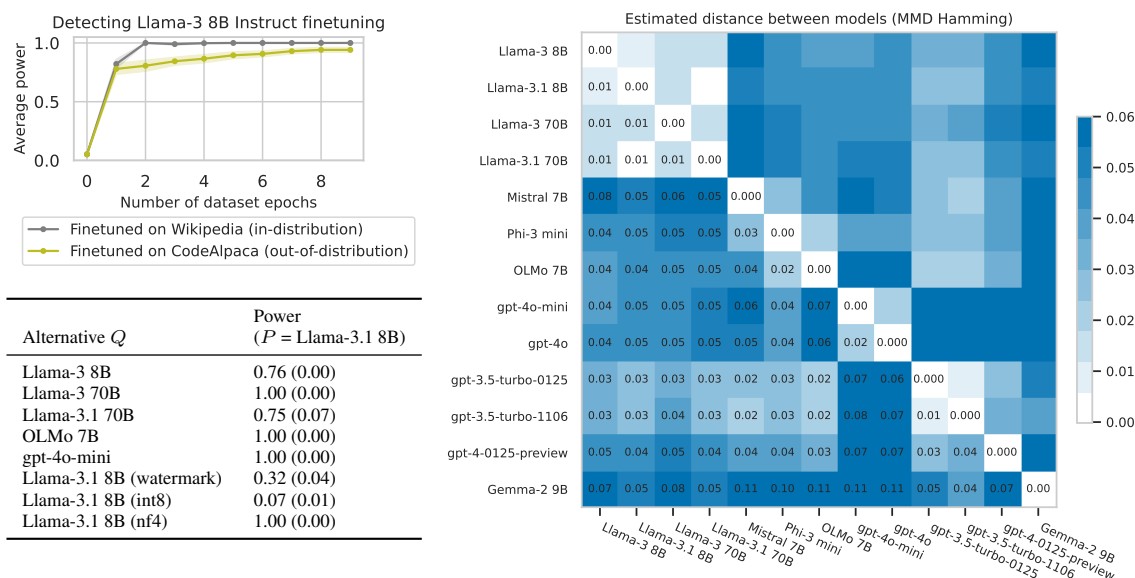

| Alternative $Q$ | Power ($P$ = Llama-3.1 8B) |
|---|---|
| Llama-3 8B | 0.76 (0.00) |
| Llama-3 70B | 1.00 (0.00) |
| Llama-3.1 70B | 0.75 (0.07) |
| OLMo 7B | 1.00 (0.00) |
| gpt-4o-mini | 1.00 (0.00) |
| Llama-3.1 8B (watermark) | 0.32 (0.04) |
| Llama-3.1 8B (int8) | 0.07 (0.01) |
| Llama-3.1 8B (nf4) | 1.00 (0.00) |

Figure 3: *(Upper left)* The Hamming MMD test is able to detect when Llama-3 8B has been finetuned on datasets of 1000 samples, even after a single epoch. Power is higher, earlier, when the finetuning distribution is *i.i.d.* with the testing distribution. *(Lower left)* The Hamming MMD test can also detect when two models are different with near-perfect power. Standard errors are over prompt distributions. Full results are in Appendix C.2. *(Right)* The MMD framework allows us to estimate statistical distance between any models from which we can draw samples. The cells show average estimated MMDs over 10 bootstraps. Rows are sorted using spectral clustering with two components. Models within a family are typically clustered together, suggesting that factors like training data, rather than scale, determine model similarity.

models with different tokenizers, samples must be compared in character space. We sample $L = 50$-token completions to the Wikipedia language modeling task as before, and then we decode these to characters, ignoring special tokens. The vocabulary of interest is now all Unicode characters ($|\mathcal{V}| = 155\,063$), the maximum completion length is $L \approx 1000$ characters, and we test with $N = 10m$ samples.

Switching to character space makes the testing problem significantly more challenging, as the sample space is higher dimensional: Figure 3 (lower left) shows that power to detect quantization and watermarking is lower in this setting than in the token space setting (§4.1). However, even in this more challenging setting, we find that all model swaps are detectable with 100% power, except for pairs within the Llama family, e.g., Llama-3.1 8B and Llama-3.1 70B (75% power) or Llama-3 8B and Llama-3.1 8B (76% power; Figure 3 lower left). Model swaps are significantly more detectable than watermarking or quantization. Appendix D.2 shows qualitative examples of completions from different models.

**Estimating distances between models.** A useful feature of the MMD tests is that the test statistic is an estimator of a distance. As a result, we can reuse the machinery to quantify the *degree* to which two models differ by estimating $\mathbb{E}_{x \sim \pi}[\text{MMD}(P(\cdot \mid x), Q(\cdot \mid x))]$.[5] Figure 3 (right) estimates the Hamming MMD between all pairs of models. To decrease estimator error, we increase the sample size to $N = 100m$ samples from each model and report the average $\widehat{\text{MMD}}$ over 10 simulations, along with standard errors. We observe that models within a family are typically clustered together, suggesting that training data, rather than

---

[5]Note that the Hamming MMD is a pseudometric: $\text{MMD}_{\text{Hamming}}(P, Q) = 0$ does not imply $P = Q$ (Appendix A.1).

scale, determines model similarity. Surprisingly, while generations of Llama models (3 and 3.1) are close in distance, some generations of GPT models (e.g., 4-preview and 3.5-turbo) are not. This result suggests the use of different training data or procedures between these models.

## 5 AUDITING INFERENCE API PROVIDERS

As a case study, we now apply our test to 31 commercial inference endpoints for four of Meta's Llama models (Figure 1). These endpoints are distributed across nine API providers from Summer 2024: Amazon Bedrock, Anyscale,[6] Azure AI Studio, Deepinfra, Fireworks AI, Groq, Perplexity, Replicate, and Together.ai.[7] We are interested in whether endpoints differ from weights published by Meta on Hugging Face inferenced at commonly accepted precisions. Specifically, we consider two possible null distributions: the full-precision weights ($P_1$) and the fp16-precision weights ($P_2$). The null and alternative hypotheses are

$$
\begin{aligned}
H_0: \quad & \pi(x)P_1(y \mid x) = \pi(x)Q(y \mid x) \quad \text{OR} \quad \pi(x)P_2(y \mid x) = \pi(x)Q(y \mid x), \\
H_1: \quad & \pi(x)P_1(y \mid x) \neq \pi(x)Q(y \mid x) \quad \text{AND} \quad \pi(x)P_2(y \mid x) \neq \pi(x)Q(y \mid x).
\end{aligned}
\tag{10}
$$

To test this composite hypothesis, we collect three samples: $\mathcal{D}_{P_1}$ and $\mathcal{D}_{P_2}$ from the two null distributions, and $\mathcal{D}_Q$ from the API. We then conduct 2 two-sample tests, one for $(\mathcal{D}_{P_1}, \mathcal{D}_Q)$ and another for $(\mathcal{D}_{P_2}, \mathcal{D}_Q)$, and obtain p-values $p_1$ and $p_2$. We set the overall rejection rule to

$$
\delta(\mathcal{D}_{P_1}, \mathcal{D}_{P_2}, \mathcal{D}_Q) = \mathbf{1}\{p_1 < \alpha \wedge p_2 < \alpha\}.
\tag{11}
$$

This rule continues to control the FPR at $\alpha$ under the composite null hypothesis: without loss of generality, suppose $Q = P_1$. Since $P_{Q=P_1}(p_1 < \alpha) \leq \alpha$, we have $P_{Q=P_1}(p_1 < \alpha \wedge p_2 < \alpha) = P_{Q=P_1}(p_1 < \alpha)P_{Q=P_1}(p_2 < \alpha \mid p_1 < \alpha) \leq \alpha$. Note that this rule may be pessimistic, reducing power.

**Experiment details.** We consider testing with three prompt distributions $\pi$. For all models, we test with one set of the Wikipedia completion task from §4, where $\pi$ is uniform over $m = 25$ prompts, and $L = 50$ tokens or around 1000 characters. For the smaller Llama-3 8B and Llama-3.1 8B models, we also test with the coding task HumanEval (Chen et al., 2021b) and instruction task UltraChat (Ding et al., 2023). Both $\pi$ are uniform over $m = 20$ prompts, and $L = 250$ tokens or 3000 characters. Because APIs often return decoded completions, rather than individual tokens, we conduct all tests in character space, as in §4.3. We explicitly requested all samples at temperature 1. Tests are conducted at level $\alpha = 0.01$ using $N = 10m$ samples. To reduce variance, we repeat tests over ten samples and fail endpoints if the average rejection rate is $\geq 0.5$. For the most expensive endpoint (Azure's Llama-3 70B), a Wikipedia audit costs \$0.14, HumanEval \$0.83, and UltraChat \$0.93. For the cheapest endpoint (Fireworks' Llama-3 8B), all three audits cost less than \$0.02. For additional details, including the dates we collect API samples, see Appendix B.

**Results.** Despite power being generally reduced due to the composite decision rule, the test flags several endpoints (Table 1).[8] Notably, Amazon Bedrock and Perplexity have the most endpoints flagged, with the latter failing all tests. In Box 2, we include an example comparing samples from the fp32 null and Perplexity; these samples suggest that Perplexity serves a lower entropy distribution than the full-precision model. Additional qualitative samples can be found in Appendix D.3. The HumanEval and UltraChat prompt distributions elicit more failures than the Wikipedia distribution; this may be because these knowledge-intensive distributions are more sensitive to changes such as quantization.

To estimate the effect size of deviations, we estimate the MMD between providers and the nulls using ten bootstraps of $N = 100m$ samples each. We find that some deviations are quite large: some APIs'

---

[6]We collected samples from Anyscale's serverless endpoints from before they were deprecated in August 2024.

[7]At the time of writing, only two providers disclosed distribution-altering optimizations: Fireworks AI noted semantic caching, and Together.ai noted quantization.

[8]In Figure 1, we combine the results of the tests on the prompt distributions via a Bonferroni correction, setting the level of each test to $\alpha = 0.01/3$ and rejecting if the endpoint fails on any of the three prompt distributions.

| | Wikipedia | | | | HumanEval | | UltraChat | |
| | 3 8B | 3.1 8B | 3 70B | 3.1 70B | 3 8B | 3.1 8B | 3 8B | 3.1 8B |
|---|---|---|---|---|---|---|---|---|
| Amazon | ✓ | ✗ | ✓ | ✗ | ✓ | ✗ | ✗ | ✗ |
| Anyscale | ✓ | — | — | — | — | — | — | — |
| Azure | ✓ | ✓ | ✓ | ✓ | ✓ | ✓ | ✓ | ✓ |
| Deepinfra | ✓ | ✓ | ✓ | ✓ | ✓ | ✓ | ✓ | ✓ |
| Fireworks | ✓ | ✓ | ✓ | ✓ | ✓ | ✗ | ✓ | ✗ |
| Groq | ✓ | ✓ | ✓ | ✗ | ✓ | ✗ | ✓ | ✓ |
| Perplexity | ✗ | ✗ | ✗ | ✗ | ✗ | ✗ | ✗ | ✗ |
| Replicate | ✓ | — | ✗ | — | ✓ | — | ✓ | — |
| Together | ✓ | ✓ | ✓ | ✓ | ✓ | ✓ | ✓ | ✓ |

Table 1: Audit results for 31 API endpoints across nine companies and four language models. ✗ denotes an endpoint failure, i.e., the average rejection rate over ten samples is $\geq 50\%$.

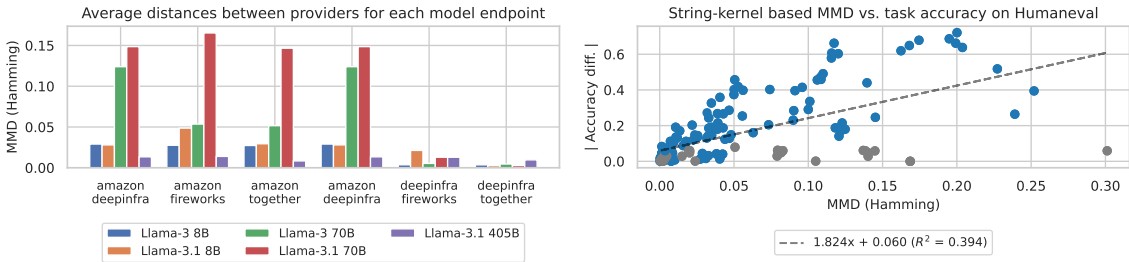

Figure 4: *(Left)* Average MMD (Hamming) between providers for each model. Amazon Bedrock's Llama-3 and -3.1 70B models are the most different from the other providers. *(Right)* Absolute difference in HumanEval average accuracy vs. the MMD (Hamming). There is a moderate positive correlation between MMD and task accuracy. Gray points indicate pairs where both distributions have accuracy $< 10\%$. There are multiple ways to be wrong for a task, and the MMD captures these differences.

implementations are further from the reference weights than if the provider had substituted in an entirely different language model. For example, the deviation between Perplexity's Llama-3 8B and the fp32 null on the Wikipedia testing task is 0.03. This is comparable to the deviation between Llama-3 8B and GPT-3.5-Turbo (0.03; Figure 3 right), Phi-3 mini (0.04), or OLMo 7B (0.04).

**Correlating MMD with task accuracy.** We now ask how well the Hamming MMD correlates with task accuracy when available. Automated verifiers exist for one of our prompt distributions, HumanEval. We find a moderate positive correlation between the absolute average accuracy difference and the Hamming MMD (Figure 4 right, $R^2 = 0.392$). In several cases, the MMD is high but the accuracy difference is low. These are often when both task accuracies are low: the gray points in the figure highlight pairs where both distributions have accuracy $< 10\%$. In these cases, the MMD captures that there are multiple ways to be wrong for a task. For example, the bottom-right-most point in the figure contrasts Llama-3.1 8B from Perplexity vs. the nf4 version; Perplexity answers with incorrect code snippets, while nf4 outputs nonsense completions (e.g., "simulation!!!!!!!!!![...]"). Samples from this comparison can be found in Appendix D.3.

**Which models do providers differ on most?** Finally, we ask whether providers differ on some more models more than others. Figure 4 (left) estimates the MMD between providers for each model, and the large Llama-3.1 405B model. We find that deviations are larger on the earlier Llama models than the recent Llama-3.1 405B model.

---

**Box 2: Samples from fp32 vs. Perplexity for Llama-3 8B (Instruct)**

**Prompt:** Continue the paragraph. Do not output anything except the continuation to the paragraph. Start the continuation immediately.
"Weaver, Tom. I Talked with a Zombie: Interviews with 23 Veterans of Horror and Sci-Fi Films and Tele..."

| *fp32* | *Perplexity* |
|---|---|
| "...vision, Some of Whom Are Still Alive. New York: Pocket Books, 1981. According to Tom Weaver, Vincent Price was known for being a bit of a practical joker, and he would often play harmless pranks on the crew | "...visions. New York: St. Martin's Press, 1988. This collection of interviews with horror and science fiction professionals is a fascinating look at the creative process and the people behind some of the most iconic films and TV shows of the |
| "...vision, voodoo, and the Apocalypse. New York: St. Martin's Press, 1990. The book features interviews with notable horror and science fiction personalities, including Greg Cannom, makeup artist for The Lost Boys and RoboCop | "...visions. New York: St. Martin's Press, 1988. In this collection of interviews, Weaver delves into the experiences of some of the most iconic figures in the horror and science fiction genres, including George Romero, John Carpenter, |
| "...vision. Written by Tom Weaver, a renowned authority on classical horror cinema, this book brings together intimate and revealing conversations with some of the most legendary figures in the genre, including Vincent Price, Boris Karloff, and Bela Lugosi. Interviews | "...vision. New York: St. Martin's Press, 1988. 224 pp. This collection of interviews with horror and science fiction professionals is a fascinating look at the people behind the genre's most iconic films. Weaver's conversational style |

## 6 RELATED WORK

**Auditing ML APIs.** Prior work either only applies to the classifier setting (Chen et al., 2021a; 2022) or audit by monitoring performance on low-temperature multiple-choice or short-answer benchmarks (ArtificialAnalysis, 2024; Chen et al., 2023; Eyuboglu et al., 2024). Concurrent work studies a general language model setting, but does not provide guarantees of audit correctness (Sam et al., 2025). In contrast, we formalize auditing as a hypothesis testing problem, allowing us to provide FPR guarantees, and we study the general setting of sampling completions from language models, including for tasks without automated verifiers. In cryptography, a body of work has aimed to verify ML API predictions through proof protocols (Ghodsi et al., 2017; Feng et al., 2021; Liu et al., 2021; Kang et al., 2022; Weng et al., 2023; Lee et al., 2024). These methods require APIs to cooperate and opt-in to providing proofs of valid inference alongside each prediction; public verifiers then check these proofs for correctness with perfect accuracy. Unfortunately, these methods scale too poorly to apply to large language models: in Kang et al. (2022), generating a proof takes 41 minutes *per prediction* for a 68M parameter MobileNet (Sandler et al., 2018).

**Two-sample testing.** Testing whether two samples come from the same distribution is an established problem in statistics. A line of work focuses on testing multinomial distributions (Batu et al., 2013; Chan et al., 2014; Canonne, 2020), including when samples have unequal sizes (Bhattacharya & Valiant, 2015; Diakonikolas & Kane, 2016; Balakrishnan & Wasserman, 2018). MMD-based tests are a general approach that do not assume a specific distributional form (Gretton et al., 2012). These tests can be applied to structured data when paired with appropriate kernels (Lodhi et al., 2002; Borgwardt et al., 2006).

## 7 CONCLUSION

As the public grows more dependent on black-box APIs to interact with language models, tools for auditing these APIs are increasingly important. In this work, we unified several API auditing tasks under the formalization of Model Equality Testing, a two-sample distribution testing problem, and we extensively validated candidate tests for this problem. Future work could explore stronger tests than those we have presented, or explore how to adapt these tests to other modalities, such as image generation models. To help facilitate this research, we open-source the dataset of 1 million LLM completions used for this work at https://github.com/i-gao/model-equality-testing. This dataset contains completions from five models across quantized, watermarked, and API alternatives; also see Appendix B.1.

## ACKNOWLEDGEMENTS

The authors would like to thank Mert Yuksekgonul, Steven Cao, Chenchen Gu, Chenglei Si, Nicole Meister, Yifan Mai, Simon Guo, Teddi Worledge, Yuhui Zhang, and other members of the Guestrin and P-Lambda labs for feedback on this paper.

## CONFLICTS OF INTEREST

PL is a co-founder of Together AI; this work was done in his Stanford capacity. The topic, research, and results of this work were not shared with Together, or any other API provider evaluated, until the public release of the paper.

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
