## A  ADDITIONAL NOTES ON TESTS

### A.1  WHICH MMD KERNELS LEAD TO VALID METRICS?

Recall that a metric on probability distributions satisfies (1) symmetry, (2) the triangle inequality, and (3) $d(P, Q) = 0$ if and only if $P = Q$. On the other hand, a pseudo-metric satisfies the first two properties and has $d(P, P) = 0$. Regardless of the kernel, the MMD as defined in Equation 3 is clearly symmetric. It also satisfies triangle inequality, since $\text{MMD}(P_1, P_3) = \text{MMD}(P_1, P_2) + \text{MMD}(P_2, P_3)$ for any $P_1, P_2, P_3$. The question left is whether our kernels make the MMD injective as in Condition 3.

- **One-hot kernel**. (Gretton et al., 2012) prove that universal kernels (Steinwart, 2001) result in an injective MMD. (Borgwardt et al., 2006) show that a kernel defined on a finite domain $\mathcal{X}$ is universal if $k$ satisfies *strict positive definiteness*: i.e., $k$ induces a nonsingular Gram matrix for any finite set of points $X \subseteq \mathcal{X}$. This is true if $\phi(x^{(1)}), \cdots, \phi(x^{(n)})$ are linearly independent for any set of distinct points $x^{(1)}, \cdots, x^{(n)} \in \mathcal{X}$. For the one-hot kernel, the associated feature map $\phi_{\text{one-hot}}$ is of length $|V|^L$, where the $i$th entry is an indicator for whether $x$ is equal to the $i$th string. Since all $\phi(x)$ is one-hot and for distinct sets $X$, no two $\phi(x)$ are both 1 at the same index, the $\phi(x)$ are linearly independent. Therefore this kernel is universal, and $\text{MMD}_{\text{one-hot}}$ is a valid metric.

- **All-substrings kernel**. Borgwardt et al. (2006) (Theorem 2.7) prove that this kernel is universal, and thus the MMD is a metric.

- **Hamming kernel**. We will show that the MMD is not injective by showing that the mean embedding $\mathbb{E}_P[\phi(x)]$ is not injective, i.e., there exist $P \neq Q$ with $\mathbb{E}_P[\phi(x)] = \mathbb{E}_Q[\phi(x)]$. For the Hamming kernel, the associated feature map is of length $|\mathcal{V}| \times L$:

$$
\phi(x) = \begin{bmatrix} \mathbf{1}\{x_1 = v_1\} \\ \mathbf{1}\{x_1 = v_2\} \\ \cdots \\ \mathbf{1}\{x_1 = v_{|\mathcal{V}|}\} \\ \cdots \\ \mathbf{1}\{x_L = v_1\} \\ \mathbf{1}\{x_L = v_2\} \\ \cdots \\ \mathbf{1}\{x_L = v_{|\mathcal{V}|}\} \end{bmatrix} \implies \mathbb{E}[\phi(x)] = \begin{bmatrix} P(x_1 = v_1) \\ P(x_1 = v_2) \\ \cdots \\ P(x_1 = v_{|\mathcal{V}|}) \\ \cdots \\ P(x_L = v_1) \\ P(x_L = v_2) \\ \cdots \\ P(x_L = v_{|\mathcal{V}|}) \end{bmatrix}
$$

  i.e., the mean embedding stacks all marginal distributions of $P$. But this shows the mean embedding is not injective: we know that multiple joint distributions $P \neq Q$ can map to the same marginal distributions. Thus the Hamming MMD is not injective, and it is only a pseudo-metric.

### A.2  SIMULATING P-VALUES

P-values for MMD tests may be simulated in two ways:

1. **Simulating the test statistic under the null (Algorithm 1).** This is done by repeatedly sampling $\mathcal{D}_Q$ and $\mathcal{D}_P$ from $P$ and caching $\widehat{\text{MMD}}(\mathcal{D}_Q, \mathcal{D}_P)$. The p-value is then the proportion of times the test statistic is greater than or equal to the observed test statistic. We conduct tests using this method in the main text, reusing the same cached empirical distribution of the test statistic under the null for all alternatives at that sample size. Note that this method requires significant sampling access to $P$.

2. **Permutation procedure (Lehmann et al. (1986); Algorithm 2).** Given samples $\mathcal{D}_Q$ and $\mathcal{D}_P$, the permutation procedure randomly shuffles the labels of the samples and computes the test statistic on the permuted samples. This process is repeated many times to estimate the null distribution of the test statistic. The p-value is then the proportion of times the permuted test statistic is greater

than or equal to the observed test statistic. This method does not require additional sampling access to $P$ but may have lower power. We conduct experiments using this method in Appendix C.7.

---

**Algorithm 1** Simulating the test statistic under the null

---

1: **Input:** Number of simulations $B$, null distribution $P$, test samples $\mathcal{D}_Q$ and $\mathcal{D}_P$
2: Initialize a list $\mathcal{T} = []$ to store simulated test statistics
3: **for** $i = 1, 2, \ldots, B$ **do**
4:     Sample $\mathcal{D}_Q^{(i)} \sim P$ and $\mathcal{D}_P^{(i)} \sim P$
5:     Compute $\widehat{\text{MMD}}(\mathcal{D}_Q^{(i)}, \mathcal{D}_P^{(i)})$ and append to $\mathcal{T}$
6: **end for**
7: Compute p-value as the proportion of $\mathcal{T}$ greater than or equal to $\widehat{\text{MMD}}(\mathcal{D}_Q, \mathcal{D}_P)$
8: **Return** p-value

---

---

**Algorithm 2** Permutation testing

---

1: **Input:** Number of permutations $B$, test samples $\mathcal{D}_Q$ and $\mathcal{D}_P$
2: Concatenate $\mathcal{D}_Q$ and $\mathcal{D}_P$ into a single dataset $\mathcal{D} = [\mathcal{D}_Q, \mathcal{D}_P]$
3: Initialize a list $\mathcal{T} = []$ to store permuted test statistics
4: **for** $i = 1, 2, \ldots, B$ **do**
5:     Randomly shuffle $\mathcal{D}$ and split into two sets: $\mathcal{D}_Q^{(i)}$ and $\mathcal{D}_P^{(i)}$
6:     Compute $\widehat{\text{MMD}}(\mathcal{D}_Q^{(i)}, \mathcal{D}_P^{(i)})$ and append to $\mathcal{T}$
7: **end for**
8: Compute p-value as the proportion of $\mathcal{T}$ greater than or equal to $\widehat{\text{MMD}}(\mathcal{D}_Q, \mathcal{D}_P)$
9: **Return** p-value

---

## B  EXPERIMENT DETAILS

### B.1  SAMPLING AND DATASET DETAILS

All experiments were conducted by sampling with replacement from a pre-collected dataset of language model completions, which we release alongside this paper at https://github.com/i-gao/model-equality-testing. The dataset consists of completions from five models: mistralai/Mistral-7B-Instruct-v0.3, meta-llama/Meta-Llama-3-8B-Instruct, meta-llama/Meta-Llama-3.1-8B-Instruct, meta-llama/Meta-Llama-3-70B-Instruct, and meta-llama/Meta-Llama-3.1-70B-Instruct. We collected multiple completions per prompt for prompts from Wikipedia (Cohere, 2023), HumanEval (Chen et al., 2021b), and UltraChat (Ding et al., 2023). In total, our dataset contains 440 prompts: 80 from Wikipedia in each of English, Spanish, French, German, and Russian, as well as 20 from HumanEval and 20 from UltraChat. We repeated the collection process for each model at each precision (fp32, fp16, fp16, int8, and nf4) and when watermarked using the method in Kirchenbauer et al. (2023), as well as from each API audited in the paper. In total, this dataset size is 1.1M completions.

---

**Box 3: Sample prompt for HumanEval**

Complete the code. Do not output anything except the completion. Start the continuation immediately.

```
from typing import List

def has_close_elements(numbers: List[float], threshold: float) /> bool:
    """ Check if in given list of numbers, are any two numbers closer
    to each other than given threshold.
    >>> has_close_elements([1.0, 2.0, 3.0], 0.5)
    False
    >>> has_close_elements([1.0, 2.8, 3.0, 4.0, 5.0, 2.0], 0.3)
    True
    """
```

---

**Box 4: Sample prompt for UltraChat**

Explain how the invention and widespread use of digital music formats such as MP3s and streaming services like Spotify have impacted the way music is distributed, consumed, and monetized in the music industry, and how this has affected the relationship between artists and their fans. Provide specific examples of how technological advancements have changed the production and consumption of music, including changes to the format and length of songs, the role of record labels, and the use of social media to promote artists and their work. Additionally, discuss possible future developments in music technology and their potential impact on the industry and consumer habits.

---

For the goodness-of-fit experiments in Appendix C.5, we collected completions for a disjoint set of 20 additional prompts from each of language of Wikipedia for Mistral 7B and Llama-3 8B. This set also includes the log probabilities of each sample under the full precision model. All other experiments outside of this goodness-of-fit experiment were conducted on the previous, larger dataset.

**Details about local sampling (fp32, fp16, int8, nf4 4, and watermark).** We collected 15 000 completions per prompt for the fp32 precision, and 5000 completions per prompt for the other precisions and watermarked alternatives. Local sampling was performed on a mix of RTX 6000, RTX 3090, Quadro RTX 8000, and A100 GPUs using a mixture of the Transformers (Wolf et al., 2020) and VLLM (Kwon et al., 2023) libraries. The watermarking, nf4, and int8 implementations are from the Transformers library. We use the default watermarking parameters of 2.5 bias and context width 1. All samples were collected with vanilla decoding

parameters of temperature 1 without top-k or top-p sampling, and parameter max_new_tokens set to $L = 50$ for Wikipedia and $L = 250$ for HumanEval and UltraChat.

We used the default chat templates from the Transformers library for models. It was important for us to match the chat templates during local sampling to those we believe the APIs to use, as the chat templates can affect the completions generated. We confirmed that the number of tokens in our local rendering of the prompts matched the returned number of prompt tokens from API calls. The exception was for the Llama-3.1 models, where the default Transformers chat template includes the current date. We found that this template did not match the number of prompt tokens returned by APIs; however, the Llama-3 chat template did.[9] As a result, we used the Llama-3 chat template for Llama-3.1 models in our local sampling.

**Details about API sampling.** API samples are collected by repeatedly querying endpoints for one completion at a time. We aimed to collect 250 completions per prompt for each API within a 24 hour window, but due to rate limits and request failures, some prompts had samples collected over multiple days. The dates we query each endpoint are listed in Tables 2 – 5. We query serverless endpoints offered by providers and use the same decoding parameters as for local sampling. When available, we called the providers using their Python packages; otherwise, we made raw HTTP requests.

Below are provider-specific details:

- *Anyscale.* Because Anyscale deprecated its endpoints during our data collection in August 2024, we were only able to collect samples from their Llama-3 8B endpoint for Wikipedia.

- *Together.* We collected Llama-3 8B and 70B samples from Together.ai before they introduced separate reference, turbo, and lite endpoints. We collected Llama-3.1 8B, 70B, and 405B from the turbo endpoints, which was the only option available at the time of collection.

| Model | Dataset | Provider | Dates queried |
|---|---|---|---|
| 3 8B | Wikipedia | Anyscale | 7/4 |
| | | Amazon | 7/8, 8/1 |
| | | Azure | 8/19-20, 8/24 |
| | | Deepinfra | 7/4, 8/1 |
| | | Fireworks | 7/4, 7/19 |
| | | Groq | 7/4, 8/1-4, 8/8 |
| | | Perplexity | 7/4 |
| | | Replicate | 7/4, 7/19 |
| | | Together | 7/4 |
| | HumanEval | Amazon | 7/29 |
| | | Azure | 8/24 |
| | | Deepinfra | 8/1 |
| | | Fireworks | 8/1, 8/12 |
| | | Groq | 8/4, 8/24 |
| | | Perplexity | 8/12 |
| | | Replicate | 8/1 |
| | | Together | 8/1, 8/12 |
| | UltraChat | Amazon | 7/29 |
| | | Azure | 8/24 |
| | | Deepinfra | 8/1 |
| | | Fireworks | 8/1, 8/12 |
| | | Groq | 8/4, 8/24 |
| | | Perplexity | 8/12 |
| | | Replicate | 8/1-2 |
| | | Together | 8/1, 8/12 |

Table 2: Dates we queried Llama-3 8B inference endpoints.

---

[9]In October 2024, this behavior has changed for Together AI, which now uses the Llama-3.1 chat template that includes the current date. At the time we collected samples with the Llama-3 template, this was not the case.

| Model | Dataset | Provider | Dates queried |
|---|---|---|---|
| 3.1 8B | Wikipedia | Amazon | 8/1 |
| | | Azure | 8/19-21, 8/23-24 |
| | | Deepinfra | 8/1-2 |
| | | Fireworks | 7/26-27, 8/6 |
| | | Groq | 8/1-4, 8/8-11, 8/24-26 |
| | | Perplexity | 8/1-2 |
| | | Together | 7/26-27, 8/6-8 |
| | HumanEval | Amazon | 8/1 |
| | | Azure | 8/24 |
| | | Deepinfra | 7/31 |
| | | Fireworks | 7/30 |
| | | Groq | 7/31, 8/24, 8/27 |
| | | Perplexity | 7/30 |
| | | Together | 7/30 |
| | UltraChat | Amazon | 7/26-27, 8/1, 8/6 |
| | | Azure | 8/24 |
| | | Deepinfra | 7/31 |
| | | Fireworks | 7/26, 7/30, 8/6 |
| | | Groq | 7/31-8/1, 8/27 |
| | | Perplexity | 7/30 |
| | | Together | 7/26-27, 7/30, 8/6 |

Table 3: Dates we queried Llama-3.1 8B inference endpoints.

| Model | Dataset | Provider | Dates queried |
|---|---|---|---|
| 3 70B | Wikipedia | Amazon | 7/8, 8/1 |
| | | Azure | 8/19-21, 8/24 |
| | | Deepinfra | 7/4, 8/5-6 |
| | | Fireworks | 7/4, 7/31-8/1 |
| | | Groq | 7/4, 7/31, 8/2-13 |
| | | Perplexity | 7/4, 7/8, 8/5-6 |
| | | Replicate | 7/4, 7/19, 7/31-8/1 |
| | | Together | 7/4, 7/31-8/1 |
| | HumanEval | Amazon | 7/29 |
| | | Azure | 8/24 |
| | | Deepinfra | 8/1 |
| | | Fireworks | 8/6 |
| | | Groq | 8/1 |
| | | Perplexity | 8/6 |
| | | Replicate | 8/6 |
| | | Together | 8/6, 8/24 |
| | UltraChat | Amazon | 8/24 |
| | | Azure | 8/25-26 |
| | | Deepinfra | 8/1-2 |
| | | Fireworks | 8/6-7, 8/24 |
| | | Groq | 8/1-2, 8/4, 8/24 |
| | | Perplexity | 8/6, 8/24 |
| | | Replicate | 8/6 |
| | | Together | 8/6, 8/24 |

Table 4: Dates we queried Llama-3 70B inference endpoints.

## B.2 MONTE CARLO SIMULATIONS

To construct the ten Wikipedia prompt distributions in §4, we randomly sampled 25 prompts per distribution from the Wikipedia prompts in our dataset. The HumanEval and UltraChat prompt distributions were constructed by using all available prompts from those sources.

| Model | Dataset | Provider | Dates queried |
|---|---|---|---|
| 3.1 70B | Wikipedia | Amazon | 8/2 |
| | | Azure | 8/24 |
| | | Deepinfra | 8/2 |
| | | Fireworks | 7/27-28, 8/2, 8/5-6 |
| | | Groq | 8/2-5, 8/8-11, 8/21-24 |
| | | Perplexity | 8/2, 8/6 |
| | | Together | 7/27-28, 8/6 |
| | HumanEval | Amazon | 8/24 |
| | | Azure | 8/24, 8/26 |
| | | Deepinfra | 7/31 |
| | | Fireworks | 7/30 |
| | | Groq | 7/31, 8/1-2, 8/24 |
| | | Perplexity | 7/30-31 |
| | | Together | 7/30, 8/6 |
| | UltraChat | Amazon | 8/24-25 |
| | | Azure | 8/25-26 |
| | | Deepinfra | 7/31 |
| | | Fireworks | 7/30 |
| | | Groq | 7/31-8/1, 8/24 |
| | | Perplexity | 7/30-31 |
| | | Together | 7/30, 8/6 |

Table 5: Dates we queried Llama-3.1 70B inference endpoints.

| Model | Dataset | Provider | Dates queried |
|---|---|---|---|
| 3.1 405B | Wikipedia | Amazon | 8/16-17, 8/23-24 |
| | | Deepinfra | 8/16, 8/23-24 |
| | | Fireworks | 8/16, 8/23-24 |
| | | Together | 8/16, 8/20, 8/23-24 |
| | HumanEval | Amazon | 8/24-25 |
| | | Deepinfra | 8/24 |
| | | Fireworks | 8/24 |
| | | Together | 8/24 |
| | UltraChat | Amazon | 8/24-25 |
| | | Deepinfra | 8/24-25 |
| | | Fireworks | 8/24 |
| | | Together | 8/24 |

Table 6: Dates we queried Llama-3.1 405B inference endpoints.

In most experiments, we estimated power as the average rejection rate over 100 simulations, where we sample a fresh $\mathcal{D}_P$ and $\mathcal{D}_Q$ each time. We simulated p-values by sampling 1000 datasets $\mathcal{D}_P$ and $\mathcal{D}_Q$ from $P$ and computing the test statistic on each pair, and then we reused this empirical distribution when testing against all alternatives for the same $\pi P$. The exception is for the MMD all-substrings test statistic: because this was exceptionally slow to compute, we simulated p-values using 100 samples instead of 1000, and we estimated power from 20 simulations instead of 100.

| Model | Dates queried |
|---|---|
| gpt-4o-mini | 8/21, 8/23-24 |
| gpt-4o | 8/29 |
| gpt-3.5-turbo-0125 | 8/29 |
| gpt-3.5-turbo-1106 | 8/29 |
| gpt-4-0125-preview | 8/29 |

Table 7: Dates we queried OpenAI endpoints (Wikipedia task).

## C    ADDITIONAL RESULTS

### C.1    ADDITIONAL RESULTS FROM §4.1

In §4.1, we evaluated two-sample tests on their ability to distinguish samples (in token space, $L \leq 50$) from full-precision models from those of quantized models and watermarked models. Concretely, we evaluated tests' power on pairs of distributions $(P, Q)$, where $P$ represents samples from the fp32 model and $Q$ represents samples from one of {fp16, int8, nf4, or watermarked} versions of the same model. To ensure generalizable results, we experimented with five language models (Mistral 7B Instruct, Llama-3 8B and 70B Instruct, and Llama-3.1 8B and 70B Instruct) and ten Wikipedia-based prompt distributions (Box 1).

Figure 2 in the main text plotted sample complexity and length results, *averaging across language models and prompt distributions*. In this appendix, we show plots for the same experiments, but *stratified by language model and prompt distribution*.

- Figure 5 shows the sample complexities for different two-sample tests, stratified by the alternative distribution $Q$, but averaged over the five language models and ten prompt distributions.

- Figure 6 shows the sample complexities for different two-sample tests, stratified by the alternative distribution $Q$ and model, but averaged over ten prompt distributions.

- Figure 7 shows the simulated powers for different completion lengths $L$, stratified by the alternative distribution $Q$.

Additionally, Table 8 provides the power of the Hamming MMD test to distinguish $(P, Q)$ for other language models in $L = 50$ token space. This table adds Phi-3 Mini (Instruct), OLMo 7B (Instruct), and Gemma-2 9B (Instruct). Table 9 evaluates a Levenshtein MMD test, which uses the Levenshtein distance as the kernel function. The Levenshtein kernel is defined as $k(x, y) = \exp(-\text{Levenshtein}(x, y))$. It is less powerful than the Hamming kernel, but still outperforms traditional two-sample tests.

**Discussion: the Hamming MMD is the most powerful test.**    Across Figure 2 (left), Figure 5, and Figure 6, we observe that the Hamming MMD test is significantly more sample-efficient for all alternatives except fp16 quantization at completion length $L = 50$ tokens. We also observe that the Hamming MMD is more robust to changes in completion length than other tests (Figure 2 middle, Figure 7).

**Discussion: tests' powers depend on the particular $(P, Q)$ pair.**    In Figure 6 and Table 8, we observe that tests' powers are heterogeneous across different $(P, Q)$ pairs. The choice of model affects both the reference distribution $P$ and the effect of interventions like quantization, i.e., the alternative $Q$. For example, 4-bit quantization of the Llama and OLMo models is consistently noticeable. On the other hand, 8-bit quantization and watermarking have more inconsistent effects. The impact of quantization is heterogeneous even within the same model family: for example, the Llama-3 8B model is more affected by int8 quantization than the Llama-3.1 8B model. On average however, as discussed above, our results above still support our claim that the Hamming MMD is more powerful than other kernel choices.

### C.2    ADDITIONAL RESULTS FROM §4.3

In §4.3, we moved from evaluating tests in an easier token space to a higher-dimensional character space. Specifically, in §4.1 and §4.2, we evaluated tests in $L = 50$ token space, where $|\mathcal{V}| = 128\,256$. In §4.3 and §5, we evaluated tests in $L = 1000$ or $L = 5000$ character space, where $|\mathcal{V}| = 155\,063$. In general, since the set of all prompt-completion pairs has size $O(m|\mathcal{V}|^L)$, we expect testing to be harder in the latter, *significantly* higher-dimensional space. One can come to the same conclusion by extrapolating from Figure 2 (middle) in the main text; this figure also shows that the Hamming MMD test is more robust to this increase in dimensionality than other tests.

**Discussion: power decreases in higher-dimensional space.**    Table 10 shows the power of the Hamming MMD test to distinguish between pairs of distributions $(P, Q)$ in $L = 1000$ character space. As expected,

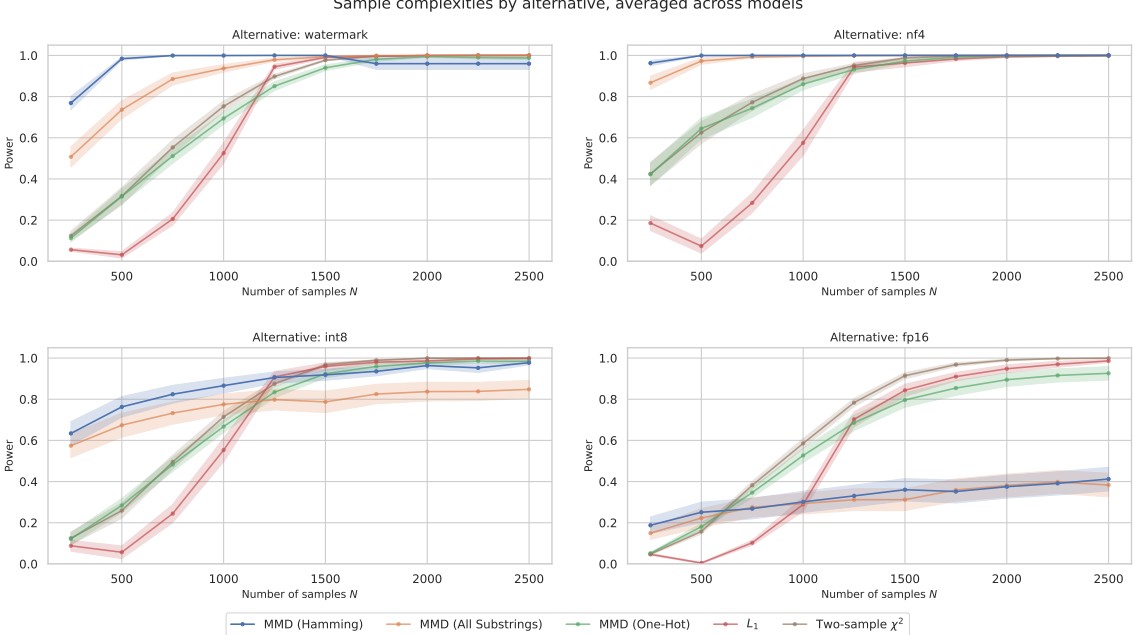

Figure 5: Sample complexities for different two-sample tests. Each subplot represents power for an alternative distribution $Q$ averaged over five language models and ten prompt distributions $\pi$. Tests are conducted with sample sizes ranging from $N = 10m$ to $N = 100m$, where $m = 25$ is the number of prompts in the prompt distribution. The completion length is fixed to $L = 50$ tokens. The Hamming MMD test is significantly more sample-efficient for all alternatives except fp16 quantization: for this alternative, while other two-sample tests can attain perfect power with enough samples, the kernel test increases power slowly. This suggests that fp16 and fp32 differ in ways that are difficult to capture with the Hamming kernel.

moving to this higher-dimensional space decreases the power of the test; the powers in the last three rows are lower than those in Table 8 for the same models in $L = 50$ token space. These power losses can be compensated for by increasing the sample size: Table 11 repeats the experiments in Table 10 with $N = 50m$ instead of $N = 10m$. If we were to use this sample size in §5, each audit would still cost $< \$5$.

Note that even in the reduced power setting, model swaps are easy to detect (Table 10, top rows).

## C.3 DETECTING INCORRECT DECODING PARAMETERS

Our problem formulation in §2 compares two distributions in general: this includes cases where the API $Q$ samples from the same model as the reference distribution $P$, but with different decoding parameters (e.g., temperature $\tau$). Table 12 shows that detecting differences in the temperature parameter is quite easy; this parameter significantly affects the distribution of completions.

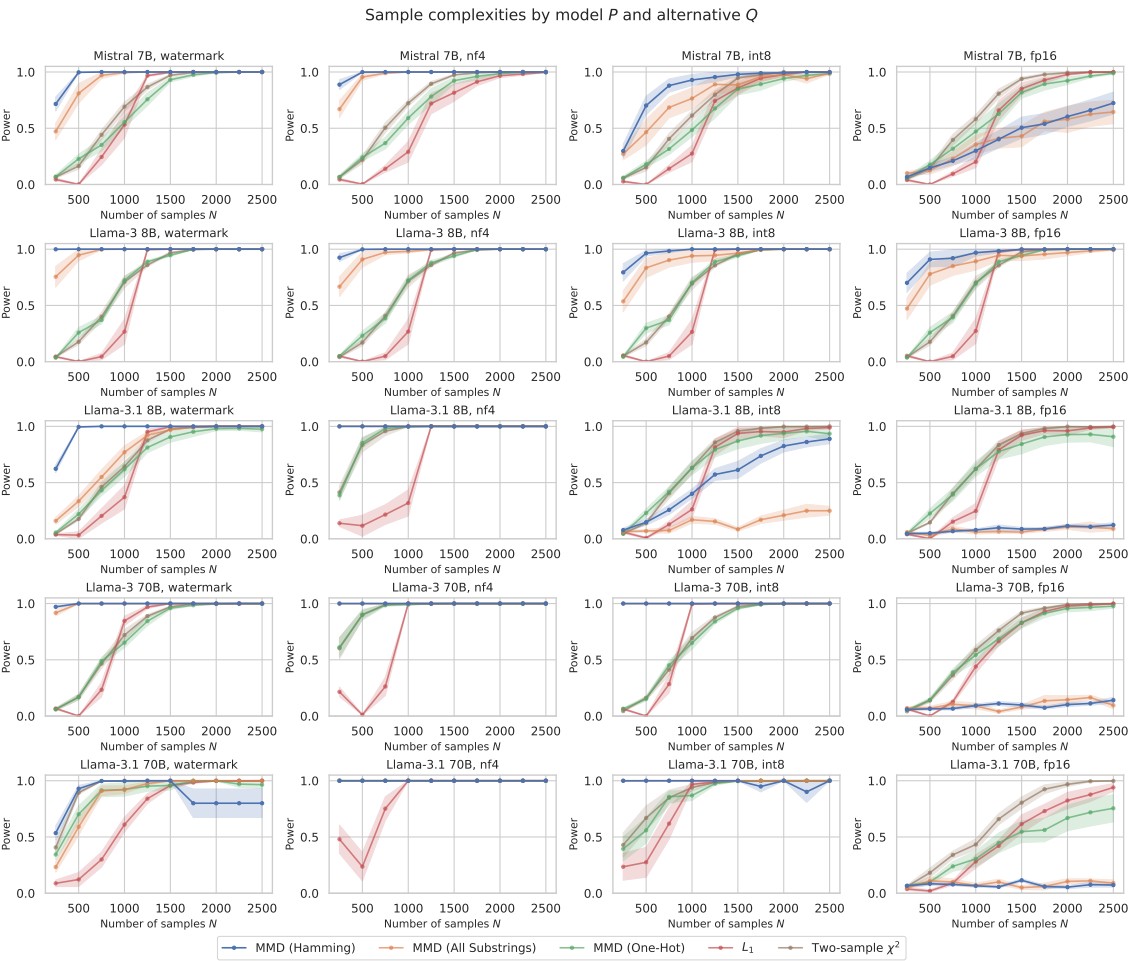

Figure 6: Sample complexities for different two-sample tests. Each subplot represents power for a particular alternative distribution $Q$ and model, but averaged over ten prompt distributions $\pi$. Tests are conducted with sample sizes ranging from $N = 10m$ to $N = 100m$, where $m = 25$ is the number of prompts in the prompt distribution. The completion length is fixed to $L = 50$ tokens. Some model and alternative combinations are more difficult to detect than others.

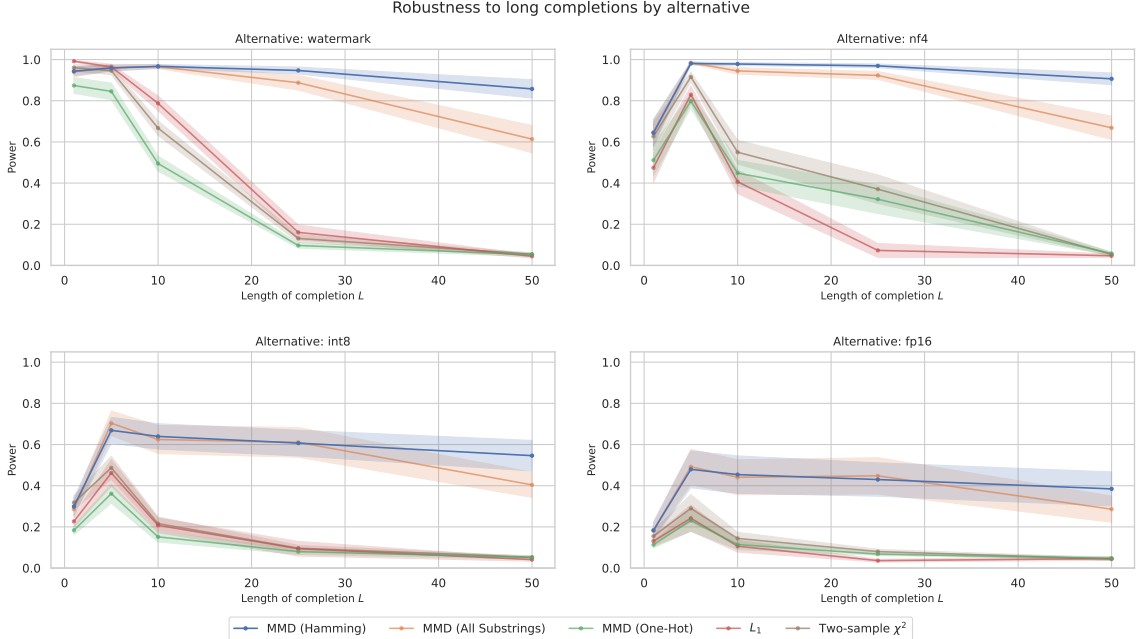

Figure 7: Simulated powers for different completion lengths $L \in \{1, \cdots, 50\}$, stratified by the alternative distribution $Q$. The sample size is fixed to $N = 10m$, where $m = 25$ is the number of prompts in the prompt distribution. Across all alternatives, kernel tests suffer less drop in power as the length of completions increases. Traditional two-sample tests observe a slight power increase from $L = 1$ tokens to $L = 5$, tokens, but power dramatically degrades afterwards.

| Model | watermark | nf4 | int8 | FPR (fp32) |
|---|---|---|---|---|
| Llama-3 8B | 1.00 (0.00) | 0.93 (0.04) | 0.79 (0.08) | 0.05 (0.01) |
| Llama-3.1 8B | 0.62 (0.04) | 1.00 (0.00) | 0.08 (0.02) | 0.05 (0.01) |
| Llama-3 70B | 0.97 (0.01) | 1.00 (0.00) | 1.00 (0.00) | 0.07 (0.01) |
| Llama-3.1 70B | 0.54 (0.06) | 1.00 (0.00) | 1.00 (0.00) | 0.06 (0.01) |
| Mistral 7B | 0.72 (0.06) | 0.89 (0.04) | 0.30 (0.06) | 0.04 (0.01) |
| OLMo 7B | 0.47 (0.08) | 0.99 (0.01) | 0.36 (0.08) | 0.06 (0.01) |
| Gemma-2 9B | 0.43 (0.02) | 0.12 (0.01) | 0.06 (0.01) | 0.06 (0.01) |
| Phi-3 Mini | 0.75 (0.04) | 0.63 (0.05) | 0.24 (0.01) | 0.06 (0.01) |

Table 8: Power of the Hamming MMD test to distinguish the fp32 model from alternatives in $L = 50$ token space at $N = 10m$ samples, where $m = 25$ is the number of prompts in the prompt distribution. Averages and standard errors are reported over ten prompt distributions. The column represents the alternative distribution $Q$. The FPR (fp32) column represents the false positive rate when comparing the full-precision model to itself.

| Model | watermark | nf4 | int8 | FPR (fp32) |
|---|---|---|---|---|
| Llama-3 8B | 0.14 (0.01) | 0.15 (0.01) | 0.16 (0.01) | 0.07 (0.01) |
| Llama-3 70B | 0.13 (0.02) | 1.00 (0.00) | 0.12 (0.01) | 0.04 (0.01) |
| Mistral 7B | 0.11 (0.01) | 0.19 (0.01) | 0.12 (0.01) | 0.05 (0.01) |

Table 9: Power of the Levenshtein MMD test to distinguish the fp32 model from alternatives in $L = 50$ token space at $N = 10m$ samples, where $m = 25$ is the number of prompts in the prompt distribution. Averages and standard errors are reported over ten prompt distributions. The column represents the alternative distribution $Q$. The FPR (fp32) column represents the false positive rate when comparing the full-precision model to itself.

| model alternative | Mistral 7B | Llama-3 8B | Llama-3.1 8B | Llama-3 70B | Llama-3.1 70B |
|---|---|---|---|---|---|
| Mistral 7B | *0.06 (0.01)* | 1.00 (0.00) | 1.00 (0.00) | 1.00 (0.00) | 1.00 (0.00) |
| Llama-3 8B | 1.00 (0.00) | *0.05 (0.01)* | 0.76 (0.00) | 0.98 (0.00) | 0.95 (0.00) |
| Llama-3.1 8B | 1.00 (0.00) | 0.83 (0.05) | *0.07 (0.01)* | 1.00 (0.00) | 0.53 (0.00) |
| Llama-3 70B | 1.00 (0.00) | 0.99 (0.00) | 1.00 (0.00) | *0.07 (0.01)* | 0.89 (0.00) |
| Llama-3.1 70B | 1.00 (0.00) | 0.98 (0.00) | 0.75 (0.07) | 0.99 (0.00) | *0.06 (0.01)* |
| Phi-3 mini | 1.00 (0.00) | 1.00 (0.00) | 1.00 (0.00) | 1.00 (0.00) | 1.00 (0.00) |
| Gemma-2 9B | 1.00 (0.00) | 1.00 (0.00) | 1.00 (0.00) | 1.00 (0.00) | 1.00 (0.00) |
| OLMo 7B | 1.00 (0.00) | 1.00 (0.00) | 1.00 (0.00) | 1.00 (0.00) | 1.00 (0.00) |
| gpt-4o-mini | 1.00 (0.00) | 1.00 (0.00) | 1.00 (0.00) | 1.00 (0.00) | 1.00 (0.00) |
| gpt-4o | 1.00 (0.00) | 1.00 (0.00) | 1.00 (0.00) | 1.00 (0.00) | 1.00 (0.00) |
| gpt-3.5-turbo-0125 | 1.00 (0.00) | 1.00 (0.00) | 1.00 (0.00) | 1.00 (0.00) | 1.00 (0.00) |
| gpt-3.5-turbo-1106 | 1.00 (0.00) | 1.00 (0.00) | 1.00 (0.00) | 1.00 (0.00) | 1.00 (0.00) |
| gpt-4-0125-preview | 1.00 (0.00) | 1.00 (0.00) | 1.00 (0.00) | 1.00 (0.00) | 1.00 (0.00) |
| watermark | 0.23 (0.02) | 0.62 (0.05) | 0.32 (0.04) | 0.57 (0.06) | 0.26 (0.03) |
| int8 | 0.15 (0.02) | 0.30 (0.06) | 0.07 (0.01) | 1.00 (0.00) | 0.99 (0.01) |
| nf4 | 0.44 (0.05) | 0.38 (0.05) | 1.00 (0.00) | 1.00 (0.00) | 1.00 (0.00) |

Table 10: Power of the Hamming MMD test to distinguish between pairs of models (and other alternatives) in $L = 1000$ character space at $N = 10m$, where $m = 25$ is the number of prompts covered by $\pi$. The column represents the null distribution $P$. Italicized entries are FPRs. Note that unlike §4.1, the test is now run in higher-dimensional character space, which generally reduces power.

| Sample size | watermark | nf4 | int8 | FPR (fp32) |
|---|---|---|---|---|
| $N = 10m$ | 0.62 (0.05) | 0.38 (0.05) | 0.30 (0.06) | 0.05 (0.01) |
| $N = 50m$ | 1.00 (0.00) | 1.00 (0.01) | 0.96 (0.02) | 0.07 (0.01) |

Table 11: Power of the Hamming MMD test to distinguish between the fp32 Llama-3 8B and quantized or watermarked alternatives in $L = 1000$ character space. We compare power at $N = 10m$ and $N = 50m$, where $m = 25$ is the number of prompts covered by $\pi$. Increasing the sample size can compensate for losses of power.

| Model | $\tau = 0.5$ | $\tau = 1.5$ |
|---|---|---|
| Llama-3 8B | 1.00 (0.00) | 1.00 (0.00) |
| Llama-3.1 8B | 1.00 (0.00) | 1.00 (0.00) |
| Mistral 8B | 1.00 (0.00) | 0.97 (0.02) |

Table 12: Power of the Hamming MMD test to distinguish between models sampled at temperature $\tau = 1$ ($P$) vs. other temperatures ($Q$). This decoding parameter mismatch can be detected with near-perfect power.

### C.4 EFFECT OF THE PROMPT DISTRIBUTION

In §4, we evaluated tests on a prompt distribution $\pi$ drawn from a Wikipedia language modeling task and supported on $m = 25$ prompts. In this appendix, we extend these evaluations to different prompt distributions.

**Effect of the number of prompts** $m$. In Figure 8, we continue to use the Wikipedia task, but vary the number of prompts $m$ in the prompt distribution. Larger $m$ values increase the power of the test, suggesting that users benefit from testing many prompts together, so long as the sample size is increased proportionally.

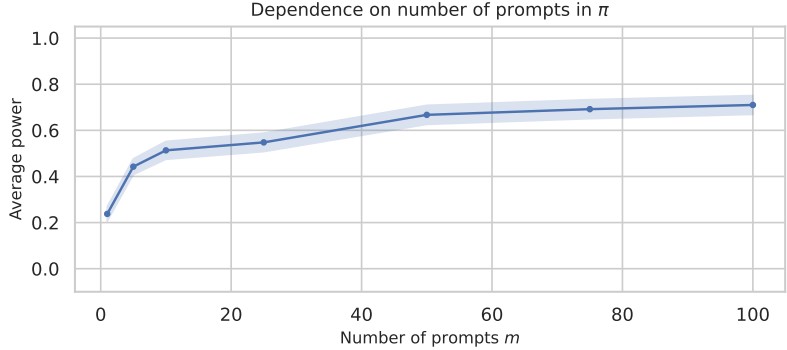

Figure 8: Power of the Hamming MMD test for different numbers of prompts $m$ in the prompt distribution. Results are averaged over ten random samples of $m$ prompts (for the Wikipedia task). The test is run with $N = 10m$ and $L = 50$ tokens, with $m$ varying from 1 to 100. Power increases with $m$, suggesting that users benefit from testing many prompts together.

**Effect of the task.** We explored whether the "open-endedness" of the task affects power: intuitively, one might expect that more creative tasks lead to completion distributions that are higher entropy, which might make testing harder. To test this, we evaluated power across several prompt distributions: on the most constrained side of the spectrum, we experimented with HumanEval (code, Box 3); on the most creative side, UltraChat (chatbot dialogues, Box B.1), and in the middle, language modeling with Wikipedia (Box 1). We additionally experimented with tightly concentrated prompts (just Wikipedia in English, just Wikipedia in French, etc.) vs. diverse prompt distributions (Wikipedia mixed with HumanEval and UltraChat).

Table 13 lists the Hamming MMD's power against the local alternatives (nf4, int8, watermark) at $N = 10m$ with tests conducted in token space ($L = 50$ token completions). We make several observations:

- We do not observe that creative tasks are harder to test than constrained tasks: if this were the case, we would expect the power to be lowest for UltraChat and highest for HumanEval across all alternatives.

- We also do not observe that diverse prompt distributions are harder to test than concentrated prompt distributions: if this were the case, we would expect the power to be higher for the single-language Wikipedia prompt distributions than the mixed-task prompt distributions.

- The prompt distribution mainly affects the difficulty of detecting quantization, not watermarking.

- For comparison, we also provide the results of the one-hot kernel in Table 14. The Hamming kernel consistently outperforms the one-hot baseline on all prompt distributions, matching our main conclusion of §4.1 about the relative strength of kernels.

| | | $m$ | watermark | nf4 | int8 | FPR (fp32) |
|---|---|---|---|---|---|---|
| HumanEval (constrained) | | 20 | 0.99 (0.00) | 1.00 (0.00) | 0.32 (0.00) | 0.06 (0.00) |
| | English only | 25 | 0.98 (0.01) | 0.49 (0.04) | 0.11 (0.01) | 0.06 (0.01) |
| | French only | 25 | 1.00 (0.00) | 1.00 (0.00) | 1.00 (0.00) | 0.06 (0.01) |
| Wikipedia (language modeling) | German only | 25 | 0.98 (0.01) | 0.71 (0.08) | 0.27 (0.06) | 0.05 (0.00) |
| | Spanish only | 25 | 1.00 (0.00) | 1.00 (0.00) | 1.00 (0.00) | 0.05 (0.01) |
| | Russian only | 25 | 0.98 (0.01) | 0.71 (0.05) | 0.24 (0.04) | 0.05 (0.01) |
| | All languages | 25 | 1.00 (0.00) | 0.93 (0.04) | 0.79 (0.08) | 0.05 (0.01) |
| UltraChat (creative) | | 20 | 0.89 (0.00) | 1.00 (0.00) | 0.04 (0.00) | 0.04 (0.00) |
| Wikipedia + UltraChat | | 25 | 0.81 (0.05) | 1.00 (0.00) | 0.10 (0.02) | 0.06 (0.01) |
| Wikipedia + UltraChat + HumanEval | | 25 | 0.95 (0.01) | 1.00 (0.00) | 0.35 (0.05) | 0.04 (0.01) |

Table 13: Hamming MMD's power against the local alternatives (nf4, int8, watermark) at $N = 10m$ with tests conducted in token space ($L = 50$ token completions).

| | | $m$ | watermark | nf4 | int8 | FPR (fp32) |
|---|---|---|---|---|---|---|
| HumanEval (constrained) | | 20 | 0.46 (0.00) | 0.77 (0.00) | 0.16 (0.00) | 0.05 (0.00) |
| | English only | 25 | 0.06 (0.01) | 0.05 (0.01) | 0.04 (0.01) | 0.01 (0.00) |
| | French only | 25 | 0.04 (0.01) | 0.06 (0.01) | 0.05 (0.00) | 0.01 (0.00) |
| Wikipedia (language modeling) | German only | 25 | 0.04 (0.01) | 0.04 (0.01) | 0.04 (0.01) | 0.01 (0.00) |
| | Spanish only | 25 | 0.04 (0.01) | 0.06 (0.01) | 0.04 (0.01) | 0.02 (0.00) |
| | Russian only | 25 | 0.04 (0.01) | 0.05 (0.01) | 0.03 (0.01) | 0.01 (0.00) |
| | All languages | 25 | 0.04 (0.00) | 0.05 (0.01) | 0.04 (0.01) | 0.01 (0.00) |
| UltraChat (creative) | | 20 | 0.07 (0.00) | 0.48 (0.00) | 0.04 (0.00) | 0.01 (0.00) |
| Wikipedia + UltraChat | | 25 | 0.09 (0.01) | 0.05 (0.01) | 0.07 (0.01) | 0.04 (0.01) |
| Wikipedia + UltraChat + HumanEval | | 25 | 0.22 (0.02) | 0.32 (0.03) | 0.05 (0.01) | 0.05 (0.01) |

Table 14: One-hot MMD's power against the local alternatives (nf4, int8, watermark) at $N = 10m$ with tests conducted in token space ($L = 50$ token completions).

## C.5 COMPARING TWO-SAMPLE AND APPROXIMATE GOODNESS-OF-FIT TESTS

In §2, we assumed only sample access to both the reference distribution $P$ and API $Q$. An alternative problem setup might give the auditor privileged access to $P$ to evaluate probabilities $P(y \mid x)$ for arbitrary (prompt, completion) pairs. Here, we compare the performance of two-sample tests to goodness-of-fit tests that leverage this privileged setting.

In an extreme case, evaluating probabilities is free – the auditor can fully describe $P(y \mid x)$ for all completions $y \in \mathcal{V}^{\leq L}$. Then goodness-of-fit tests, like the one-sample $L_1$ statistic, can be used to compare observed counts in $\mathcal{D}_Q$ to expected counts under $P$:

$$\mathrm{L1}_{\mathrm{gof}}(P, \mathcal{D}_Q) = \sum_{z \in \mathcal{X} \times \mathcal{V}^{\leq L}} |\mathbf{n}_z\{\mathcal{D}_Q\} - N \cdot P(y|x)\pi(x)| . \tag{12}$$

However, in practice, evaluating $P(y \mid x)$ for all $y \in \mathcal{V}^{\leq L}$ and all $x$ is intractable: as a concrete example, for our language modeling task on Llama-3, $m \cdot |\mathcal{V}^{\leq L}| = 25 \cdot 128\,000^{50} \approx 5 \times 10^{256}$. A more realistic scenario is that the auditor can only evaluate $P(y \mid x)$ for the observed $(x, y)$ in $\mathcal{D}_Q$. This leads to an approximation

of the goodness-of-fit test statistic:

$$\widehat{\text{L1}_{\text{gof}}}(P, \mathcal{D}_Q) = \sum_{z \in \mathcal{D}_Q} |\mathbf{n}_z\{\mathcal{D}_Q\} - N \cdot P(y|x)\pi(x)| . \tag{13}$$

We take a similar strategy for the one-sample $L_2$ test

$$\text{L2}_{\text{gof}}(P, \mathcal{D}_Q) = \sum_{z \in \mathcal{X} \times \mathcal{V}^{\leq L}} (\mathbf{n}_z\{\mathcal{D}_Q\} - N \cdot P(y|x)\pi(x))^2$$
$$\widehat{\text{L2}_{\text{gof}}}(P, \mathcal{D}_Q) = \sum_{z \in \mathcal{D}_Q} (\mathbf{n}_z\{\mathcal{D}_Q\} - N \cdot P(y|x)\pi(x))^2 , \tag{14}$$

the Pearson $\chi^2$ test

$$\chi^2_{\text{pearson}}(P, \mathcal{D}_Q) = \sum_{z \in \mathcal{X} \times \mathcal{V}^{\leq L}} \frac{(\mathbf{n}_z\{\mathcal{D}_Q\} - N \cdot P(y|x)\pi(x))^2}{N \cdot P(y|x)\pi(x)}$$
$$\widehat{\chi^2_{\text{pearson}}}(P, \mathcal{D}_Q) = \sum_{z \in \mathcal{D}_Q} \frac{(\mathbf{n}_z\{\mathcal{D}_Q\} - N \cdot P(y|x)\pi(x))^2}{N \cdot P(y|x)\pi(x)} , \tag{15}$$

and the truncated $\chi^2$ test (Balakrishnan & Wasserman, 2018)

$$\chi^2_{\text{truncated}}(P, \mathcal{D}_Q) = \sum_{z \in \mathcal{X} \times \mathcal{V}^{\leq L}} \frac{(\mathbf{n}_z\{\mathcal{D}_Q\} - N \cdot P(y|x)\pi(x))^2 - \mathbf{n}_z\{\mathcal{D}_Q\}}{\max\left(P(y|x)\pi(x), \frac{1}{|\mathcal{V}^{\leq L}|}\right)}$$
$$\widehat{\chi^2_{\text{truncated}}}(P, \mathcal{D}_Q) = \sum_{z \in \mathcal{D}_Q} \frac{(\mathbf{n}_z\{\mathcal{D}_Q\} - N \cdot P(y|x)\pi(x))^2 - \mathbf{n}_z\{\mathcal{D}_Q\}}{\max\left(P(y|x)\pi(x), \frac{1}{|\mathcal{V}^{\leq L}|}\right)} . \tag{16}$$

The only goodness-of-fit test that we consider which is unaffected by the approximation is the likelihood ratio test:

$$\text{LRT}(P, \mathcal{D}_Q) = -2 \sum_{z \in \mathcal{D}_Q} \mathbf{n}_z\{\mathcal{D}_Q\} \log\left(\frac{\mathbf{n}_z\{\mathcal{D}_Q\}}{N \cdot P(y|x)\pi(x)}\right) . \tag{17}$$

Figure 9 plots sample complexities of these (approximate) goodness-of-fit tests alongside the two-sample tests evaluated in the main text. The best goodness-of-fit tests outperform their two-sample counterparts in the extremely low-sample regime ($N < 1000$), but this trend reverses as $N$ increases. This is surprising — in theory, we would expect probability access to only increase power. These results suggest that the approximations compensating for limited evaluation budget can introduce bias, reducing the power of goodness-of-fit tests. We leave to future work ideas for the correction of this bias.

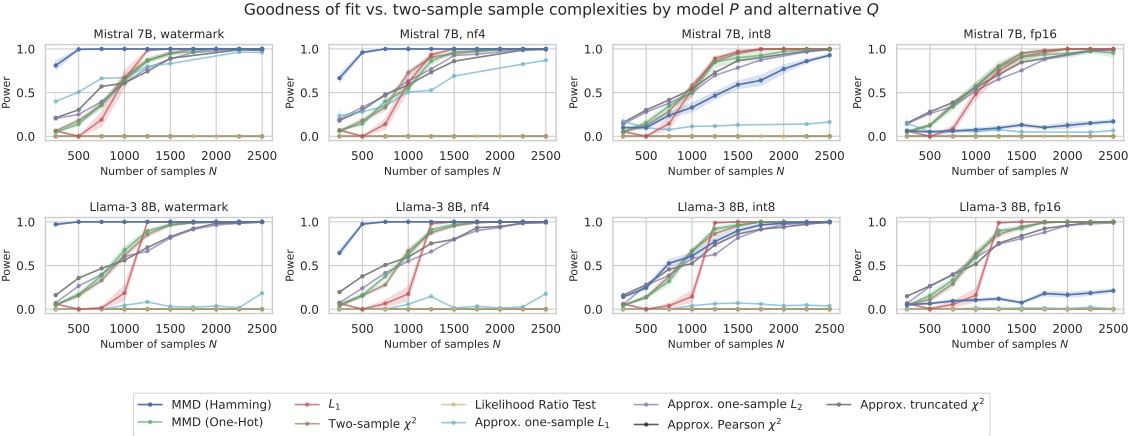

Figure 9: Power of two-sample MMD tests and goodness-of-fit tests. Each subplot represents a particular language model and alternative distribution $Q$. The goodness-of-fit tests are approximated by evaluating $P(y \mid x)$ only for observed $(x, y)$ in $\mathcal{D}_Q$. The truncated chi-squared and one-sample $L_2$ tests perform best out of the goodness-of-fit tests, while the likelihood ratio test and one-sample $L_1$ tests perform worst. Note that experiments were run on a different set of 10 prompt distributions than the main text; these prompts specifically had model probabilities saved (see Appendix B.1).

### C.6 ASYMMETRIC SAMPLING COSTS

In some cases, it may be significantly less expensive to sample from one distribution than the other. For example, the auditor may have unlimited compute to sample from the null distribution $P$, but limited monetary budget to sample from the API $Q$. In these cases, we show that it is possible to achieve slight power gains by increasing the sample size of the cheaper distribution, even while keeping the sample size of the expensive distribution fixed. Figure 10 fixes $|\mathcal{D}_Q| = 10m$ and varies $|\mathcal{D}_P|$ between $10m$ and $400m$. All test statistics see some increases in power, with the $L_1$ test seeing surprisingly large gains.

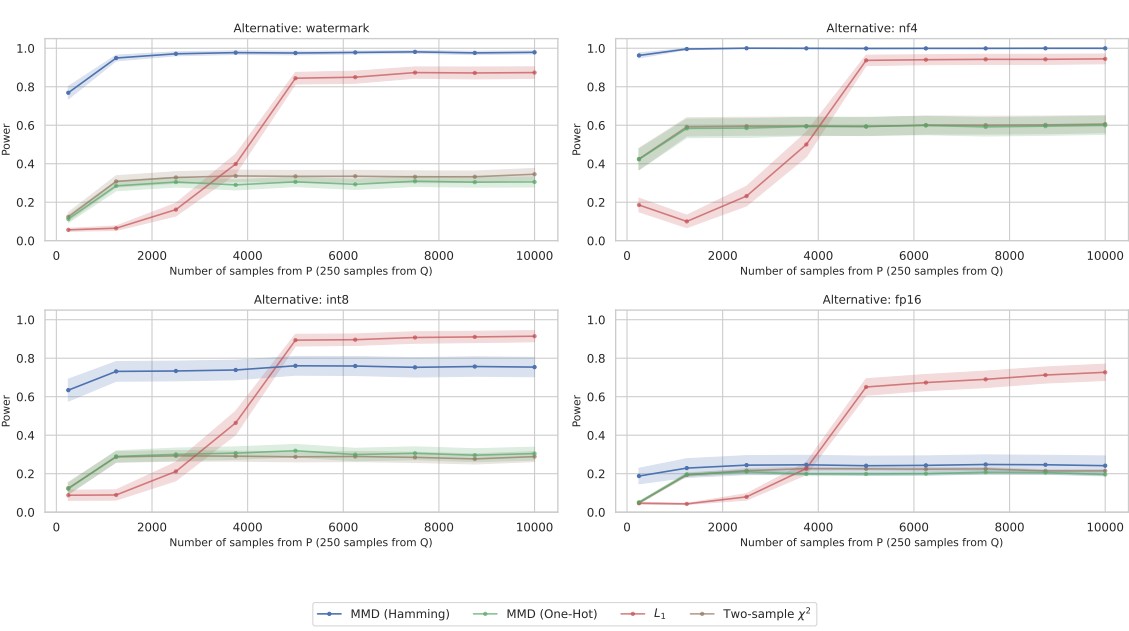

Figure 10: Power when the sample size from $P$ increases, when the sample size from $Q$ is fixed to 250. There are slight power gains for all test statistics, with the $L_1$ test seeing the largest gains.

## C.7 PERMUTATION PROCEDURE

Results in the main text simulate the test statistic's empirical distribution under the null by sampling datasets $\mathcal{D}_Q$ and $\mathcal{D}_P$ both from $P$. Here we validate these trends by conducting the same tests using a permutation procedure to estimate p-values (see Appendix A.2). Figures 11, 12, and 13 repeat the sample complexity, length, and asymmetric sampling cost experiments, but use the permutation procedure to estimate p-values. Because of the computational complexity of this step, we use 100 permutations, estimate power using 20 simulations, and only test Mistral 7B and Llama-3 8B. We observe that the permutation procedure maintains similar power levels to the bootstrap method, and trends from the previous figures are replicated.

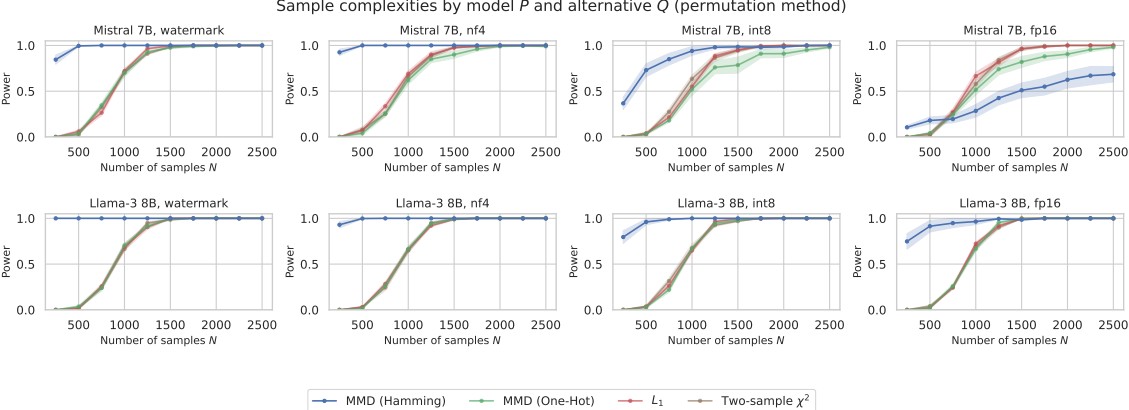

Figure 11: Sample complexities for different two-sample tests, stratified by the alternative distribution $Q$, but averaged over five language models and ten prompt distributions $\pi$. This figure parallels Figure 5 but uses the permutation procedure, rather than repeated sampling from $P$, to estimate p-values.

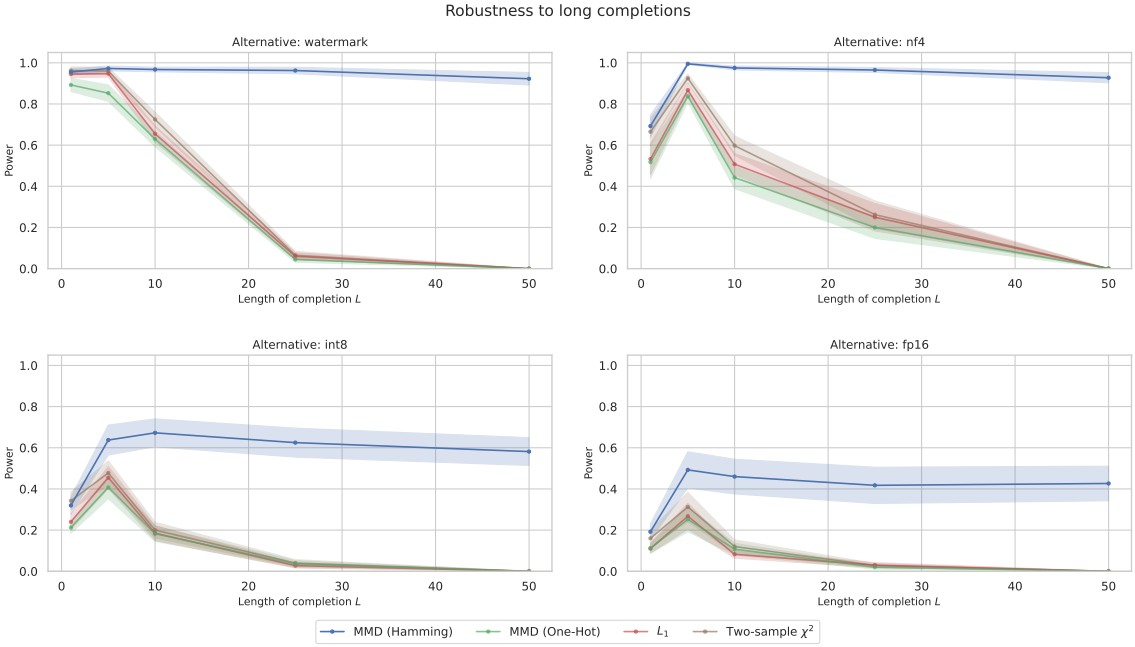

Figure 12: Simulated powers for different completion lengths $L$, stratified by the alternative distribution $Q$. This figure parallels Figure 7 but uses the permutation procedure, rather than repeated sampling from $P$, to estimate p-values.

Figure 13: Simulated power when the sample size from $P$ increases, when the sample size from $Q$ is fixed to 250. This figure parallels Figure 10 but uses the permutation procedure, rather than repeated sampling from $P$, to estimate p-values.

## C.8 ADDITIONAL RESULTS FROM §5

| | Wikipedia | | | | HumanEval | | UltraChat | |
| | 3 8B | 3.1 8B | 3 70B | 3.1 70B | 3 8B | 3.1 8B | 3 8B | 3.1 8B |
| --- | --- | --- | --- | --- | --- | --- | --- | --- |
| Amazon | 0.07 | 1.00 | 0.32 | 1.00 | 0.48 | 1.00 | 1.00 | 1.00 |
| Anyscale | 0.02 | — | — | — | — | — | — | — |
| Azure | 0.01 | 0.00 | 0.01 | 0.01 | 0.01 | 0.29 | 0.00 | 0.13 |
| Deepinfra | 0.04 | 0.00 | 0.04 | 0.00 | 0.08 | 0.19 | 0.04 | 0.09 |
| Fireworks | 0.04 | 0.04 | 0.01 | 0.01 | 0.01 | 1.00 | 0.03 | 0.90 |
| Groq | 0.03 | 0.07 | 0.02 | 0.59 | 0.01 | 0.98 | 0.05 | 0.35 |
| Perplexity | 1.00 | 1.00 | 1.00 | 1.00 | 1.00 | 1.00 | 1.00 | 1.00 |
| Replicate | 0.12 | — | 0.33 | — | 0.07 | — | 0.06 | — |
| Together | 0.01 | 0.00 | 0.00 | 0.00 | 0.00 | 0.27 | 0.01 | 0.07 |
| watermark | 0.30 | 0.00 | 0.07 | 0.03 | 0.40 | 0.85 | 0.06 | 0.38 |
| int8 | 0.07 | 0.00 | 1.00 | 0.96 | 0.01 | 0.45 | 0.01 | 0.03 |
| nf4 | 0.09 | 1.00 | 1.00 | 1.00 | 1.00 | 1.00 | 1.00 | 1.00 |
| *fp16* | *0.02* | *0.01* | *0.00* | *0.01* | *0.00* | *0.02* | *0.00* | *0.04* |
| *fp32* | *0.01* | *0.00* | *0.00* | *0.00* | *0.00* | *0.10* | *0.01* | *0.04* |

Table 15: Power against alternatives for the full audit. Table 1 in the main text thresholds power at 0.5 for the APIs. The FPRs against the full- and half-precision nulls are italicized.

| | 3 8B | 3.1 8B | 3 70B | 3.1 70B |
| --- | --- | --- | --- | --- |
| Amazon | ✗ | ✗ | ✗ | ✗ |
| Anyscale | ✓ | — | — | — |
| Azure | ✓ | ✓ | ✓ | ✓ |
| Deepinfra | ✓ | ✓ | ✓ | ✓ |
| Fireworks | ✓ | ✗ | ✓ | ✓ |
| Groq | ✓ | ✗ | ✓ | ✗ |
| Perplexity | ✗ | ✗ | ✗ | ✗ |
| Replicate | ✓ | — | ✓ | — |
| Together | ✓ | ✓ | ✓ | ✓ |

Table 16: Overall audit results, as also copied in Figure 1. The 8B models are tested on three prompt distributions (Wikipedia, HumanEval, UltraChat), while the 70B models are tested on Wikipedia. Tests from different prompt distributions are combined using a Bonferroni correction.

Table 15 shows the overall audit results. The 8B models are tested on three prompt distributions (Wikipedia, HumanEval, UltraChat), while the 70B models are tested on Wikipedia. Tests from different prompt distributions are combined using a Bonferroni correction in Table 17.

**Discussion: the composite null reduces power.** In §5, we use a composite null hypothesis that combines the fp32 and fp16 distributions. Table 18 shows the power of the Hamming MMD test in this composite null setting, stratified by model and prompt distribution. In general, power is reduced using the composite null. Power is generally highest on HumanEval, which collects longer completions than Wikipedia (250 vs. 50 tokens).

**Discussion: the Hamming MMD is correlated with absolute differences in HumanEval average accuracy.** In Figure 4 in the main text, we show that the Hamming MMD is correlated with the *absolute* differences in HumanEval average accuracy. However, the direction of this difference is inconsistent across APIs: sometimes APIs with nonzero MMDs have higher HumanEval accuracy than the local model, and

|            | 3 8B  | 3.1 8B | 3 70B | 3.1 70B |
|------------|-------|--------|-------|---------|
| Amazon     | 1.00  | 1.00   | 0.58  | 1.00    |
| Anyscale   | 0.02  | —      | —     | —       |
| Azure      | 0.02  | 0.10   | 0.01  | 0.01    |
| Deepinfra  | 0.01  | 0.04   | 0.04  | 0.01    |
| Fireworks  | 0.00  | 1.00   | 0.03  | 0.00    |
| Groq       | 0.01  | 0.75   | 0.00  | 0.59    |
| Perplexity | 1.00  | 1.00   | 1.00  | 1.00    |
| Replicate  | 0.02  | —      | 0.48  | —       |
| Together   | 0.01  | 0.09   | 0.00  | 0.00    |
| nf4        | 1.00  | 1.00   | 1.00  | 1.00    |
| int8       | 0.00  | 0.12   | 1.00  | 0.92    |
| watermark  | 0.12  | 0.45   | 0.50  | 0.06    |
| fp16       | *0.00* | *0.00* | *0.00* | *0.00* |
| fp32       | *0.00* | *0.06* | *0.00* | *0.00* |

Table 17: Power against alternatives for the full audit with the Bonferroni correction. The FPRs against the full- and half-precision nulls are italicized.

|           | Wikipedia | | | | HumanEval | | UltraChat | |
|-----------|-------|--------|-------|---------|-------|--------|-------|--------|
|           | 3 8B  | 3.1 8B | 3 70B | 3.1 70B | 3 8B  | 3.1 8B | 3 8B  | 3.1 8B |
| watermark | 0.30  | 0.00   | 0.07  | 0.03    | 0.40  | 0.85   | 0.06  | 0.38   |
| int8      | 0.07  | 0.00   | 1.00  | 0.96    | 0.01  | 0.45   | 0.01  | 0.03   |
| nf4       | 0.09  | 1.00   | 1.00  | 1.00    | 1.00  | 1.00   | 1.00  | 1.00   |
| fp16      | *0.02* | *0.01* | *0.00* | *0.01* | *0.00* | *0.02* | *0.00* | *0.04* |
| fp32      | *0.01* | *0.00* | *0.00* | *0.00* | *0.00* | *0.10* | *0.01* | *0.04* |

Table 18: Power against local alternatives for the composite null setting (§5) in character space ($L = 1000$), stratified by model and prompt distribution.

sometimes they have lower accuracy (see Table 19). Our main argument in the main text is that regardless of direction, one can predict the *magnitude* of change using the MMD. This is important in its own right; for example, it suggests that when the effect size returned by our test is large, users should do a manual examination for quality. For tasks with an automated verifier, like HumanEval, this two-step procedure may seem roundabout; however, for tasks without an automated accuracy metric, the MMD effect size provides users signal on when to invest manual resources into reviewing if the API is affecting performance.

| Version | Llama-3 8B Instruct | Llama-3.1 8B Instruct |
|---------|---------------------|------------------------|
| fp32 | 0.17 | 0.67 |
| fp16 | 0.17 | 0.39 |
| Amazon | 0.19 | 0.52 |
| Azure | 0.19 | 0.72 |
| Deepinfra | 0.18 | 0.64 |
| Fireworks | 0.19 | 0.68 |
| Groq | 0.14 | 0.26 |
| int8 | 0.15 | 0.53 |
| nf4 | 0.00 | 0.00 |
| Perplexity | 0.00 | 0.06 |
| Replicate | 0.22 | — |
| Together | 0.19 | 0.66 |
| watermark | 0.08 | 0.55 |

Table 19: Raw HumanEval average accuracy for the Llama-3 8B Instruct and Llama-3.1 8B Instruct models.

### C.9    COMPARING APIS TO EACH OTHER

In Figures 14 – 25, we compute the pairwise MMDs between APIs and quantized model weights for all prompt distributions (Wikipedia, HumanEval, UltraChat) and available models (Llama-3 8B, Llama-3.1 8B, Llama-3 70B, Llama-3.1 70B). We use spectral clustering with two components to discover groups of implementations. Providers that pass the audit in Table 1 are typically clustered with the null distributions $fp16$ and $fp32$, reflecting that they are distributionally close to these nulls.

Additionally, Figure 26 shows the estimated MMDs between APIs for each of the three prompt distributions for the Llama-3.1 405B model. Due to their size, we could not sample from the released weights for this large model directly, but we can still estimate the distances between APIs for this model.

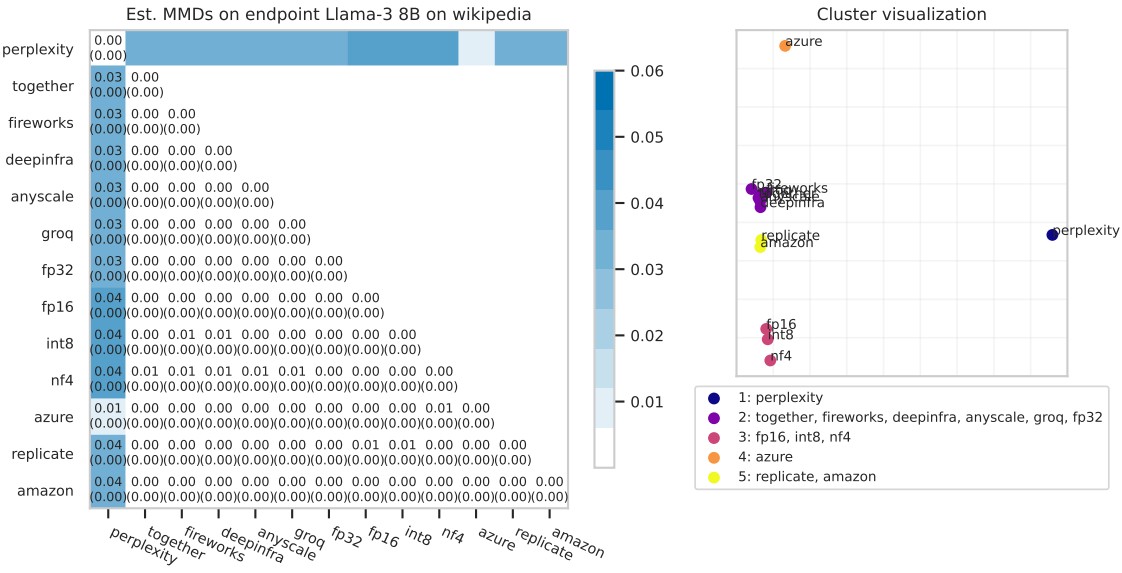

Figure 14: *(Left)* Hamming MMDs between APIs for the Llama-3 8B model on the Wikipedia prompt distribution. *(Right)* Visualization of the 2D spectral clustering components. Clusters are colored together.

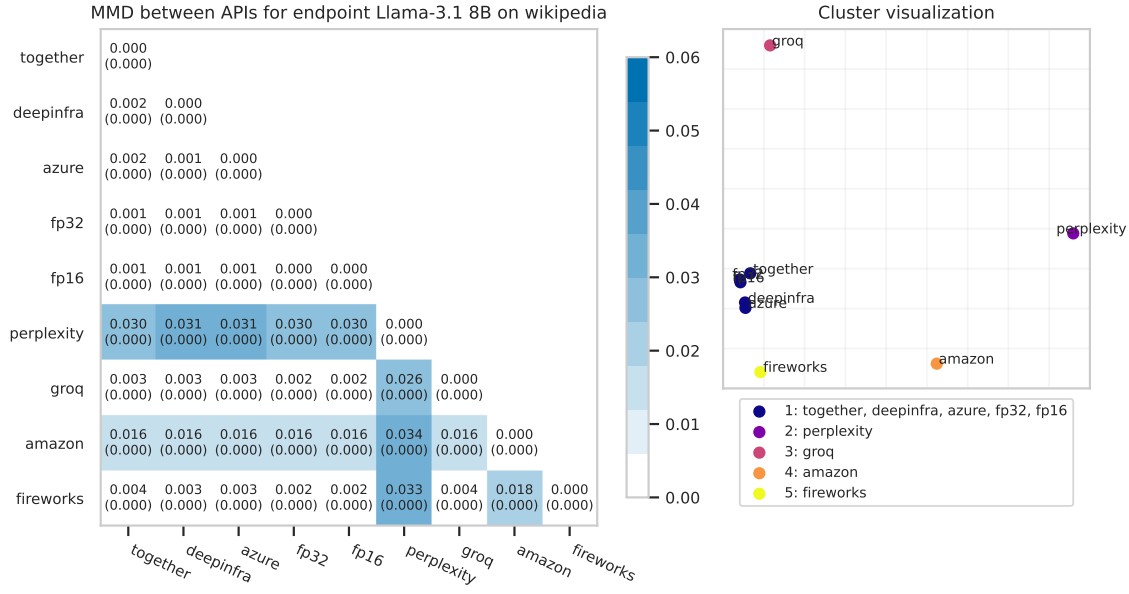

Figure 15: *(Left)* Hamming MMDs between APIs for the Llama-3.1 8B model on the Wikipedia prompt distribution. *(Right)* Visualization of the 2D spectral clustering components. Clusters are colored together.

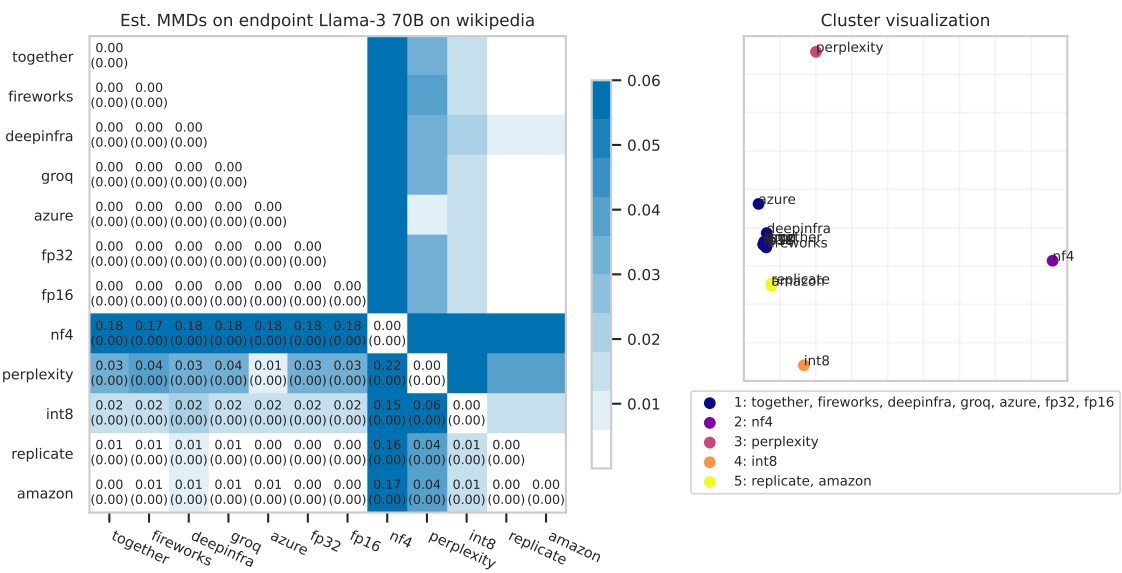

Figure 16: *(Left)* Hamming MMDs between APIs for the Llama-3 70B model on the Wikipedia prompt distribution. *(Right)* Visualization of the 2D spectral clustering components. Clusters are colored together.

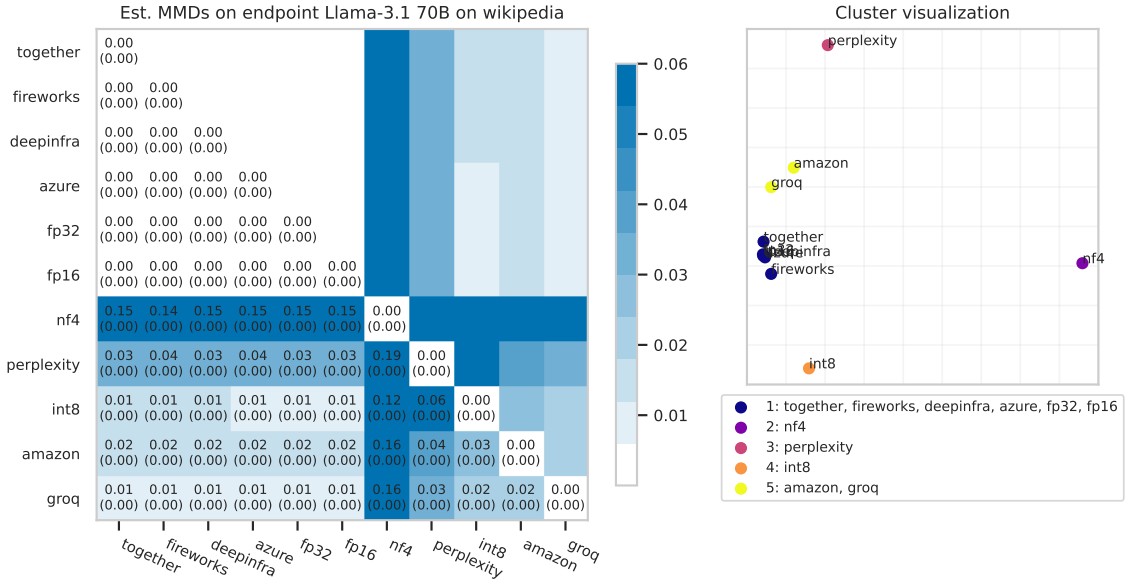

Figure 17: *(Left)* Hamming MMDs between APIs for the Llama-3.1 70B model on the Wikipedia prompt distribution. *(Right)* Visualization of the 2D spectral clustering components. Clusters are colored together.

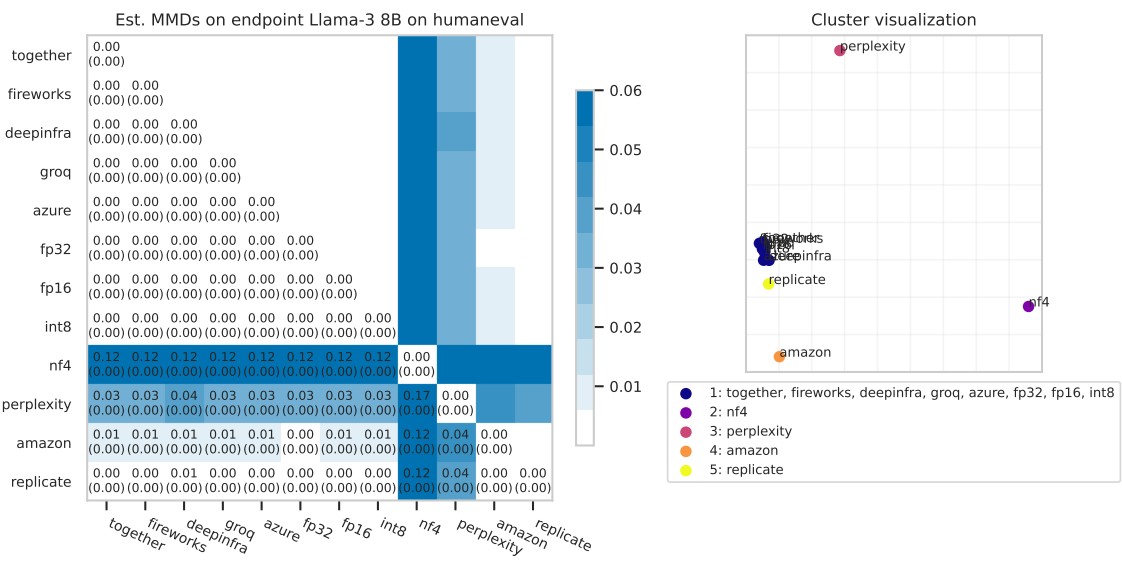

Figure 18: *(Left)* Hamming MMDs between APIs for the Llama-3 8B model on the HumanEval prompt distribution. *(Right)* Visualization of the 2D spectral clustering components. Clusters are colored together.

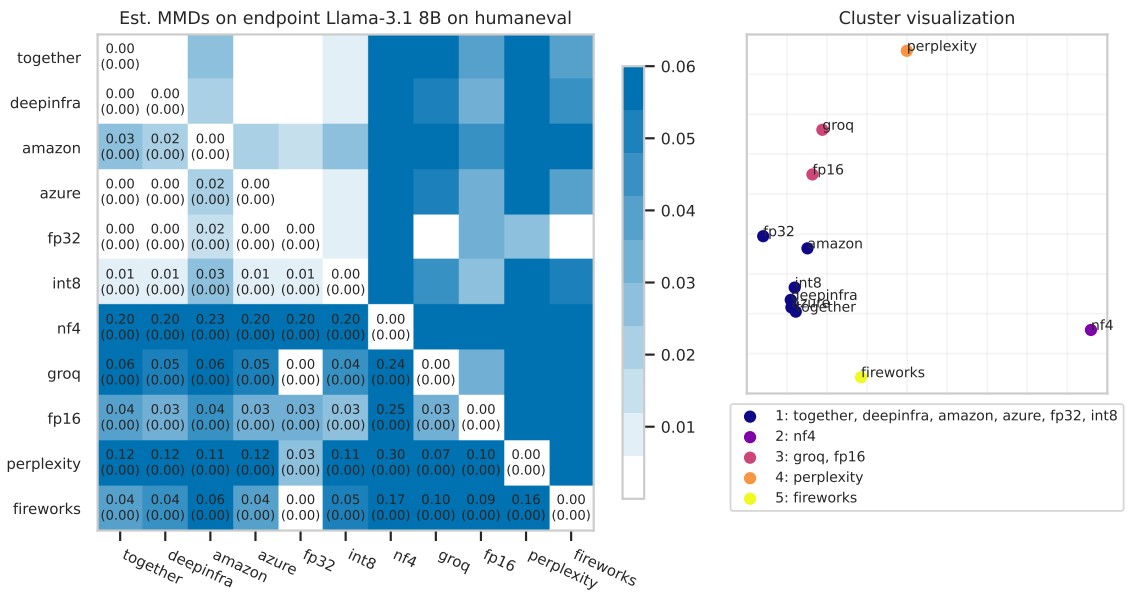

Figure 19: *(Left)* Hamming MMDs between APIs for the Llama-3.1 8B model on the HumanEval prompt distribution. *(Right)* Visualization of the 2D spectral clustering components. Clusters are colored together.

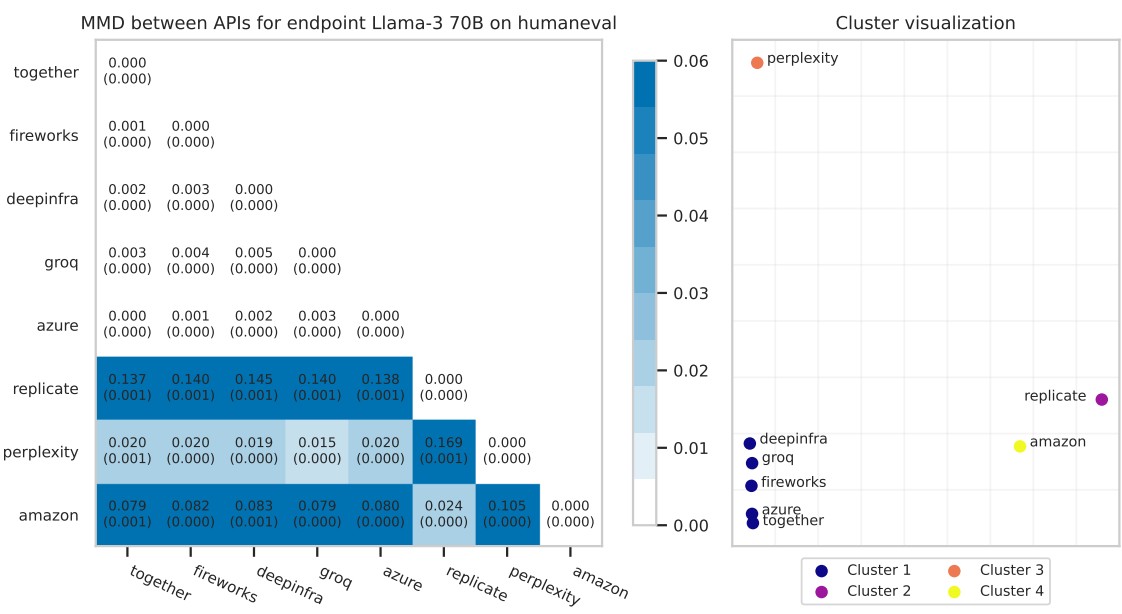

Figure 20: *(Left)* Hamming MMDs between APIs for the Llama-3 70B model on the HumanEval prompt distribution. *(Right)* Visualization of the 2D spectral clustering components. Clusters are colored together.

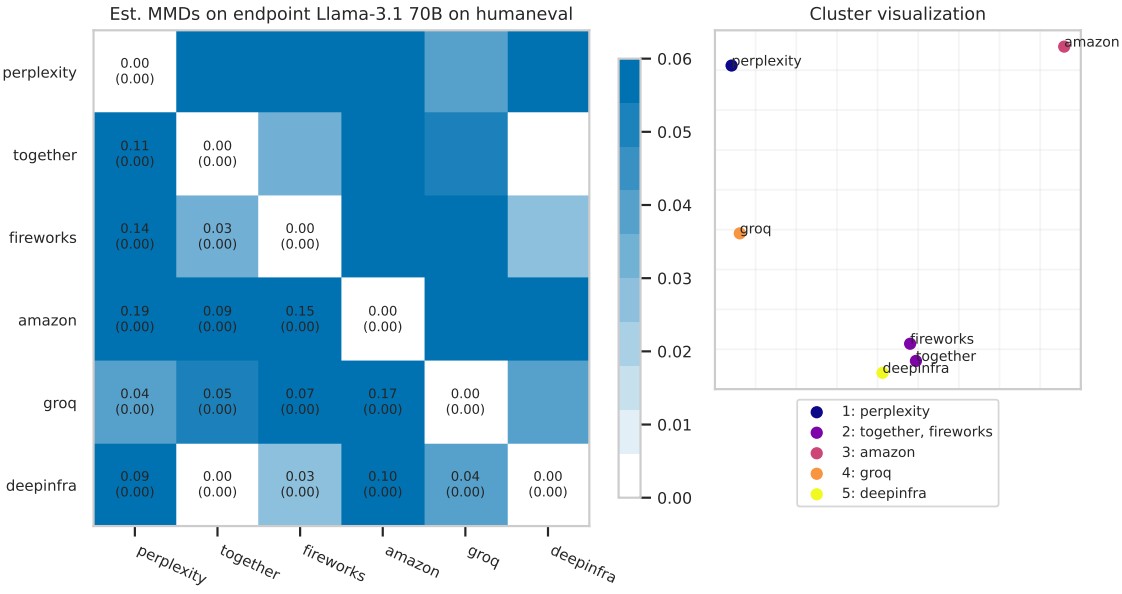

Figure 21: *(Left)* Hamming MMDs between APIs for the Llama-3.1 70B model on the HumanEval prompt distribution. *(Right)* Visualization of the 2D spectral clustering components. Clusters are colored together.

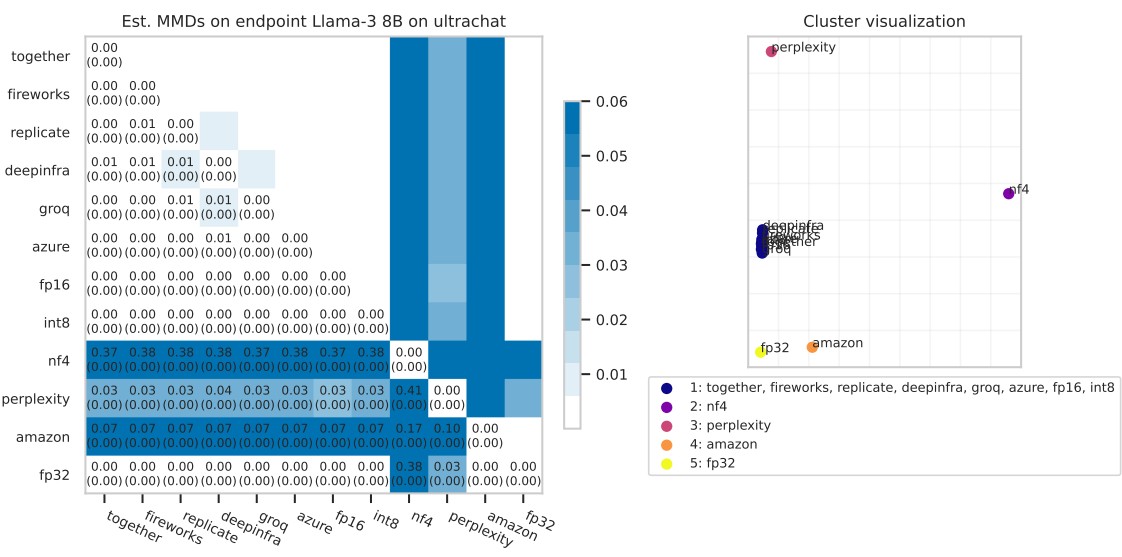

Figure 22: *(Left)* Hamming MMDs between APIs for the Llama-3 8B model on the UltraChat prompt distribution. *(Right)* Visualization of the 2D spectral clustering components. Clusters are colored together.

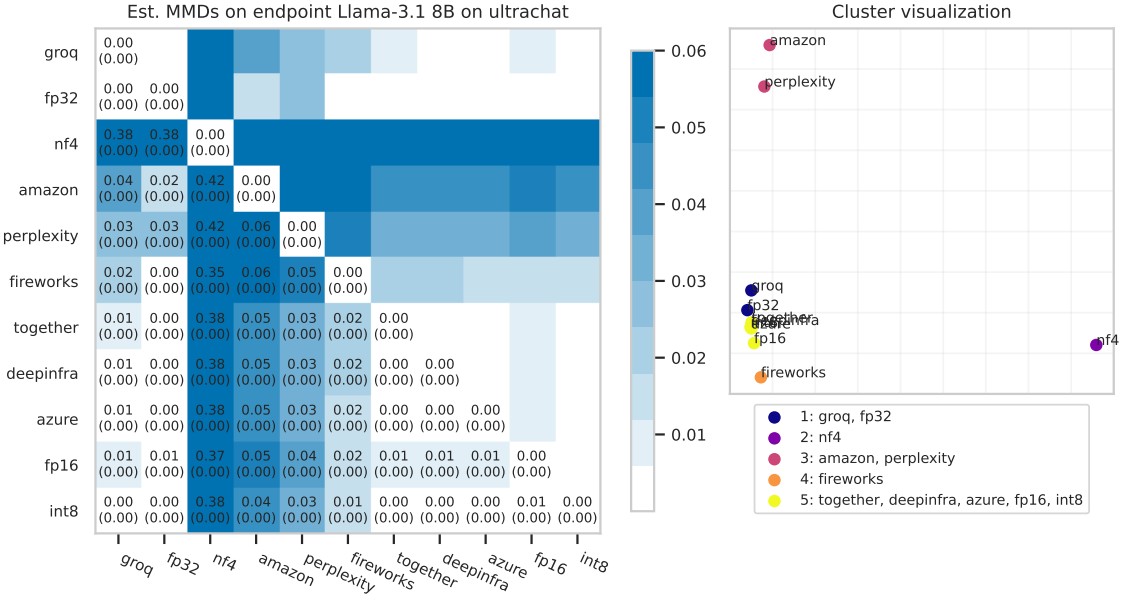

Figure 23: *(Left)* Hamming MMDs between APIs for the Llama-3.1 8B model on the UltraChat prompt distribution. *(Right)* Visualization of the 2D spectral clustering components. Clusters are colored together.

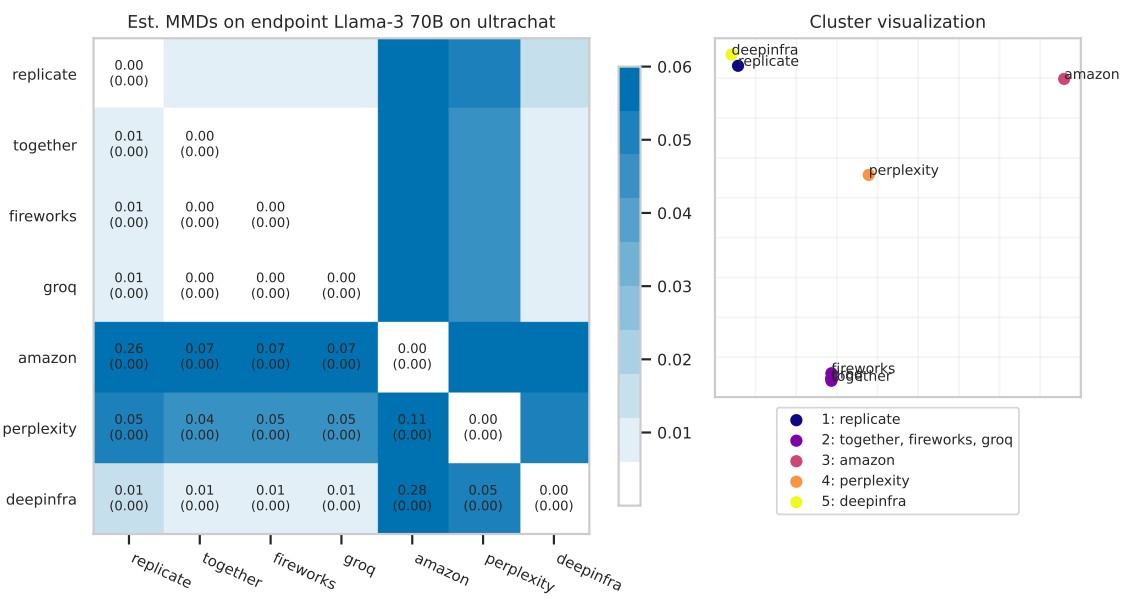

Figure 24: *(Left)* Hamming MMDs between APIs for the Llama-3 70B model on the UltraChat prompt distribution. *(Right)* Visualization of the 2D spectral clustering components. Clusters are colored together.

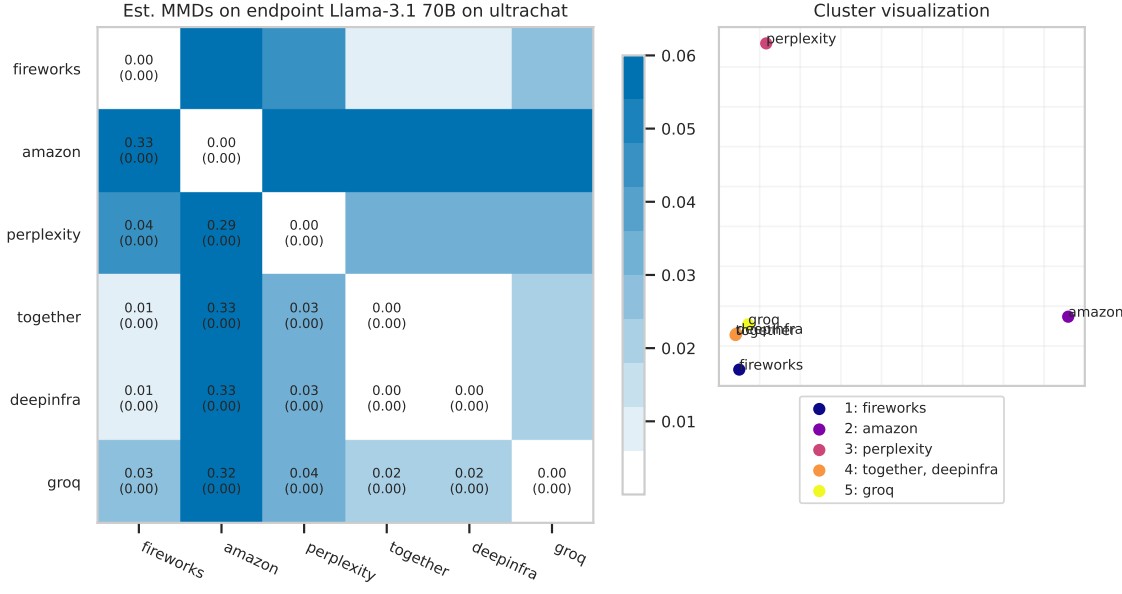

Figure 25: *(Left)* Hamming MMDs between APIs for the Llama-3.1 70B model on the UltraChat prompt distribution. *(Right)* Visualization of the 2D spectral clustering components. Clusters are colored together.

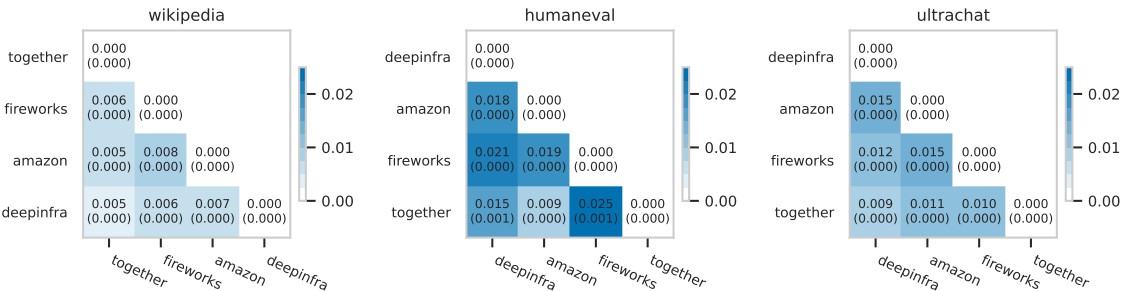

Figure 26: Estimated MMDs between Llama-3.1 405B APIs on three prompt distributions. Deepinfra publishes that they serve Llama-3.1 405B at fp8 precision. (Source: Deepinfra website)

## D DATA SAMPLES

In this appendix, we randomly sample completions from pairs of distributions to illustrate the differences between them. In general, we observe that it is challenging to visually distinguish between samples from different distributions, especially for longform tasks. This is because each distribution produces diverse outputs. Formal statistical tests that we describe in the main text are necessary to detect these differences.

In a few cases, detected differences between distributions are also visually obvious. For example, we observe that the Llama 70B-scale models quantize poorly, and their nf4 completions are degenerate. Different language models also often differ in how they begin completions. We also observe that some APIs (in particular, Perplexity) seem to be producing lower-entropy completions than the reference distribution, suggesting some form of caching or incorrect implementation of the temperature parameter.

### D.1 COMPARING MODELS TO QUANTIZED OR WATERMARKED VERSIONS

---

**Box 5: Samples from fp32 vs. fp16 Llama-3 70B (Instruct)**

**Prompt:** Continue the paragraph. Do not output anything except the continuation to the paragraph. Start the continuation immediately. "Calectasia is a genus of about fifteen species of flowering plants in the family Dasypogonaceae and ..."

| *fp32* | *fp16* |
|---|---|
| "...are native to eastern Australia and New Zealand, with one species, Calectasia articulata, also occurring in Lord Howe Island. The genus is thought to have evolved from a single ancestral species in the mid-Tertiary period. Calectasia | "...is endemic to Australia, with a majority of the species found in the southern and western parts of the country. The plants in this genus are shrubs or small trees with simple or compound leaves, and they produce small yellow or white flowers in clusters |
| "...is native to the Pacific region, with species found in Australia, New Zealand, and the Pacific Islands, including New Caledonia and the Solomon Islands. The genus is characterised by its unique inflorescences, which are racemes | "...are native to Australia and New Caledonia, which are characterized by their club-shaped inflorescences and solitary, axillary flowers. The flowers are typically yellow or mauve in color and are often highly fragrant, attracting a range |
| "...are native to the coastal regions of southeastern Australia, southern New South Wales, Victoria, Tasmania, and the coastal islands of Southern and Southwestern Australia, typically preferring well-drained sandy soils and full sun to partial shade." | "... is endemic to northern and eastern Australia, including Tasmania and southern New Guinea. They are spinifex grasses of shrub-like habit, typically ranging in size from 0.5 to 4 metres in height. The leaves are strap-shaped |
| "...is endemic to Australia and New Guinea, with most species being restricted to the southwestern corner of Western Australia. The plants are typically small to medium in size, and have leaves that are often reduced in size or absent, producing stems and inflores | "...is endemic to Australia, with species found in the coastal and mountainous regions of eastern New South Wales, the Australian Capital Territory, and southeastern Queensland. The plants are characterised by their simple, solitary, terminal flowers, which appear in late winter |
| "...is found in Australia and New Zealand, where they grow in coastal areas with sandy soils. They are also known as calectases or lomandras in some regions, and are commonly encountered in dry, sandy areas and scrublands. The | "... native to eastern Australia and New Caledonia. They are shrub to small tree species, often growing up to 2-6 metres in height, with simple serrated leaves that are usually 10-40 mm long and a thick cop |

---

---

**Box 6: Samples from fp32 vs. fp16 Llama-3 70B (Instruct)**

**Prompt:** Continue the paragraph. Do not output anything except the continuation to the paragraph. Start the continuation immediately.
"The agreement provides for training teams from the Libyan Stock Exchange in Tripoli and in London to..."

| *fp32* | *fp16* |
|---|---|
| "...enhance their skills in various aspects of securities trading and exchange management, including market regulation, risk management, and investor protection. The training program is designed to equip the Libyan Stock Exchange with the necessary expertise to operate efficiently and effectively, promoting economic growth | "...enable them to develop their technical skills and knowledge in areas such as market regulation, surveillance, and settlement systems. Additionally, the partnership will facilitate the exchange of best practices and expertise between the two exchanges, with the aim of enhancing the Libyan capital market |
| "...enhance their skills in areas such as market regulation, trading systems, and investor relations, with the goal of increasing the efficiency and transparency of the Libyan capital market. Additionally, the London Stock Exchange will provide technical assistance to help the Libyan Stock Exchange | "...gain expertise in modern trading practices, risk management, and regulatory compliance, with the aim of enhancing the Libyan capital market's infrastructure and attracting foreign investment to support the country's economic growth and development. The partnership is also expected to facilitate the listing of |
| "...enhance their skills and knowledge in modern stock exchange operations, with a focus on trading, settlement, and market surveillance systems. Additionally, the London Stock Exchange will provide technical assistance to help the Libyan Stock Exchange develop its regulatory framework and operational infrastructure, | "...to enhance their skills in trading, regulation, and market operations, with a focus on implementing international best practices and standards. The training programs will cover a range of topics, including trading rules, risk management, and corporate governance, and will be designed |
| "exchange knowledge and best practices in areas such as market operations, listing requirements, and investor relations, with the goal of strengthening the Libyan Stock Exchange's capacity to attract foreign investment and promote economic growth in Libya." | "...to enhance their skills and knowledge in areas such as market regulation, trading systems, and risk management. Additionally, the deal will facilitate the exchange of information and expertise between the two countries, with the LSE providing guidance on developing a more efficient and |
| "...to enhance their skills in areas such as market regulation, securities trading, and risk management, with the ultimate goal of strengthening the Libyan capital market and promoting economic development in the region." | "...exchange information and best practices on securities regulation, market surveillance, and trading operations. The training will focus on enhancing the capacity of the Libyan Stock Exchange to effectively regulate and monitor the Libyan capital market, as well as to improve its operational efficiency and competitiveness |

---

**Box 7: Samples from fp32 vs. int8 Llama-3 70B (Instruct)**

**Prompt:** Continue the paragraph. Do not output anything except the continuation to the paragraph. Start the continuation immediately.
"Weaver, Tom. I Talked with a Zombie: Interviews with 23 Veterans of Horror and Sci-Fi Films and Tele..."

| *fp32* | *int8* |
|---|---|
| "...vision, from the 1930s to the 1980s. Jefferson, NC: McFarland, 2009. Print. This book provides a unique perspective on the history of horror and science fiction, as it is told through | "...including Max Brooks,ed Troma Nightmares and Brian De Palma's Mission: Impossible III,ed Stan Winston,ed Edward Scissorhands,ed..." |
| "...vision. McFarland, 2011. This book provides a unique perspective on the horror and science fiction genres by offering firsthand accounts from individuals who played a significant role in their development. The interviews, which span several decades, offer insight into | "...including legendary actors like Bruce Campbell,..." |
| "...vision About Their Most Memorable Roles. McFarland, 2011. Print. This book offers a unique perspective on the horror and sci-fi genre, as it is comprised of interviews with actors and actresses who have played iconic roles in various | "...Famous Monsters,ed by Tom Weaver,ed features in-depth interviews with ed ed legends of horror and sci-fi,ed films and television,ed. Amonged the ed ed is Bruce Dern,ed,ed horror |
| "...vision. McFarland, 2011. This collection of interviews with professionals from the horror and science fiction genres offers valuable insights into the production of classic films and television shows. The book includes discussions with actors, writers, directors, and special | "...and TV,ed with the cast and crew of George Romero's Diary of the Dead,ing to recall the good old days of horror movie making. We caught up with Tippi Hedren,..." |
| "...vision, McFarland, 2008. This book provides a unique insight into the experiences of actors, directors, and special effects artists who worked on classic horror and science fiction films and television shows. The interviews offer a behind-the-scenes look | "...Famous Monsters and Others." |

---

**Box 8: Samples from fp32 vs. nf4 Mistral 7B (Instruct)**

**Prompt:** Continue the paragraph. Do not output anything except the continuation to the paragraph. Start the continuation immediately.
"The agreement provides for training teams from the Libyan Stock Exchange in Tripoli and in London to..."

| *fp32* | *nf4* |
|---|---|
| "collaborate on developing market infrastructure, regulatory frameworks, and trading systems. Additionally, the partnership allows for knowledge exchange on best practices in financial services, risk management, and market surveillance. This cooperation aims to foster a stable, thriving, | "ensure the seamless exchange of knowledge and best practices in trading, market surveillance, and regulatory compliance. This transatlantic collaboration aims to prepare the Libyan Stock Exchange for a more robust and resilient financial market, integrating international standards |
| "work collaboratively on developing and strengthening the market infrastructure, adhering to international standards, and facilitating the growth of the Libyan capital market." | "ensure compliance with international best practices and regulations, as well as enhancing capacity-building in various areas such as trading systems, surveillance, and market data management." |
| facilitate the exchange and sharing of knowledge in financial markets and regulations, while also ensuring the proper implementation of international best practices and standards. This collaborative effort aims to strengthen the Libyan Stock Exchange, enhance its efficacy, and promote sustainable economic growth | "ensure that they are equipped with the latest financial market trading techniques and best practices. This exchange of knowledge is aimed at modernizing the Libyan Stock Exchange and making it more competitive on an international level. The teams will undergo intensive training programs |
| "facilitate the development of modern trading practices and administrative systems, ensuring a harmonious integration with global financial standards." | "to share best practices and enhance the technical competencies of the teams, ensuring the smooth operation and sustained growth of the exchange." |
| ...further enhance their capabilities and gain in-depth knowledge of modern trading practices, regulatory frameworks, and technology implementations. This collaboration aims to strengthen the Libyan Stock Exchange's position in the international financial market, contribute to economic stability, | "share their expertise in various areas, including market analysis, trading practices, regulatory compliance, and technology infrastructure. This exchange of knowledge aims to equip the Libyan Stock Exchange with the necessary tools and skills to effectively manage their market, improve its transparency |

---

**Box 9: Samples from fp32 vs. nf4 Llama-3 70B (Instruct)**

**Prompt:** Continue the paragraph. Do not output anything except the continuation to the paragraph. Start the continuation immediately.
"The agreement provides for training teams from the Libyan Stock Exchange in Tripoli and in London to..."

| *fp32* | *nf4* |
| --- | --- |
| "...enhance their skills in areas such as market regulation, surveillance, and settlement systems, with the aim of improving the overall efficiency and transparency of the Libyan capital market. Additionally, the agreement outlines plans for the development of a new trading platform, which | Course!!!!!!!!!!!!!!!!!!!!!!!!!!!!!!!!!!!!!!!!!!!!!!!!! |
| "familiarize themselves with the operational systems and regulatory frameworks of the London Stock Exchange, with the aim of enhancing the capacity and competitiveness of the Libyan capital market. Additionally, the partnership will facilitate the exchange of best practices and expertise in areas such as risk | Earth!!!!!!!!!!!!!!!!!!!!!!!!!!!!!!!!!!!!!!!!!!!!!!!!! |
| "...enhance their knowledge and skills in areas such as market regulation, trading operations, and risk management, thus enabling them to develop a more efficient and transparent capital market in Libya. The cooperation is expected to facilitate the integration of the Libyan Stock Exchange into | certain!!!!!!!!!!!!!!!!!!!!!!!!!!!!!!!!!!!!!!!!!!!!!!!! |
| "...gain expertise in modern stock exchange operations, including trading, settlement, and regulatory practices. This training will be conducted by experienced staff from the London Stock Exchange and will take place over a period of six months. Additionally, the agreement includes the provision of | healthy!!!!!!!!!!!!!!!!!!!!!!!!!!!!!!!!!!!!!!!!!!!!!!!! |
| "...design and implement a new trading system, which will enable the Libyan Stock Exchange to operate more efficiently and in line with international standards. The training will cover all aspects of the new system, including its technical capabilities, operational procedures, and risk management strategies | Reverse!!!!!!!!!!!!!!!!!!!!!!!!!!!!!!!!!!!!!!!!!!!!!!!! |

---

**Box 10: Samples from fp32 vs. watermarked Llama-3.1 8B (Instruct)**

**Prompt:** Continue the paragraph. Do not output anything except the continuation to the paragraph. Start the continuation immediately.
"The agreement provides for training teams from the Libyan Stock Exchange in Tripoli and in London to..."

| *fp32* | *Watermark* |
|---|---|
| "...further develop and refine their skills in the areas of corporate governance, risk management, and financial markets operations, with the goal of strengthening the Libyan Stock Exchange's capacity to effectively play a central role in the country's economic development." | "...enable them to develop the needed expertise to list Libyan companies on both exchanges, with the ultimate goal of increasing access to international finance for these companies and promoting economic growth in Libya." |
| "...learn about international best practices, listing rules, and compliance regulations, as well as to discuss ways to enhance the Libyan Stock Exchange's capacity and increase investor confidence." | "...enable the implementation of international best practices and improve market operations, assessment, and enforcement. The project also aims to develop the institutional capacity of the Libyan Stock Exchange to list and regulate Sharia-compliant securities, and to establish a robust framework for corporate governance |
| "...enhance their capacity and knowledge in areas such as corporate governance, risk management, and trading system implementation, ultimately contributing to the development of a more robust and efficient Libyan capital market." | "learn best practices for listing and disclosure standards, as well as the use of the London Stock Exchange's ELITE programme to support the growth of Libyan companies." |
| "...familiarize them with international best practices in exchange operations, market making, and listing requirements. The teams will also undergo negotiation and stakeholder management training to enhance their capability to engage effectively with investors and issuers. Additionally, the agreement includes a | "...familiarise them with the best practices of international exchange organisations, and for regular monitoring of Libya's progress towards regulatory compliance with international standards." |
| "...facilitate greater integration of Libyan listed companies into the international capital market and to enhance the exchange's technical capabilities. The London Stock Exchange's internationally recognized standards and rules will be shared with the Libyan Stock Exchange to improve the efficiency and transparency of trading | "...optimize trading and listings systems, and also allocated a sum for implementing a state-of-the-art trading platform at the Libyan Stock Exchange in Tripoli." |

## D.2 COMPARING MODELS TO OTHER MODELS

---

**Box 11: Samples from Llama-3 8B (Instruct) vs. Phi-3 Mini (Instruct)**

**Prompt:** Continue the paragraph. Do not output anything except the continuation to the paragraph. Start the continuation immediately.
"Gung Ho! (full title: Gung Ho!: The Story of Carlsoń Makin Island Raiders) is a 1943 American war ..."

| *Llama-3 8B* | *Phi-3 Mini* |
|---|---|
| "...drama film directed by Ray Enright, starring Randolph Scott and James Cagney. The story is based on the true events of Carlson's Makahiki Raiders, a group of U.S. Marine Raiders who, under the command of Lt | war film directed by Robert Wise and produced by Pandro S. Berman for Samuel Goldwyn Productions. It is based on the true story of the Makin Island raid during World War II. The film tells the tale of U |
| "...drama film directed by Robert Wise, produced by Samuel G. Engel, and starring Randolph Scott. The movie is based on the true story of Alamo Scout Company B, 2nd Battalion, 1st Marine Regiment, during the Pacific | ...comedy-drama film directed by Mel Ferrer. Set during the Second Battle of the Philippines, the story is based on the true events of World War II and follows the 21st Marine Raider Battalion, led by Captain Howard |
| ..."comedy-drama film directed by Sam Wood, starring Randolph Scott, James Gleason, and Robert Ryan. The film is based on the true story of the United States Marine Corps' 2nd Marine Raider Battalion, who were tasked with the | war biographical film directed by Robert Lewis and John Ford. Set during World War II, the film focuses on the exploits of Lieutenant Commander Evans Carlson and his specialized American task force in Japan, highlighting the innovative guerrilla |
| ..."film directed by Ray Enright and starring Randolph Scott, Alan Hale, Jr., and Noah Beery Jr. The film is based on a true story about a group of United States Marine Raiders who launched a successful raid on the Japanese occupied Makin | war film directed by John Sturges, written by Ward Elliott, and adapted from the book "Gung Ho!" by E.R. Hewett. This film chronicles the daring and harrowing experiences of the Makin |
| "...film directed by Ray Enright and starring Randolph Scott, Barbara Britton, and Frank Mitchell. The movie is based on the true story of the formation of the Makin Island Attack Unit, a group of United States Marine Raiders who were tasked with | ...decion film directed by Robert Hendrickson and produced by William Alland. The film stars Victor Mature, Betty Field, and Sabu in the lead roles. Set during World War II, it chronicles the story of a group of |

---

---

**Box 12: Samples from Llama-3.1 70B (Instruct) vs. Gemma-2 9B (Instruct)**

**Prompt:** Continue the paragraph. Do not output anything except the continuation to the paragraph. Start the continuation immediately.
"Gung Ho! (full title: Gung Ho!: The Story of Carlsonś Makin Island Raiders) is a 1943 American war ..."

| *Llama-3.1 70B* | *Gemma-2 9B* |
|---|---|
| "... film directed by Ray Enright and starring Randolph Scott. The movie is based on the views attributed to the commander of the 2nd Raider Battalion, Evans Carlson, and the Battle of Makin Island (also known as the Battle of Butar | film directed by Ray Enright and written by .""" |
| "... film directed by Ray Enright and starring Randolph Scott. The film is largely fictional, but is based on the true story of the Carlson's Raiders, the 2nd Marine Raider Battalion, which was activated on June 19, 1942 | film, directed by Ray Enright and starring Wallace Ford. The film is based on a 1942 book by T.R. Fehrenbach about the famous amphibious raid of Makin Island in 1942 |
| "... film directed by Ray Enright, starring Randolph Scott, and based on the true story of the Carlson's Raiders, a special military unit set up by President Franklin D. Roosevelt as an American counterpart to the British Commandos, that took part in | film about the real-life exploits of United States Marine Corps scout-snipers on Makin Island during World War II. The film was directed by Ray Enright and stars Jon Hall as Lt. Alexander Bonnyman, a stern Marine officer |
| "... film directed by Ray Enright that follows the story of Lieutenant Colonel Evans Carlson's 2nd Marine Raider Battalion and their 30-day occupation of Makin Island in the Gilbert Islands during August 1942. The film stars Randolph Scott, Robert | film, directed by Henry H. Arbitell, narrated by Philip J. Freeman, with music by William O. Steward and appeared in the following excerpt from the Encarta Encyclopedia:"Gung Ho! (full title: Gung |
| film directed by Ray Enright, starring Randolph Scott, and based on the true story of the Carlson's Raiders and the Makin Island raid. The screenplay was written by W. S. Burnett and data collected from the USMC. The film was | film based on the real-life United States Marine Corps raid on Makin Island during World War II. |

**Box 13: Samples from Llama-3 8B (Instruct) vs. Llama-3.1 70B (Instruct)**

**Prompt:** Continue the paragraph. Do not output anything except the continuation to the paragraph. Start the continuation immediately.
"Gung Ho! (full title: Gung Ho!: The Story of Carlsonś Makin Island Raiders) is a 1943 American war ..."

| Llama-3 8B | Llama-3.1 70B |
|---|---|
| ...film directed by Robert Wise and starring Randolph Scott. The movie is based on the true story of the Carlson's Patrol, also known as the Makin Island Raiders, a small group of U.S. Marines who conducted a daring raid on Japanese-held | "... film directed by Ray Enright, starring Randolph Scott. The film's plot is based on the real-life World War II Makin Island raid led by Lieutenant Colonel Evans Carlson's 2nd Marine Raider Battalion. Scott plays a fictional commander of the |
| "...drama film directed by Ray Enright, starring Alan Ladd as John W. "Moe" Smith, a Marine raider who takes part in a special operations mission on Wake Island during the early years of World War II. The film | film directed by Ray Enright, starring Randolph Scott, and produced by Walter Wanger. The screenplay was written by W. S. [Walker] Crowther and Joseph Hoffman, corrected to Hoffman's dictation, based on the book of the same |
| "...drama film directed by Ray Enright and starring Randolph Scott, James Wong Howe, and Robert Ryan. The film is based on the true story of the U.S. Marine Raider Battalion, led by Major Evans Carlson, who captured the strategic Mak | film directed by Ray Enright, starring Randolph Scott and Alan Curtis. The film is based on the story of the Carlson's Raiders and their famous raid on Makin Island during World War II. It attempts to convey the message that battle is not just |
| ...film directed by Ray Enright and starring Alan Ladd, Robert Preston, and Veronica Lake. The movie is based on a true story about a group of U.S. Marines stationed on Makin Island, also known as Butaritari, | film directed by Ray Enright and starring Randolph Scott. The film is based on the true story of the Carlson's Patrol, a 30-day raid on Japanese-held Makin Island between August 17 and September 25, 1942, by |
| "...drama film directed by Ray Enright and starring Randolph Scott, Alan Curtis, and James Ellison. The movie is based on the true story of the Carlson's Raiders, a group of United States Marine Raiders who seized the strategic Makin Island from | film directed by Ray Enright and starring Randolph Scott, based on the true story of the 2nd Marine Raider Battalion's World War II raid on Makin Island, which was led by Lieutenant Colonel Evans Carlson. |

---

**Box 14: Samples from Llama-3.1 8B (Instruct) vs. OLMo 7B (Instruct)**

**Prompt:** Continue the paragraph. Do not output anything except the continuation to the paragraph. Start the continuation immediately.
"Gung Ho! (full title: Gung Ho!: The Story of Carlson's Makin Island Raiders) is a 1943 American war ..."

| *Llama-3.1 8B* | *OLMo 7B* |
|---|---|
| "... film directed by Ray Enright. The film is based on a true story about the 2nd Marine Raider Battalion, under the command of Lieutenant Colonel Evans F. Carlson, who occupied the Makin Island (now known as Butaritari | films, follows the inspiring true story of Colonel Carl L. Smith's 4,000-man Corps of famous Japanese-American Battle Ground Warriors, who triumphantly embarked on a 50-day, 90-mile Hawaiian desert race to build a trenches and |
| ...film directed by Ray Enright, starring Robert Hutton and Preston Foster. The movie is based on the book of the same name, If You Want to Look for Treasure by Maj. Carl G. Carlson, USMCR, which tells the | drama film directed by Ray DeFinis and featuring an ensemble cast that includes William "Dub" Jones, Harold Gobber, Tom Tully, Alan Cotton, and Philip Merivale. Set during World War II, the film follows the |
| ... film directed by Ray Enright, starring Randolph Scott, Jameson Thomas and Carole Landis. The film is an adaptation of the 1943 non-fiction book of the same name by Robert P. Landry and Frank R. McCoy, | mercenaries, mostly African-American, who are tasked with building and managing a new humangeous village for the local Malayan¿'s (the indigenous people) and teaching them modern agriculture and forensics. observe the orderly and disciplined |
| ...film directed by Ray Enright, starring Wallace Beery and Carl Esmond. The screenplay was written by Jo Swerling and Joseph Hoffman, based on the 1943 book "Gung Ho!" by Robert P. Henderson. | film, directed by Henry H. Arbitell, narrated by Philip J. Freeman, with music by William O. Steward and appeared in the following excerpt from the Encarta Encyclopedia: "Gung Ho! (full title: Gung |
| "...buddy film directed by Ray Enright, starring Robert Hutton as Major Jim Curry and Randolph Scott as Major Carl Entschede. The film is loosely based on a true story of Carlson's Makin Island Raiders, a United States Marine Corps | animation by Tatsuo Stella, with uneorologisticatley accurate meteorological animations by Chica Tokiji, the film celebrates the heroic deeds of a devoted team of outcasts led by frontiersman Sam Dwier Brown, whose words |

---

D.3 COMPARING APIS

---

**Box 15: Samples from fp32 vs. Perplexity for Llama-3 8B (Instruct)**

**Prompt:** Continue the paragraph. Do not output anything except the continuation to the paragraph. Start the continuation immediately. "Calectasia is a genus of about fifteen species of flowering plants in the family Dasypogonaceae and ..."

| *fp32* | *Perplexity* |
|---|---|
| "...are native to the southern and eastern regions of Australia, New Zealand, and surrounding islands. The genus is characterized by its unique flowers, which are often yellow or orange in color and have five or six petals. Calectasia plants are often small | "...is native to Australia, where they are commonly found in dry, open forests and woodlands. The plants are shrubs or small trees, typically growing to a height of 1-5 meters, with a spreading habit and a rounded crown. |
| "...is endemic to Australia, with most species found in the states of Western Australia and Northern Territory. The plants are generally woody shrubs or small trees that grow up to 10 meters tall, with leaves that are typically divided into three lobes | "...are native to Australia, where they are commonly found in dry, open forests and woodlands. The plants are shrubs or small trees that typically grow to a height of 1-5 meters, with a spread of around 1-3 |
| "...is primarily native to Australia, with some species also found in New Guinea and the Pacific Islands. The plants are generally shrubs or small trees, typically growing up to 2-5 meters tall, with simple or lobed leaves that are often | "...is native to Australia, where they are found in a variety of habitats, including forests, woodlands, and grasslands. The plants are characterized by their distinctive, often brightly colored flowers, which are typically arranged in clusters or spikes. The leaves |
| "...are native to the temperate and subtropical regions of Australia. They are small to medium-sized shrubs or trees, typically growing up to 5 meters tall, with a distinctive, layered or shrubby habit. The leaves are simple, alternate | "...are native to Australia, where they are commonly found in dry, open forests and woodlands. The plants are shrubs or small trees, typically growing to a height of 1-5 meters, and have a distinctive, sprawling habit. The |
| "...is native to Australia and New Guinea. The plants are shrubs or small trees that typically grow to a height of around 1-5 meters, with a spread of up to 2-3 meters. The leaves are simple, alternate, | "...are native to Australia, where they are found in a variety of habitats, including forests, woodlands, and grasslands. The plants are characterized by their distinctive, often brightly colored flowers, which are typically arranged in clusters or spikes, and their |

---

---

**Box 16: Samples from fp32 vs. Amazon Bedrock for Llama-3 70B (Instruct)**

**Prompt:** Continue the paragraph. Do not output anything except the continuation to the paragraph. Start the continuation immediately.
"Weaver, Tom. I Talked with a Zombie: Interviews with 23 Veterans of Horror and Sci-Fi Films and Tele..."

| *fp32* | *Amazon Bedrock* |
|---|---|
| "...visions. Jefferson, NC: McFarland, 2004. Weldon, Michael J. The Psychotronic Video Guide. New York: St. Martin's Press, 1996. Weldon, Michael J. The Psychot | "...vision. McFarland & Co, 2004. This unique collection brings together a group of people who have been behind the scenes of some the most beloved and enduring horror and science fiction films and TV series, providing insight into how their |
| ...vision. Jefferson, NC: McFarland & Company, Inc., 2004. ISBN 0-786-48181-0; Weaver, Tom. Interviews with B Science Fiction and Horror Movie Makers: Writers, Producers, | "...vision. McFarland, 2004. ISBN 0-7864-1684-5. Weaver, Tom. They Fought in the Creature Features: A History of Makeup Special Effects Artists. McFarland, 2010 |
| "...vision. Jefferson, NC: McFarland, 2004. ISBN 0-7864-1686-1. Willemen, Paul. "The Fantastic." In Film Comment 14, no. 6 (1978): | "...Films and Television Shows. McFarland & Company, 2009" The author delVES into conversations with behind-the-scenes veterans, revealing lesser-known stories from beloved franchises such as Star Trek and Friday the 13th, providing insight into |
| "...visions. Jefferson, NC: McFarland & Co., 2011, p112-114, ISBN 0786444923; Weaver, Tom. Monsters, Mutants and Heavenly Creatures: Confessions of 14 Classic Sci-Fi/H | "...vision. McFarland, 2004. ISBN 0-7864-1684-5. Weaver, Tom. They Fought in the Creature Features: A History of Makeup Special Effects Artists. McFarland, 2010 |
| "...vision. Applying an informal, conversational style to put his subjects at ease, Weaver elicits candid, often humorous, and occasionally revelatory discussions about the highs and lows of life in the genre movie trenches. Among the emotive chat-weaving | "...vision. New York: McFarland, 2004. Wells, Paul (ed.). Neo Noir. Short Films by Contemporary Film-makers. UK: Wallflower/Columbia University Press, 1999." |

---

**Box 17: Samples from fp32 vs. Deepinfra for Llama-3.1 70B (Instruct)**

**Prompt:** Continue the paragraph. Do not output anything except the continuation to the paragraph. Start the continuation immediately. "Litzman served on the Knesset's Internal Affairs Committee from 1999 to 2001, and as the Deputy Chai..."

| *fp32* | *Deepinfra* |
|---|---|
| "...ordinator of the Second Netanyahu government from 1999 to 2001. He was appointed Minister of Housing and Construction by Ariel Sharon in 2001, a position he held until 2002, and has served as the Deputy Minister of Health | "...Committee chairman. With the formation of Ariel Sharon's coalition government in 2001, Litzman was appointed Minister of Health, a position within which he continues to be until June 2011." |
| "...rman of the Knesset from 2009 to 2013. In 2001, he was appointed as the Minister of Health in Ariel Sharon's unity government, a position he held until Ehud Olmert became the leader | "per of the Knesset from 1999 to 2001. In 2001, he was appointed to the Minister of Religious Services by Ariel Sharon. Litzman served as Minister of Labor and Welfare, and Deputy Minister of Defense |
| "ra in the same committee from 2001 to 2004, serving under the chairmanship of Gideon Ezra. In 2003, Litzman was appointed the head of the committee's subcommittee on emergencies. His electoral district covered | "...Chairman of the Knesset's Health Committee from 1999 to 2005. In 2001, he was appointed by Prime Minister Ariel Sharon as the Ministry of Health's chief rabbi, a post he holds to this day |
| "rman of the Knesset from 2001 to 2008. In 2008, Litzman became Deputy Health Minister of Israel, a position he held until February 2019 when he resigned over allegations that he had obstruct | "...rector of the Ministry of Health from 2001 to 2006. During his tenure, he was responsible for implementing the Ministry's plans to improve the health services in Israel, particularly in the area of hospital management, health insurance reform, |
| "...Chairman of the committee. He was also a member of the Knesset's Economics Committee, the Committee on the Status of Women and the Committee on the Rights of the Child." | "...rman of the Committee on the Status of Women from 2001 to 2004. In 2003, he was appointed to serve on the Special Committee on the Rights of Women and the Family. Litzman also served as a |

---

**Box 18: Samples from fp32 vs. Amazon Bedrock for Llama-3 70B (Instruct)**

**Prompt:** Continue the paragraph. Do not output anything except the continuation to the paragraph. Start the continuation immediately. "He formally resigned as health minister on 26 November 2017, in protest of railroad repair work happ..."

| *fp32* | *Amazon Bedrock* |
|---|---|
| "ening on the railway line near his hometown of Likoni, which he believed was a threat to the ferry service that many locals relied on for transportation. His resignation was seen as a bold move, as he had previously been a close ally of the | "ening on a major bridge in the town of Temerín, which was causing significant disruptions to comm..." |
| "ening on a major rail line without adequate alternative transportation arrangements being made, which he felt would disproportionately affect the poor and vulnerable members of society. His decision sparked widespread criticism of the government's handling of the situation, and many saw it as a significant | ening a major downtown road, which was scheduled to coincide with a large international gathering of health experts in the city, and would likely cause significant traffic disruptions, all while the city's main hospital was still recovering from a recent devastating flood. |
| "...ening simultaneously on multiple sections of track, which he believed would cause unnecessary disruptions to commuters. Shortly after, he announced his candidacy for the opposition leader position, citing his desire to bring about change to the government's handling of infrastructure development projects. His | ening on a section of tracks that run close to the hospital where he worked as a doctor, which he claimed posed a health risk to patients and staff. |
| "ening on a major railway line, which he contended would lead to widespread disruptions and negatively impact the country's already strained healthcare system, a stance widely seen as a bold move by many in the medical community, who had long been calling for greater | "ening on a key rail line without consulting him, which he believed was a vital healthcare transportation route for patients and medical personnel." |
| "ening on a major railway line without proper planning and communication, which caused disruptions to healthcare services and put patients' lives at risk. This decision sparked widespread debate and criticism from various stakeholders, including medical professionals, politicians, and citizens, who saw it | ...ening on a Saturday, which he said was an "attack on the Jewish Sabbath" and would cause "massive disruptions" to the country's transportation system. |

---

**Box 19: Samples from fp32 vs. Together for Llama-3 70B (Instruct)**

**Prompt:** Continue the paragraph. Do not output anything except the continuation to the paragraph. Start the continuation immediately. "He formally resigned as health minister on 26 November 2017, in protest of railroad repair work happ..."

| *fp32* | *Together* |
|---|---|
| ening on the main railway line that connects Nairobi to the port city of Mombasa, which he claimed was being done without proper planning and would cause disruptions to the transportation of medical supplies, and potentially harm the healthcare system. His resignation was seen as | ening on a track that ran close to his home in the upscale Mumbai suburb of Bandra, which he claimed was causing noise pollution and disturbing the peace in his neighborhood. |
| "ening on a nearby site, which he claimed would cause noise pollution and disrupt the operations of a children's hospital that was being built on adjacent land, a project he had championed during his tenure as minister. The controversy surrounding the railroad repair work | "ening on the same day as the national memorial service for the victims of the Hwaseong capsizing disaster, which he felt was a grave mistake and showed a lack of respect for the deceased and their families." |
| "ening on a rail line that runs near the Prime Minister's residence, which he believed was unfairly prioritized over more urgent infrastructure projects in other parts of the country. His decision sparked widespread outrage and accusations of government corruption, leading to a series of | "ening on a key railway line linking the country's major urban centers, which he claimed was a risky and unnecessary project that would burden the taxpayers and was pushed through without adequate discussion or transparency. His resignation was seen as a major blow to the government |
| "ening on a major rail line that connects Moscow to the city of Kazan, which he believed would be detrimental to the already precarious healthcare system in the region. His resignation was seen as a bold move, as he was one of the most prominent | "ening on a main commuter line during rush hour, which he claimed would cause undue disruptions to the daily lives of citizens. His decision was met with a mix of surprise and admiration from the public, with many praising his willingness to take a stand against |
| "ening on a main highway in his constituency, which he claimed was not properly coordinated with local authorities and caused unnecessary disruptions to the daily lives of his constituents." | "...ening on a rail line near his hometown, which he believed would cause environmental damage and disrupt the local ecosystem. His resignation was met with widespread support from environmental groups and local residents, who saw him as a champion of their cause. Despite his departure |

---

**Box 20: Samples from fp32 vs. Fireworks for Llama-3.1 80B (Instruct)**

**Prompt:** Complete the code. Do not output anything except the completion. Start the continuation immediately.
```
from typing import List
def intersperse(numbers: List[int], delimeter: int) -> List[int]:
    """ Insert a number 'delimeter' between every two consecutive elements of input list `numbers'
    >>> intersperse([], 4)
    []
    >>> intersperse([1, 2, 3], 4)
    [1, 4, 2, 4, 3]
    """
```

**fp32:**
```
from typing import List
def intersperse(numbers: List[int], delimeter: int) -> List[int]:
    result = []
    for num in numbers:
        result.append(num)
        if numbers.index(num)!= len(numbers) - 1:
            result.append(delimeter)
    return result
```

---

```
from typing import List
def intersperse(numbers: List[int], delimeter: int) -> List[int]:
    """ Insert a number 'delimeter' between every two consecutive elements of input list `numbers'
    >>> intersperse([], 4)
    []
    >>> intersperse([1, 2, 3], 4)
    [1, 4, 2, 4, 3]
    """
    if not numbers:
        return []
    result = [numbers[0]]
    for num in numbers[1:]:
        result.extend([delimeter, num])
    return result
```

**Fireworks:**
```
from typing import List
def intersperse(numbers: List[int], delimiter: int) -> List[int]:
    """ Insert a number 'delimiter' between every two consecutive elements of input list `numbers'
    >>> intersperse([], 4)
    []
    >>> intersperse([1, 2, 3], 4)
    [1, 4, 2, 4, 3]
    """
    return [number for i, number in enumerate(numbers) for _ in range(2 - i % 2)] + [delimiter] * (int(len(numbers) / 2) if len
```

---

```
from typing import List
def intersperse(numbers: List[int], delimiter: int) -> List[int]:
    """ Insert a number 'delimiter' between every two consecutive elements of input list `numbers'
    >>> intersperse([], 4)
    []
    >>> intersperse([1, 2, 3], 4)
    [1, 4, 2, 4, 3]
    """
    result = []
    for i, num in enumerate(numbers):
        result.append(num)
        if i < len(numbers) - 1:
            result.append(delimiter)
    return result
```

---

**Box 21: Samples from Replicate vs. Perplexity for Llama-3 70B (Instruct)**

**Prompt:** Complete the code. Do not output anything except the completion. Start the continuation immediately.
```
from typing import List
def has_close_elements(numbers: List[float], threshold: float) -> bool:
    """ Check if in given list of numbers, are any two numbers closer to each other than
    given threshold.
    >>> has_close_elements([1.0, 2.0, 3.0], 0.5)
    False
    >>> has_close_elements([1.0, 2.8, 3.0, 4.0, 5.0, 2.0], 0.3)
    True
    """
```

**Replicate:** Here is the completed code:
```
from typing import List
def has_close_elements(numbers: List[float], threshold: float) -> bool:
    """ Check if in given list of numbers are any two numbers closer to each other than
    given threshold.
    """
    numbers.sort()  # sort the list in ascending order
    for i in range(len(numbers) - 1):  # iterate over the list
        if numbers[i + 1] - numbers[i] < threshold:
            return True  # if it is, return True
    return False  # if no such pair is found, return False
```

---

Here is the completed code:
```
from typing import List
def has_close_elements(numbers: List[float], threshold: float) -> bool:
    for i in range(len(numbers)):
        for j in range(i + 1, len(numbers)):
            if abs(numbers[i] - numbers[j]) < threshold:
                return True
    return False
```

**Perplexity:**
```
for i in range(len(numbers)):
    for j in range(i + 1, len(numbers)):
        if abs(numbers[i] - numbers[j]) < threshold:
            return True
return False
```

---

```
for i in range(len(numbers)):
    for j in range(i + 1, len(numbers)):
        if abs(numbers[i] - numbers[j]) < threshold:
            return True
return False
```

---

---

**Box 22: Samples from nf4 vs. Perplexity for Llama-3.1 8B (Instruct)**

**Prompt:** Complete the code. Do not output anything except the completion. Start the continuation immediately.
```
from typing import List
def intersperse(numbers: List[int], delimeter: int) -> List[int]:
    """ Insert a number 'delimeter' between every two consecutive elements of input list `numbers'
    >>> intersperse([], 4)
    []
    >>> intersperse([1, 2, 3], 4)
    [1, 4, 2, 4, 3]
    """
```

**nf4:**                                simulation!!!!!!!!!!!!!!!!!!!!!!!!!!!!!!!!!!!!!!!!!!!!!!!!!!!!!!!!!!!!!!!!!!!!!!!!!!!!!!!!!!!!!!!!

---

XCTAssertEqual!!!!!!!!!!!!!!!!!!!!!!!!!!!!!!!!!!!!!!!!!!!!!!!!!!!!!!!!!!!!!!!!!!!!!!!!!!!!!!!!!!!!!!!!!!!!!!!!

**Perplexity:** Here's the completion of the code:

```python
from typing import List

def intersperse(numbers: List[int], delimeter: int) -> List[int]:
    """ Insert a number 'delimeter' between every two consecutive elements of input list `numbers'
    >>> intersperse([], 4)
    []
    >>> intersperse([1, 2, 3], 4)
    [1, 4, 2, 4, 3]
    """
    if not numbers:
        return numbers
    result = [numbers[0]]
    for num in numbers[1:]:
        result.extend([delimeter, num])
    return result
```

---

Here is the completion of the code:
```python
from typing import List

def intersperse(numbers: List[int], delimeter: int) -> List[int]:
    """ Insert a number 'delimeter' between every two consecutive elements of input list `numbers'
    >>> intersperse([], 4)
    []
    >>> intersperse([1, 2, 3], 4)
    [1, 4, 2, 4, 3]
    """
    if not numbers:
        return []
    result = [numbers[0]]
    for num in numbers[1:]:
        result.extend([delimeter, num])
    return result
```

---

**Box 23: Samples from Amazon Bedrock vs. Perplexity for Llama-3 70B (Instruct)**

**Prompt:** Complete the code. Do not output anything except the completion. Start the continuation immediately.
```
from typing import List
def intersperse(numbers: List[int], delimeter: int) -> List[int]:
    """ Insert a number 'delimeter' between every two consecutive elements of input list `numbers'
    >>> intersperse([], 4)
    []
    >>> intersperse([1, 2, 3], 4)
    [1, 4, 2, 4, 3]
    """
```

**Amazon Bedrock:** Here is the completed code:
```
from typing import List

def intersperse(numbers: List[int], delimiter: int) -> List[int]:
    result = []
    for i, num in enumerate(numbers):
        result.append(num)
        if i < len(numbers) - 1:
            result.append(delimiter)
    return result
```

---

Here is the completed code:
```
from typing import List
def intersperse(numbers: List[int], delimeter: int) -> List[int]:
    """ Insert a number 'delimeter' between every two consecutive elements of input list `numbers`
    >>> intersperse([], 4)
    []
    >>> intersperse([1, 2, 3], 4)
    [1, 4, 2, 4, 3]
    """
    result = []
    for i in range(len(numbers)):
        result.append(numbers[i])
        if i < len(numbers) - 1:
            result.append(delimeter)
    return result
```

**Perplexity:**
```
result = []
for i, num in enumerate(numbers):
    result.append(num)
    if i < len(numbers) - 1:
        result.append(delimeter)
return result
```

---
```
result = []
for i, num in enumerate(numbers):
    result.append(num)
    if i < len(numbers) - 1:
        result.append(delimeter)
return result
```

---

**Box 24: Samples from Amazon Bedrock vs. Together for Llama-3 70B (Instruct)**

**Prompt:** Complete the code. Do not output anything except the completion. Start the continuation immediately.
```
from typing import List
def has_close_elements(numbers: List[float], threshold: float) -> bool:
    """ Check if in given list of numbers, are any two numbers closer to each other than
    given threshold.
    >>> has_close_elements([1.0, 2.0, 3.0], 0.5)
    False
    >>> has_close_elements([1.0, 2.8, 3.0, 4.0, 5.0, 2.0], 0.3)
    True
    """
```

**Amazon Bedrock:** Here is the completed code:
```
from typing import List

def has_close_elements(numbers: List[float], threshold: float) -> bool:
    """ Check if in given list of numbers are any two numbers closer to each other than
    given threshold.
    """
    for i in range(len(numbers)):
        for j in range(i + 1, len(numbers)):
            if abs(numbers[i] - numbers[j]) < threshold:
                return True
    return False
```

Here is the completed code:
```
from typing import List

def has_close_elements(numbers: List[float], threshold: float) -> bool:
    for i in range(len(numbers)):
        for j in range(i + 1, len(numbers)):
            if abs(numbers[i] - numbers[j]) < threshold:
                return True
    return False
```

**Together:**
```
for i in range(len(numbers)):
    for j in range(i + 1, len(numbers)):
        if abs(numbers[i] - numbers[j]) < threshold:
            return True
return False
```

```
for i in range(len(numbers)):
    for j in range(i + 1, len(numbers)):
        if abs(numbers[i] - numbers[j]) < threshold:
            return True
return False
```