# OpenReview forum: "Model Equality Testing: Which Model is this API Serving?"
_ICLR.cc/2025/Conference — ICLR 2025 Poster_

### Official Review · Reviewer_Sprd · 2024-10-31

**Soundness:** 4
**Presentation:** 3
**Contribution:** 3
**Rating:** 8
**Confidence:** 4

**Summary:**

The authors formulate the problem of testing whether two versions of a language models (either fully accessible, or accessible via API) are the same given two sets of samples from one model and another. They propose a simple but effective statistical test to distinguish between two sets of samples based on the Maximum Mean Discrepancy (MMD) string kernel method with the Hamming distance. They compare the chosen Hamming distance to other simple string distances and find that Hamming distance is well-suitable kernel to compare the samples from prompted language models. To evaluate the proposed MMD test, they verify that it shows high power for distinguishing different version of the same model or different models with a reasonably small number of samples, and degrades less w.r.t. the sequence length. Finally, they use the test to audit the API endpoints and find that a large portion of LLaMa 3 API providers give access to a different version of the model.

**Strengths:**

This paper touches upon a very timely problem of verifying the integrity of the black-box APIs and model versioning in the cases when full detailed for a particular API are not open. Namely, the important question this paper formulates and proposes a test for is whether two different versions of the same model are the same. This is an important test given that sometimes models are changed, compressed for efficiency or safety reasons, and it is important for practitioners to understand what to expect from a given API.

Overall, the paper is easy to follow, and methodology is sound. Experimental setup is sound enough to support the claims. Proposed method is simple and intuitive and builds upon the older works on string kernel based tests. I expect this paper will have a high impact within the language modelling community.

**Weaknesses:**

There are no critical weaknesses, in my opinion, only minor ones.

The choice of baselines is limited to different choices of string kernels.
I was a bit surprised to see Hamming distance to perform the best, and in general felt that discussion and intuition why it works well is not present.

**Questions:**

Q1. My intuition is that Hamming distance would be too sensitive to positions of tokens in the sentence by its construction, e.g. they don't capture the similarity between two shifted chunks of text. I would have started with Levenshtein distance, have you maybe tried it in your experiments?

Q2. How do you compute the kernels if a language model generates a shorter sentence compared to the maximal length?

Other questions:
- eq.4, should there z and z' be different?
- 162: what does accurate completion means?
- 210: is the only difference in the weights of the models, can generation algorithms also be different?
- Figure 2: a not well-readable sentence: "Curves first median power ..."
- 312: you write 1.1M unicode characters, but as far as I can check in Unicode 16.0 there are 155,063 unique characters, how it was computed?
- 326: "significantly different training data": do you use word significantly in statistical meaning here?
- 337: semantic caching, quantization: is it possible to put citations here?
- Figure 4 (right): what are the colours here?

---

> ### Author Response · Authors · 2024-11-16
> **Response**
>
> Thank you for your feedback! We refer the reviewer to our general comment, and we address your specific comments below.
>
> > Number of Unicode characters
>
>
>
> * Thank you for the correction --- you’re completely right. We mistakenly reported the number of possible outputs of Python’s `ord` function, which is larger than the actual number of Unicode codepoints in use. We will correct this in the manuscript.
>
> > How do you compute the kernels if a language model generates a shorter sentence compared to the maximal length?
>
> * We right-pad sequences (Line 181). All string kernels thus are aware of length (the padding token can be compared to other tokens).
>
> > The choice of baselines is limited to different choices of string kernels. I was a bit surprised to see Hamming distance to perform the best, and in general felt that discussion and intuition why it works well is not present.
>
> * We thank the reviewer for this writing comment! First, we point out that in addition to string kernels,  we also investigate non-MMD tests. For example, in Figure 2, we investigate both the L1 and two-sample chi-squared tests. Additionally, we refer the reviewer to Appendix C.3, where we compare the two-sample tests to a suite of goodness-of-fit baselines.
> * Our intuition for why the Hamming kernel is helpful is as follows: the main challenge of the Model Equality Testing problem is the high dimensionality of the space, in particular an exponential dependency on $L$, the completion length. In comparison to the one-hot kernel, for example, the Hamming kernel gives “partial credit” to samples that are more similar, even when they are not exact matches. Other kernels that have this property, e.g. the all-substrings kernel, and something based on the Levenstein distance as you suggested, probably have similar properties. We’ll look into the latter for the final manuscript.
>
> > “eq.4, should there z and z' be different?”
>
> * Yes, thank you! We will correct the notation to clarify that $z \neq z’$ in the first two summands.
>
> > “162: what does accurate completion means?”
>
>
>
> * Here we are referring to prompt distributions / tasks where there’s a clear notion of “correctness” for completions, e.g. through an automated evaluation metric. An example of this is the HumanEval task. The discussion is that for these tasks, it’s possible to define the MMD to use the automated evaluation metric, in which case we’re essentially comparing the task accuracy between samples. We then go on to discuss shortcomings of this approach in lines 163-165, in addition to the issue that many language tasks don’t have clear notions of accuracy / automated evaluation metrics. We’ll clarify the language in the final manuscript.
>
> > 210: is the only difference in the weights of the models, can generation algorithms also be different?
>
>
>
> * In Section 4.1, we only consider quantization and watermarking: quantization is a weight modification, while watermarking is a generation-time algorithm. We also refer the reviewer to our response to Reviewer qBpo, where we show additional experiments that detecting different temperature settings at generation time is very easy (100% power).
>
> > 326: "significantly different training data": do you use word significantly in statistical meaning here?
>
>
>
> * We do not; we’ll correct this language in the final manuscript.
>
> > 337: semantic caching, quantization: is it possible to put citations here?
>
>
>
> * Ah, for some reason, our links in the footnote are not working. We’ll fix the LaTeX issue in the final manuscript.
>
> > Figure 4 (right): what are the colours here?
>
>
>
> * The gray points indicate pairs where both distributions have accuracy &lt; 10% (line 402). This is to help clarify our statement in lines 413-418: “In several cases, the MMD is high but the accuracy difference is low. These are often when both task accuracies are low: the gray points in the figure highlight pairs where both distributions have accuracy &lt; 10%. In these cases, the MMD captures that there are multiple ways to be wrong for a task.”
>
> ---
>
> We thank the reviewer for your detailed feedback! Please let us know if there are additional things you would like to discuss.

---

> > ### Comment · Reviewer_Sprd · 2024-11-21
> > **Thank you for the response**
> >
> > I have read the response of the authors to me and other reviewers and maintain my high score. In my opinion, this is a solid and useful work, and I would be happy to see it accepted.

---

> ### Author Response · Authors · 2024-11-21
>
> Thank you for your feedback! We appreciate your comments.

---

### Official Review · Reviewer_jNji · 2024-11-02

**Soundness:** 2
**Presentation:** 3
**Contribution:** 2
**Rating:** 6
**Confidence:** 3

**Summary:**

The authors propose a method for determining whether a model black-box (i.e. an LM served through an API) differs from a reference model; usually a canonical model version or simply an earlier version of the API.
The proposed method relies on sampling texts from different models / APIs or the same one at different times and estimating their relation against a distance metric under some kernel. The authors propose a Hamming distance kernel as an empirically reliable approach for this estimation. They provide analysis of different foundation models and APIs over time gleaning insight into how inference service providers may affect inference reliability.

### Typos
- Line 081: extra article 'a' inserted before 'an API'

**Strengths:**

- The overall idea of providing more analytic tools for black-box models / APIs seems important and interesting. I think many people reliant on these black-box APIs are left at their mercy and have little means of understanding how different service providers could affect their system performance.
- The public release of generated samples offers valuable data for community use and future API change tracking.
- Providing a framework for analyzing APIs on non-classification based (i.e. difficult to evaluate) tasks seems fairly novel and is a good contribution.

**Weaknesses:**

- One immediate problem I see is that acquiring the reference distribution on a user-defined task requires setting up the reference LM anyways, which was something that is acknowledged to be inconvenient or infeasible in many cases. That would make this method impossible in certain scenarios.
- The hamming kernel's effectiveness across different tasks raises concerns, particularly for open-ended tasks like creative generation where diverse outputs may be desirable. The significance tests may struggle with high-variance outputs, potentially missing quality differences. Further analysis is needed on how task semantics and natural distribution affect the method's applicability. Additionally, using Wikipedia-based language modeling as the primary test task may underestimate output diversity due to the highly factual nature of the content and potential data contamination. This is partially addressed by the inclusion of HumanEval, but additional comparisons and/or analysis could strengthen the results.
- I didn't see any mention of two model customizations that I suspect may be popular: system prompt customization and generation-time safety interventions. It's not clear to me how the proposed method would behave under these customizations.

**Questions:**

- In Figure 4, you compare absolute task accuracy difference with MMD, but this is slightly less informative than showing non-absolute task difference. You imply on lines 415-418 that task accuracy difference are usually lower for the non-reference models, but this isn't necessarily obvious that a provider's interventions are always harmful to accuracy / quality. Could you provide some information about MMD and absolute accuracy?

---

> ### Author Response · Authors · 2024-11-16
> **Response (1/2)**
>
> Thank you for your feedback! We refer the reviewer to our general comments, and we address reviewer-specific comments below.
>
> > One immediate problem I see is that acquiring the reference distribution on a user-defined task requires setting up the reference LM anyways, which was something that is acknowledged to be inconvenient or infeasible in many cases. That would make this method impossible in certain scenarios.
>
> - We agree that this is a limitation. Unlike, e.g., the proof protocols suggested by the cryptography community for this problem (see Section 6), we require sample access to a reference in order to verify an API. Despite this, **we still believe that our two-sample setup is substantially more feasible than other possibilities**. We provide two examples for discussion:
>   - The cryptographic approach using proof protocols requires ongoing cooperation from the API provider; i.e., the API provider must generate a proof for every sample, which significantly increases the time to return each prediction. In contrast, a two-sample setup requires no cooperation from the API provider; we only need to obtain samples, which is the normal mode of operation.
>   - Another reasonable approach is the one-sample goodness-of-fit approach (see Appendix C.3). This would require log-probability access to the reference; in contrast, our two-sample setting only requires sample access to the reference. This opens up new applications (e.g., testing if an API has changed over time; we can cache samples from an earlier point in time and compare them to a later point in time).
> - Overall, we believe the two-sample setup is quite generalizable. In our work, we highlight several such applications: comparing APIs to locally inferenced weights (Section 5); comparing APIs to each other (Appendix C.8); comparing models before-and-after fine-tuning, even without access to the model weights (Section 4.2). Specifically, even in the case where we could not locally inference a model (Llama-3.1 405B investigations, Figure 4 left), we were still able to compare API implementations to each other (Appendix Figure 26).
>
> > The hamming kernel's effectiveness across different tasks raises concerns, particularly for open-ended tasks like creative generation where diverse outputs may be desirable. The significance tests may struggle with high-variance outputs, potentially missing quality differences. Further analysis is needed on how task semantics and natural distribution affect the method's applicability. Additionally, using Wikipedia-based language modeling as the primary test task may underestimate output diversity due to the highly factual nature of the content and potential data contamination. This is partially addressed by the inclusion of HumanEval, but additional comparisons and/or analysis could strengthen the results.
>
> - We discuss this at length in our general comment on prompt distributions.
> - Re: data contamination, this is a fascinating point. Surprisingly, we find that different models (presumably all trained on Wikipedia) remain easily distinguishable on Wikipedia prompts (Figure 3 lower left, Appendix Table 8), and we find that quantization still changes the model significantly on Wikipedia prompts (Section 4.1). These results seem to suggest that despite contamination, models still differ in their *distributions* over completions, even if, e.g., their greedy decoding results might match memorized content. Investigating this further could be an exciting direction for future work.

---

> ### Author Response · Authors · 2024-11-16
> **Response (2/2)**
>
> > In Figure 4, you compare absolute task accuracy difference with MMD, but this is slightly less informative than showing non-absolute task difference. You imply on lines 415–418 that task accuracy differences are usually lower for the non-reference models, but this isn’t necessarily obvious that a provider’s interventions are always harmful to accuracy/quality. Could you provide some information about MMD and absolute accuracy?
>
> - We did not mean to imply that the non-fp32 models are worse; in fact, some non-reference optimizations actually perform better than fp32. Below, we provide full results for two models, showing accuracies over 100 samples ($L=250$ per sample), averaged over 20 prompts:
>
> | **HumanEval avg accuracies** | **Llama-3 8B Instruct** | **Llama-3.1 8B Instruct** |
> |-------------------------------|--------------------------|----------------------------|
> | fp32                         | 17.05                   | 66.65                     |
> | fp16                         | 17.15                   | 39.45                     |
> | Amazon                       | 18.75                   | 51.80                     |
> | Azure                        | 19.10                   | 72.10                     |
> | DeepInfra                    | 18.00                   | 63.80                     |
> | Fireworks                    | 18.65                   | 67.85                     |
> | Groq                         | 14.10                   | 26.40                     |
> | int8                         | 14.55                   | 53.00                     |
> | nf4                          | 0.00                    | 0.00                      |
> | Perplexity                   | 0.00                    | 5.90                      |
> | Replicate                    | 21.50                   | (endpoint unavailable)    |
> | Together                     | 18.60                   | 66.20                     |
> | Watermark                    | 7.90                    | 55.00                     |
>
> - The direction of performance change (improvement or degradation) depends on the specific optimizations each API makes, including whether they have explicitly optimized for HumanEval when developing their inference stack. Note that when interpreting these results, one needs to account for sampling noise.
> - Our main argument in Figure 4 is that regardless of direction, one can somewhat predict the *magnitude* of change using the MMD. This is important in its own right; for example, it suggests that when the effect size returned by our test is large, users should do a manual examination for quality.
> - For HumanEval, this two-step procedure of test-and-then-check-accuracy may seem roundabout, as we have access to an automated evaluation metric. But as emphasized in Sections 1 and 3, our method aims to test *even for applications without automated evaluation*. Figure 4 shows that the MMD effect size provides a signal for when users should allocate manual resources for performance checks.
> - We thank the reviewer for raising this point and will include this discussion in the final manuscript.
>
> > I didn’t see any mention of two model customizations that I suspect may be popular: system prompt customization and generation-time safety interventions. It’s not clear to me how the proposed method would behave under these customizations.
>
> - These are very interesting questions to explore, and we leave this to future work. However, we would point the reviewer to our watermarking results, which represent an inference-time safety intervention intended to be more subtle than explicit safety refusals. For example, our qualitative samples in Appendix Box 10 suggest that watermarking is somewhat hard to identify with the naked eye, whereas explicit safety refusals are quite obvious. Despite this subtlety, our results suggest that watermarking is statistically detectable.
>
> ---
>
> We hope we’ve addressed your concerns. If so, would the reviewer consider increasing their score? Please let us know if there are additional topics to discuss.

---

> > ### Author Response · Authors · 2024-11-22
> >
> > Dear reviewer, we'd love to know if our response has addressed your concerns. Thanks!

---

> ### Comment · Reviewer_jNji · 2024-12-02
> **Thanks for your contribution. Raising my score.**
>
> While I believe there are some limitations to the methods proposed, there is significant novelty and contribution made by your work. I think a good number of my concerns have been addressed and raised my score.

---

> > ### Author Response · Authors · 2024-12-02
> >
> > Thank you! We appreciate your feedback and efforts to improve the paper.

---

### Official Review · Reviewer_vPAW · 2024-11-03

**Soundness:** 3
**Presentation:** 3
**Contribution:** 3
**Rating:** 6
**Confidence:** 4

**Summary:**

In this paper, the authors study the equivalence problem of LLMs in an API service environment. They formulate it as a two-sample test to assess the consistency of output distributions from different APIs. Ultimately, they use Maximum Mean Discrepancy (MMD) and achieve effective differentiation.

**Strengths:**

1. This paper is well-organized and written, making it easy to understand.

2. The problem addressed is valuable, as APIs have indeed become one of the mainstream forms of LLM applications.

3. The method design is reasonable and concise, and the experiments appear to be effective.

**Weaknesses:**

1. The study primarily focuses on the LLaMA series models. Although this aligns with the paper’s emphasis, I recommend that the authors verify the generalizability of MMD across more models.

2. The computation cost of MMD appears to be somewhat high.

**Questions:**

1. MMD performance declines with small sample sizes, which is likely the case in real-world scenarios. Although the paper discusses sample efficiency, additional experiments with smaller sample sizes could provide further insights into this limitation.

2. In practice, API services may not use the same model but rather similar models(like MOE). I suggest the authors provide some discussion on this point.

3. How does MMD perform in terms of efficiency in large-scale testing scenarios?

---

> ### Author Response · Authors · 2024-11-16
> **Response**
>
> Thank you for your feedback! We refer the reviewer to our general comment, and we address reviewer-specific comments below.
>
> > The study primarily focuses on the LLaMA series models. Although this aligns with the paper’s emphasis, I recommend that the authors verify the generalizability of MMD across more models.
>
> - This is a great suggestion! As the reviewer notes, we focus on the Llama models because they are the primary models served by inference providers in our case study in Section 5.
>   - For these, we refer the reviewer to Appendix C.1 for sample complexity results stratified by language model.
> - Below, we also provide **new results** in the style of Section 4.1 using Phi-3 Mini 4K, OLMO 7B, and Gemma-2 9B (all instruct versions). These are tested on Wikipedia at $N = 10m = 250$.
> - We observe that there is heterogeneity across models, including within the Llama family. The choice of model affects both the reference distribution $P$ and the effect of interventions like quantization ($Q$). For example, 4-bit quantization of the Llama and OLMO models is consistently noticeable. On the other hand, 8-bit quantization and watermarking have more inconsistent effects.
> - On average, however, our results in Section 4.1 still support our claim that the Hamming MMD is more powerful than other kernel choices.
>
> | Model               | Watermark (%)   | NF4 (%)       | Int8 (%)       | FPR (fp32) (%)  |
> |---------------------|-----------------|---------------|----------------|-----------------|
> | Llama-3 8B          | 99.9 (0.1)     | 92.5 (3.6)    | 79.4 (7.7)     | 4.9 (0.8)       |
> | Llama-3.1 8B        | 62.3 (3.5)     | 100.0 (0.0)   | 7.8 (1.8)      | 5.1 (0.9)       |
> | Llama-3 70B         | 97.1 (1.2)     | 100.0 (0.0)   | 100.0 (0.0)    | 6.8 (1.1)       |
> | Llama-3.1 70B       | 53.5 (6.3)     | 100.0 (0.0)   | 100.0 (0.0)    | 5.5 (0.7)       |
> | Mistral 7B          | 71.6 (6.3)     | 88.8 (4.4)    | 29.8 (6.1)     | 3.9 (1.0)       |
> | OLMo 7B             | 46.5 (7.7)     | 98.6 (0.8)    | 36.2 (7.8)     | 6.3 (0.5)       |
> | Gemma-2 9B          | 43.3 (2.1)     | 12.4 (1.2)    | 6.0 (0.7)      | 5.6 (0.6)       |
> | Phi-3 Mini          | 74.8 (4.4)     | 63.4 (4.6)    | 23.7 (1.3)     | 6.1 (0.9)       |
>
> > MMD performance declines with small sample sizes, which is likely the case in real-world scenarios. Although the paper discusses sample efficiency, additional experiments with smaller sample sizes could provide further insights into this limitation.
>
> - We respectfully point out that our primary findings are centered on the small-sample regime. Namely, in Section 4.1, we find that at $N = 10m$, i.e., an average of 10 samples per prompt, the Hamming kernel outperforms other kernels in defining a powerful test. For context, 10 samples per prompt, for distributions over 20–25 prompts, results in an audit that costs < 1 dollar in our Section 5 experiments (for most providers, the audit actually costs < \$0.02).
> - Our results in Sections 4.2, 4.3, and 5 all operate in this small sample regime.
> - For further discussion, we refer the reviewer to Appendix C.2, where we investigate smaller sample sizes induced by decreasing $m$ (with $N = 10m$).
>
> > The computation cost of MMD appears to be somewhat high.
>
> - This is an insightful point, and we view computational cost as **one of the main advantages of the Hamming kernel**. The Hamming kernel is fast because its computation is parallelizable via broadcasting; other string kernels, including the all-substrings kernel we study, scale poorly with $N$.
> - We discuss computational cost in Section 3.
>
> ---
>
> We hope we’ve addressed your concerns and increased your confidence in the paper. Please let us know if there are additional things to discuss.

---

> > ### Comment · Reviewer_vPAW · 2024-11-18
> > **Response to Author Rebuttal**
> >
> > Thank you for the author's response, which has addressed my concerns well. Overall, I believe this is a solid paper that tackles an important and novel problem within the current LLM ecosystem. I will maintain my positive score.

---

> > > ### Author Response · Authors · 2024-11-21
> > >
> > > Thank you! We appreciate your feedback.

---

### Official Review · Reviewer_qBpo · 2024-11-04

**Soundness:** 2
**Presentation:** 3
**Contribution:** 2
**Rating:** 6
**Confidence:** 4

**Summary:**

The paper proposes Model Equality Testing to understand the differences between models served by APIs. The proposed method is a two-sample test that uses various prompts along with multiple generations of a given prompt and tries to infer if two instances of generations come from the same distributions using the Maximum Mean Discrepancy kernel. The proposed method is studied empirically to understand its power in identifying the differences for models served by many APIs.

**Strengths:**

The paper is overall well written, organized and easy to follow. The application of two sample testing using the Maximum Mean Discrepancy kernel is novel for studying distributions coming from different models.

**Weaknesses:**

- I’m not convinced of the significance of this problem. Can this problem be solved through policy if the users ask from providers to disclose such changes? The API provides can be even motivated to charge users differently based on the optimizations done to models.
- Evaluations can be stronger:
    - The authors claim that the proposed method works using an average of 10 samples per prompt across 20-25 prompts. Since the paper relies on empirical analysis, I would love to see more analysis backing this claim/more guidelines around these requirements. For instance, do I still need to provide 10 samples per prompt if I generate the samples using 0 temperature? What about if my prompts are all from a niche topic versus if I’m testing across various different unrelated topics? Does one still obtain high power using 20-25 prompts?
    - The evaluations are done using samples from English, German, Spanish, French, and Russian Wikipedia. I’d love to see more diversity in the evaluation tasks. For instance, consider showing the results for coding tasks, which can help with the broader applicability of the proposed method.

**Questions:**

- Regarding the kernel:
    - For the choice of the kernel, you are basically trading off bias and variance. I think this might be good to call out in the manuscript. My understanding is that as L goes to infinity, the hamming kernel approaches the universal one-hot kernel. Similarly as L goes to 0, one biases towards the first tokens. I’d love to understand the implications of this bias-variance tradeoff on tasks that require more creativity like writing.
    - Also L is set to 50 and the sensitivity to L is studied until L=50. I think 50 tokens for generations is typically on the lower end when it comes to LLM-based applications. I’d recommend studying it at least until 10 folds.
    - I’d think L is affected by the vocabulary size of the tokenizer. Did the authors study the effect of L under different tokenizers?
    - The kernel checks for token equivalence but an important piece to capture might be around semantic equivalence. Have the authors considered other kernels that can capture semantic equivalence?
- Evaluations:
    - Figure 3 lower left. The power of detecting Llama3.1 8B from Llama3.1 8B (int8) and (watermark) are 0.07 and 0.32, respectively, which are very low. I’d love to hear the author's thoughts on this as one of the premises of the method is to be able to detect such changes in the model.
    - Consider adding an analysis that shows that the method does not detect any changes if the two samples come from the same model. It would be great to show the detection power using samples obtained at different temperatures.
    - Can you extend the analysis in Figure 3 lower left to other models? One thing I’m interested to understand is that the method is able to detect different model sizes and looking at Figure 3 on the right hand side, the models from the same family look very close in terms of distance.
- Nit: Mention that L is the number of tokens in the manuscript.

---

> ### Author Response · Authors · 2024-11-16
> **Response (1/2)**
>
> Thank you for your feedback! Based on your suggestions, we ran 3 additional experiments (including the effect of task creativity and detecting temperature mismatches) and identified an area of improvement in the writing (extended discussion about the effect of $L$). We address your comments below.
>
> > I’m not convinced of the significance of this problem. Can this problem be solved through policy if the users ask from providers to disclose such changes? The API providers can even be motivated to charge users differently based on the optimizations done to models.
>
> - This is a great question. Please see our general comment on contributions.
>
> > “The authors claim that the proposed method works using an average of 10 samples per prompt across 20-25 prompts. Since the paper relies on empirical analysis, I would love to see more analysis backing this claim/more guidelines around these requirements….What about if my prompts are all from a niche topic versus if I’m testing across various different unrelated topics? Does one still obtain high power using 20-25 prompts?...The evaluations are done using samples from English, German, Spanish, French, and Russian Wikipedia. I’d love to see more diversity in the evaluation tasks. For instance, consider showing the results for coding tasks, which can help with the broader applicability of the proposed method.”
>
> - Please see our general comment on prompt distributions for additional experiments.
>
> > “Figure 3 lower left. The power of detecting Llama3.1 8B from Llama3.1 8B (int8) and (watermark) are 0.07 and 0.32, respectively, which are very low. I’d love to hear the author's thoughts on this as one of the premises of the method is to be able to detect such changes in the model.”
>
> - This is a wonderful question. First, we clarify why these numbers are lower than those in Figure 2 or the general comment on prompt distributions: as explained on Line 310, in Figure 3, we have changed the tokenization used for testing from the individual model’s tokenizer ($|\mathcal V| \leq 128256, L=50$) to a Unicode tokenization ($|\mathcal V| = 155063, L=1000$). Recall that the size of the sample space is $m|\mathcal V|^L$; increasing $L$ by 20x makes the testing problem *significantly* more high dimensional. As one would extrapolate from Figure 2 (middle), a larger $L$ hurts power, although the MMD-based tests are more robust to this harder setting than other baselines.
>     - An additional opportunity for improvement is in Section 5, where we suffer an additional loss in power when using the composite null hypothesis setting; see Appendix Table 9 for a discussion.
> - These power losses reflect why the Model Equality Testing problem is hard: language is a high-dimensional space! Our contributions are, again, to formalize this problem and evaluate a first set of baselines on this task; we think there’s room for a fruitful line of work on developing more powerful tests for this regime.
> - We would like to clarify, however, that these power losses do not affect the soundness of our conclusions. In Section 4.1, our conclusion is that the Hamming kernel outperforms other kernels; in Section 4.2, we provide an “existence result” that the Hamming kernel can detect finetuning; in Section 4.3 we show that *even in the reduced power setting,* model swaps are easy to detect (and that the MMD provides a way to discuss distances between models); and finally, in Section 5, *even in the reduced power setting,* 11/31 API endpoints are different than reference weights released by Meta. We do not claim that our results have eliminated false negatives; only that they control false positives.
> - Finally, for the sake of discussion, we confirm (as in Figure 2 left) that power losses can be compensated for by collecting more samples. Below we repeat the Unicode-space experiments with $N=50m$ instead of $N=10m$. If we were to use this sample size in Section 5, each audit would still cost < $5.
>
> | Model              | Watermark (%)    | nf4 (%)          | int8 (%)         | FPR (fp32, %)   |
> |---------------------|------------------|------------------|------------------|-----------------|
> | Llama-3 8B (N=10m) | 62.0 (4.8)       | 37.8 (5.2)       | 29.5 (5.7)       | 4.8 (1.2)       |
> | Llama-3 8B (N=50m) | 100.0 (0.0)      | 99.5 (0.5)       | 95.9 (2.4)       | 6.6 (1.2)       |
>
> > “I’d think $L$ is affected by the vocabulary size of the tokenizer. Did the authors study the effect of $L$ under different tokenizers?”
>
> - Yes; see discussion above.
>
> > “Also $L$ is set to 50 and the sensitivity to $L$ is studied until $L=50$. I think 50 tokens for generations is typically on the lower end when it comes to LLM-based applications. I’d recommend studying it at least until 10 folds.”
>
> - See discussion above; we already study up to $L=1000$ under the Unicode tokenization.
> - Additionally, we conduct all HumanEval and UltraChat experiments with $L=250$ in token space.

---

> ### Author Response · Authors · 2024-11-16
> **Response (2/2)**
>
> > Consider adding an analysis that shows that the method does not detect any changes if the two samples come from the same model.
>
> - Please see the italicized numbers in Appendix Tables 8, 9, 10, and 12.
> - In addition, we’ve printed the empirical FPRs on tables in this rebuttal.
>
> > Can you extend the analysis in Figure 3 lower left to other models? One thing I’m interested to understand is that the method is able to detect different model sizes and looking at Figure 3 on the right hand side, the models from the same family look very close in terms of distance.
>
> - Please see Appendix Table 8. Even within the Llama family, it’s possible to detect when a larger model of the family (e.g., 70B) has been replaced with a smaller one (e.g., 8B) or vice versa. For example, when Llama-3 70B is replaced with Llama-3 8B, we can detect this with 98% power.
>
> > For instance, do I still need to provide 10 samples per prompt if I generate the samples using 0 temperature?
>
> - Since greedy decoding is deterministic, the suggested testing problem is trivial: one only needs to check exact match between one sample from $P_\text{greedy}$ and from $Q_\text{greedy}$ to determine if $P_\text{greedy} = Q_\text{greedy}$. As we discussed in Section 1, this is equivalent to checking if the modes of the unscaled next-token distributions match.
> - In contrast, we are interested in whether the overall distributions match. In theory, if the non-temperature-adjusted distributions match, then adjusting both by the same temperature parameters will also result in matching distributions.
>
> > It would be great to show the detection power using samples obtained at different temperatures.
>
> - This is a great suggestion; we provide preliminary results below and will include other models in the final manuscript. In general, detecting incorrect temperature settings is very easy, because the distributions are quite different.
>
> | Model (Reference: Temperature = 1.0) | Temperature = 0.5 (%) | Temperature = 1.5 (%) |
> |---------------------------------------|------------------------|------------------------|
> | Llama-3 8B                            | 100.0 (0.0)           | 100.0 (0.0)           |
>
> > The kernel checks for token equivalence but an important piece to capture might be around semantic equivalence. Have the authors considered other kernels that can capture semantic equivalence?
>
> - This is an interesting suggestion! As we discussed in Section 3, checking for task accuracy is a variant of this; we do some examinations about task accuracy in Figure 4 (right).
> - However, a more general experiment one might consider could use an embedding model as the feature vector $\phi$. Unfortunately, this would make testing slower and more expensive for the user (requires embedding all samples), which is why we did not explore it. Compare this to the Hamming kernel, which can be done on CPU and takes less than a second to compute in wall clock time.
> - Although outside the scope of this work, we think this (or in general, learning $\phi$) could be a fruitful line of future inquiry to develop stronger tests.
>
> ---
>
> We hope we’ve addressed the reviewer’s concerns. Please let us know if there are additional things to discuss.

---

> > ### Author Response · Authors · 2024-11-22
> >
> > Dear reviewer, we'd love to hear if our response has addressed your concerns. Thank you!

---

> > > ### Comment · Reviewer_qBpo · 2024-11-29
> > >
> > > Thank you authors! This was a strong rebuttal and the authors diligently addressed all my comments. I've read other reviewers' comments as well. The significance of the work was a big limitation I was considering in my previous evaluation. Authors' response combined with other reviewers' assessments helped mitigate my concern. Therefore, I've raised my score to 6.

---

> > > > ### Author Response · Authors · 2024-11-29
> > > >
> > > > We're glad to have addressed your comments -- thank you so much for your feedback! It's improved the paper.

---

### Author Response · Authors · 2024-11-16
**General comment (1/2): recap of paper contributions**

Thank you to all of the reviewers for your thoughtful feedback! We’ve worked to answer your questions in individual threads, and we’d like to address two comments to all reviewers.

The first comment is a summary of our paper contributions. **Our first contribution is the formalization of the Model Equality Testing problem.** We believe formalizing this problem is impactful in and of itself --- a sentiment shared by 3/4 reviewers (vPAW, jNji, and Sprd) --- and we would like to engage with Reviewer qBpo’s comment about problem significance to explain why we believe this is the case.


### 1. Why is this problem impactful?



* This paper addresses long-standing, yet understudied, pain points for average ML-as-a-service (MLaaS) users: API providers have always been opaque about how endpoints are implemented and/or changing. As we write in Section 1, the issue is not just that providers are incentivized to *consciously* optimize models to save costs; inference stacks can also contain *unintentional* bugs, and all providers, including for closed-weight models like GPT, can fail to disclose when underlying models are finetuned.
* **Under the status quo, users are unable to rigorously verify these questions for themselves** --- they can only wonder about what they’re given. **This is a real problem that users already experience:** for evidence of this, we direct the reviewers to samples of non-academic “chatter” below:
    * Users attempting to build collective understanding about perceived differences in Llama 3.1 implementations: [[Reddit post]](https://www.reddit.com/r/LocalLLaMA/comments/1ebzd5a/warning_the_quality_of_hosted_llama_31_may_vary/) [[Twitter thread]](https://x.com/pandaashwinee/status/1816966288905998829?s=46&t=fmzCzaSrcpp2jNTx6-oaKw)
    * Users trying to reason about whether OpenAI APIs are changing over time: [[Reddit post]](https://www.reddit.com/r/OpenAI/comments/zoez3p/is_it_just_me_or_is_chatgpt_getting_worse_by_the/)
    * Academic papers documenting such changes, including in pre-LLM classification APIs: [[1]](https://arxiv.org/abs/2307.09009) [[2]](https://arxiv.org/abs/2107.14203) [[3]](https://dl.acm.org/doi/10.1145/3630106.3659047) [[4]](https://arxiv.org/abs/2210.08674)
* Our work empowers users to test for these changes themselves, on their own tasks of interest, for less than $1. Further, we are the first to formalize this problem for the general LLM setting. As discussed in Section 6 and by Reviewer jNji, the language setting is novel because of the high dimensionality of language, and because we lack automated evaluation metrics for many language use cases.

### 2. Why is our technical solution appropriate?



* By nature of being the first to study this problem, we are also the first to propose a solution to the problem. Our other contributions include identifying the MMD framework as a productive lens for formulating new tests (narrowing the search space of tests to the search space of string kernels), and in Section 4, we benchmark a first set of baselines for this task. Our empirical finding is that the computationally efficient Hamming kernel is a strong baseline, especially compared to the other baselines we imagine.
* This technical solution is appropriate because users can apply such tests *now,* even without provider or policy cooperation. We take particular care to focus on the small-sample regime. As a result, our tests are **immediately applicable and affordable:** the tests in Section 5 cost &lt; $1 each. They directly empower a user to check an API’s implementation and to track if APIs have changed over time, which are all current pain points.
* We think Reviewer qBpo’s question about policy interventions is interesting, but we point out that even if policy could be passed for this problem, **enforcing policy would still require technical tools for auditing**, such as the one we have proposed. Further, in a vacuum absent of technical auditing tools, companies are unlikely to spontaneously become more transparent --- inference stacks are proprietary and valuable trade secrets. Rather, we hope that **because we empower users to test these questions for themselves, this bottom-up pressure will organically hold providers accountable** --- if not to full transparency, then to testable claims of equivalency.

Our remaining contributions are to provide case studies (Sections 4.2-5) illustrating how these tests can detect common changes. As we state in Sections 1 and 7, we believe there’s room for a fruitful line of work to develop more powerful tests; our final contribution is to open-source a large dataset and codebase to facilitate future research.

---

> ### Author Response · Authors · 2024-11-16
> **General comment (2/2): effect of the prompt distribution on power**
>
> Reviewers qBpo and jNji both asked about the dependence of power on the prompt distribution. We agree this is an important question that deserves more discussion. We present new experiments below.
>
> ### Setup
>
> - Reviewer qBpo asked for power results on coding tasks, and Reviewer jNji asked about power results on creative tasks. The underlying question is whether the “open-endedness” of the task affects power: intuitively, one might expect that more creative tasks lead to completion distributions that are higher entropy, which might make testing harder.
> - To answer this question, we evaluate power across several prompt distributions.
>   - On the most “constrained” side of the spectrum, we experimented with HumanEval (code); on the most “creative” side, UltraChat (chatbot dialogues), and in the middle, language modeling with Wikipedia. Please see Appendix Boxes 3 and 4 for samples of what these prompts look like.
>   - To answer Reviewer qBpo’s question about concentration of topics (“What about if my prompts are all from a niche topic versus if I’m testing across various different unrelated topics?”), we additionally experiment with tightly concentrated prompts (just Wikipedia in English, just Wikipedia in French, etc.) vs. diverse prompt distributions (Wikipedia mixed with HumanEval and UltraChat).
>   - We tested power of the Hamming kernel against the local alternatives (nf4, int8, watermark) at $ N=10m $ with tests conducted in token space (50-token completions). Standard errors are reported following the procedure in Section 4.1.
>   - For comparison, we also provide the results of the one-hot kernel. This is to analyze whether our claims about the relative ordering of which kernels are stronger in the $ N=10m $ regime are correct.
>
> ### Results
>
> In the subsequent comment, the top table gives results for the Hamming kernel, and the bottom for the one-hot kernel. We make these observations:
>
> - Reviewer jNji hypothesized that more creative tasks might be harder to detect: in this case, we should see UltraChat < Wikipedia < HumanEval across all alternatives in power. Instead, the creativity level of tasks doesn't clearly correlate with detection difficulty. int8 quantization is hard to detect across several distributions, while nf4 quantization is only hard to detect on English Wikipedia.
> - Reviewer qBpo wondered if tightly clustered topics are harder to detect than mixtures of topics. Instead, topic clustering (single-language vs mixed content) shows no consistent pattern in detection difficulty.
> - The prompt distribution mainly affects the difficulty of detecting quantization, not watermarking.
> - The Hamming kernel consistently outperforms the one-hot baseline on all prompt distributions, matching our main conclusion of Section 4.1 about the relative strength of kernels.
> - As a reminder, power can always be increased by increasing the sample size. For example, collecting $ N=50m $ samples instead of $ 10m $ would still cost < $5 per audit.
>
> **Summary:** While the prompt distribution $ \pi $ does affect power, the effect is specific to the $ (P, Q) $ pair and inconsistent. In general, our recommendation of the Hamming MMD as the strongest baseline holds.

---

> > ### Author Response · Authors · 2024-11-16
> > **General comment (2/2): effect of the prompt distribution on power [Tables]**
> >
> > ### Results for Hamming MMD
> >
> > | Dataset                          | Number of prompts ($ m $) | watermark (%) | nf4 (%)         | int8 (%)       | FPR (fp32, %) |
> > |----------------------------------|----------------------------|---------------|-----------------|----------------|---------------|
> > | HumanEval (constrained)          | 20                         | 99.0 (0.0)    | 100.0 (0.0)     | 32.0 (0.0)     | 6.0 (0.0)     |
> > | Wikipedia (English only)         | 25                         | 97.9 (0.7)    | 49.2 (3.9)      | 11.0 (1.2)     | 6.0 (1.2)     |
> > | Wikipedia (French only)          | 25                         | 100.0 (0.0)   | 100.0 (0.0)     | 100.0 (0.0)    | 5.5 (0.7)     |
> > | Wikipedia (German only)          | 25                         | 98.3 (0.8)    | 71.4 (7.7)      | 27.1 (6.2)     | 4.9 (0.4)     |
> > | Wikipedia (Spanish only)         | 25                         | 100.0 (0.0)   | 100.0 (0.0)     | 100.0 (0.0)    | 5.2 (1.0)     |
> > | Wikipedia (Russian only)         | 25                         | 98.4 (0.5)    | 70.7 (5.3)      | 24.0 (4.4)     | 5.3 (0.6)     |
> > | Wikipedia (All languages, mixed) | 25                         | 99.9 (0.1)    | 92.5 (3.6)      | 79.4 (7.7)     | 4.9 (0.8)     |
> > | UltraChat (creative)             | 20                         | 89.0 (0.0)    | 100.0 (0.0)     | 4.0 (0.0)      | 4.0 (0.0)     |
> > | Mixed (Wikipedia + UltraChat)    | 25                         | 81.2 (5.1)    | 100.0 (0.0)     | 10.0 (1.7)     | 5.5 (0.9)     |
> > | Mixed (Wikipedia + UltraChat + HumanEval) | 25                  | 95.4 (1.2)    | 100.0 (0.0)     | 35.1 (5.1)     | 3.6 (0.6)     |
> >
> > #### Results for One-hot MMD
> >
> > | Dataset                          | Number of prompts (\( m \)) | watermark (%) | nf4 (%)         | int8 (%)       | FPR (fp32, %) |
> > |----------------------------------|----------------------------|---------------|-----------------|----------------|---------------|
> > | HumanEval (constrained)          | 20                         | 46.0 (0.0)    | 77.0 (0.0)      | 16.0 (0.0)     | 5.0 (0.0)     |
> > | Wikipedia (English only)         | 25                         | 6.2 (0.8)     | 5.0 (0.8)       | 4.3 (0.6)      | 1.4 (0.3)     |
> > | Wikipedia (French only)          | 25                         | 4.3 (0.7)     | 5.8 (0.6)       | 4.8 (0.4)      | 1.0 (0.2)     |
> > | Wikipedia (German only)          | 25                         | 4.1 (0.7)     | 4.4 (0.5)       | 4.3 (0.6)      | 0.9 (0.2)     |
> > | Wikipedia (Spanish only)         | 25                         | 4.4 (0.6)     | 5.9 (0.7)       | 3.9 (0.5)      | 1.6 (0.3)     |
> > | Wikipedia (Russian only)         | 25                         | 3.9 (0.7)     | 4.9 (0.7)       | 2.9 (0.7)      | 1.4 (0.4)     |
> > | Wikipedia (All languages, mixed) | 25                         | 3.5 (0.3)     | 5.0 (0.7)       | 4.3 (1.0)      | 0.6 (0.2)     |
> > | UltraChat (creative)             | 20                         | 7.0 (0.0)     | 48.0 (0.0)      | 4.0 (0.0)      | 1.0 (0.0)     |
> > | Mixed (Wikipedia + UltraChat)    | 25                         | 9.1 (1.4)     | 5.2 (0.6)       | 6.6 (0.7)      | 3.6 (0.9)     |
> > | Mixed (Wikipedia + UltraChat + HumanEval) | 25                  | 21.8 (2.1)    | 31.5 (2.9)      | 4.9 (0.9)      | 4.7 (0.9)     |

---

### Author Response · Authors · 2024-11-25

We sincerely thank all the reviewers for their time and effort in evaluating our work. As the discussion deadline approaches, we would appreciate any feedback on our rebuttal, especially if there are remaining concerns that have not yet been addressed.

---

### Meta-Review · Area_Chair_to6R · 2024-12-19

**Metareview:**

In this paper, the authors utilize Maximum Mean Discrepancy (MMD) to evaluate whether large language models (LLMs) offered by different cloud vendors are identical or have been modified in significant or subtle ways. From a technical perspective, the paper offers limited innovation. However, for practitioners deploying models across various cloud service providers, the findings are important. They highlight that the models can differ and may exhibit varying behaviors depending on the server or vendor, which is a crucial consideration for real-world applications.

The reviewers have a positive opinion of this paper and believe it would be a valuable addition to the conference. However, the Area Chair has reservations about the paper's long-term relevance, as it primarily addresses a transitory problem that is unlikely to impact real-world applications in the future.

**Additional Comments On Reviewer Discussion:**

The authors and reviewers engaged during the discussion phase, resulting in one reviewer becoming very positive about the paper. However, the reviewers did not engage with me directly. I believe the value of the paper is quite limited, and if not for the high score given by one reviewer, I would recommend rejection. While I have recommended acceptance, I believe the paper is borderline and could reasonably be rejected.

If the decision is to reject, please include the following paragraph in the meta-review. Alternatively, I am happy to add it myself if you inform me of your decision.

--- additional paragraph for meta-review ----

The reviewers were generally positive about the paper, but none were willing to champion it for the conference. While the topic is relevant, any researcher or practitioner deploying models across different machines or vendors would likely assess model consistency with their specific application in mind, rather than relying on a general tool like the one proposed. Additionally, vendors have a strong incentive to ensure their models perform consistently, especially as these LLMs are adopted for critical processes. As a result, this paper may have limited longevity once LLMs become widely used in commercial, high-stakes applications.

---

### Decision · Program_Chairs · 2025-01-22

Accept (Poster)